# LEMoN: Label Error Detection using Multimodal Neighbors

## Abstract

Large repositories of image-caption pairs are essential for the development of vision-language models. However, these datasets are often extracted from noisy data scraped from the web, and contain many mislabeled instances. In order to improve the reliability of downstream models, it is important to identify and filter images with incorrect captions. However, beyond filtering based on image-caption embedding similarity, no prior works have proposed other methods to filter noisy multimodal data, or concretely assessed the impact of noisy captioning data on downstream training. In this work, we propose, theoretically justify, and empirically validate LEMoN, a method to automatically identify label errors in image-caption datasets. Our method leverages the multimodal neighborhood of image-caption pairs in the latent space of contrastively pretrained multimodal models to automatically identify label errors. Through empirical evaluations across eight datasets and ten baselines, we find that LEMoN outperforms the baselines by over 3% in label error detection, and that training on datasets filtered using our method improves downstream captioning performance by 2 BLEU points.

## 1 Introduction

Machine learning datasets used to train and finetune large vision, language, and vision-language models frequently contain millions of labeled instances (Schuhmann et al., 2021; Li et al., 2022; Wang et al., 2022a; Changpinyo et al., 2021). Prior work highlights that some instances in such datasets may be mislabeled (Northcutt et al., 2021b; Luccioni & Rolnick, 2023; Liao et al., 2021; Beyer et al., 2020; Plummer et al., 2015), as seen in Figure 1. This is especially problematic in settings such as healthcare, where the reliability of downstream models may depend on the quality of data used for pretraining (Chen et al., 2024; Liu et al., 2023; Longpre et al., 2023).

Identifying and correcting label errors in existing datasets at scale would lead to more reliable and accurate models in the real world (Zhu et al., 2022; Vasudevan et al., 2022; Liao et al., 2021; Beyer et al., 2020). However, given the large size of such datasets, manual detection of errors is practically infeasible. This is evidenced by the growth of models trained on noisy data with the web (Li et al., 2022; Wang et al., 2022a; Liu et al., 2024), or with model generated pseudo-labels (Menghini et al., 2023; Lai et al., 2023).

Machine learning (ML) based approaches to automatically identifying label errors have also been proposed in prior work (Pleiss et al., 2020; Swayamdipta et al., 2020; Liang et al., 2023; Bahri et al., 2020; Zhu et al., 2022; Northcutt et al., 2021a). However, we identify two critical limitations: (1) a majority of such works are *unimodal*: i.e., they only utilize image-based representations and detection strategies, and (2) many of the best-performing approaches depend on having access to a model already trained on the downstream tasks of interest (Pleiss et al., 2020; Swayamdipta et al., 2020). We hypothesize that applying a neighborhood-based approach to multimodal representations in the form of image-text pairs can improve label error detection without requiring task-specific training, which may be costly and/or domain specific for some datasets.

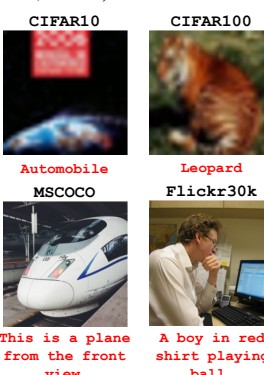

Figure 1: Samples from classification and captioning datasets discovered to be mislabeled by our method.

Additionally, a common assumption made in prior works is that each label is one-of-k classes (Bahri et al., 2020; Zhu et al., 2022). The vast majority of label error detection methods proposed in prior works are hence for *classification* datasets. In contrast, datasets used to train large vision-language

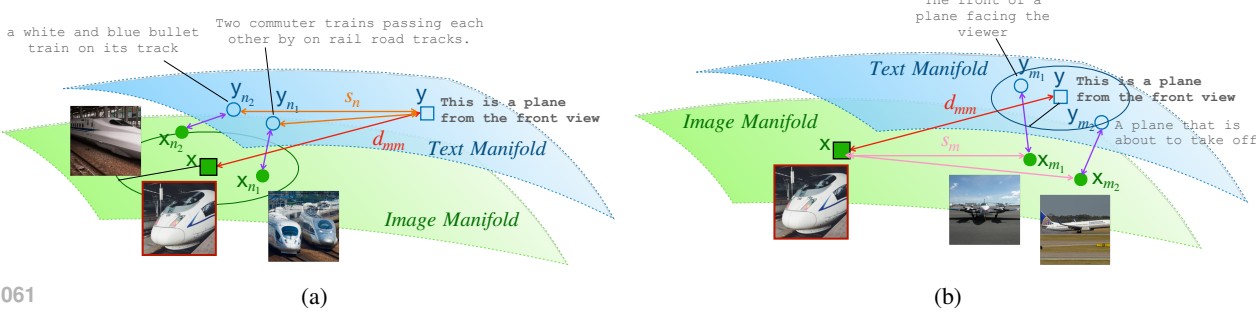

Figure 2: **Outline of LEMON**, our proposed method for multimodal label error detection. We demonstrate LEMON on a real sample from the MSCOCO dataset, where an image of a train ($\mathbf{x}$) is mislabeled as $\mathbf{y}$ = "This is a plane from the front view". (a) We compute the simple CLIP similarity $d_{mm}(\mathbf{x}, \mathbf{y})$. We then find the nearest neighbors of $\mathbf{x}$ in the image space ($\mathbf{x}_{n_j}$) and compute the distance between the corresponding texts and $\mathbf{y}$ to compute the score component $s_n$. (b) To compute the score component $s_m$, we find the nearest neighbors of $\mathbf{y}$ in the text space ($\mathbf{y}_{m_k}$), and compute the distance between the corresponding images and $\mathbf{x}$.

models contain natural language labels such as image captions (Li et al., 2022; 2023; Wang et al., 2022a). Methods to filter out instances with noisy labels – e.g., based on the similarity of image and caption representations – have been utilized in prior work with some success (Li et al., 2022; Kang et al., 2023) for such datasets. However, to the best of our knowledge, no prior works have proposed or rigorously compared methods to identify errors in datasets with natural language labels, or assessed the impact of detection on downstream tasks like image captioning.

In this work, we propose LEMON– **L**abel **E**rror detection using **M**ultim**o**dal **N**eighbors – a method for multimodal label error detection, which can be applied to image-text pairs in datasets such as MSCOCO (Lin et al., 2014). While prior techniques utilize unimodal neighbors for label error detection, LEMON leverages multi-modal neighborhoods derived using contrastively pretrained vision-language models such as Contrastive Language-Image Pretraining (CLIP) (Radford et al., 2021). Specifically, in addition to considering pairwise image-text distances, we also retrieve nearest neighbors in the image and text space as illustrated in Figure 2. This is motivated about rich neighborhood geometry in the joint embedding space of multimodal models (Liang et al., 2022; Schrodi et al., 2024). We then compute distance scores with neighbors in each modality and combine these into a single score measuring the likelihood of a label error, with the intuition that higher discordance (or higher distance) with neighbors indicates a higher chance of label error. We validate the utility of these scores across eight datasets, including one in a healthcare setting, and compare to over ten baselines.

Our key contributions and findings are as follows:

- We propose LEMON, a novel, theoretically justified multimodal method capable of detecting label errors in large image-caption datasets (Section 3).
- We show that LEMON outperforms all downstream task-unaware baselines for label error detection in the classification setting, by up to 3.4% AUROC (Section 6.1).
- We empirically show that LEMON outperforms baselines in three out of four captioning datasets, by up to 3.9% AUROC (Section 6.1).
- We demonstrate that LEMON improves performance on downstream classification and captioning models by filtering out data predicted to be label errors. (Section 6.2).
- Finally, we verify that the predictions generated by LEMON are meaningful through a real world analysis of LEMON on existing datasets without known label errors (Section 6.5).

## 2 RELATED WORKS

**Label Noise Detection** Noisy and incorrect labels (Beyer et al., 2020) in training data may lead to decreased or "destabilized" (Northcutt et al., 2021a; Luccioni & Rolnick, 2023) performance on downstream tasks (Chen et al., 2023; Northcutt et al., 2021b). Two orthogonal approaches can be taken to reduce the adverse effects of such labels: developing methods to learn in the presence of label errors (Cui et al., 2020; Natarajan et al., 2013; Huang et al., 2023), and/or detecting and filtering out instances with label errors (Zhu et al., 2024). In this work, we focus on the latter approach. Prior approaches (Swayamdipta et al., 2020; Bahri et al., 2020; Pleiss et al., 2020; Northcutt et al., 2021a; Liang et al., 2023; Wu et al., 2020; Kim et al., 2021) for automatic label error detection include

relying on the training dynamics of task-specific downstream models (Swayamdipta et al., 2020) and neighborhood-based strategies (Bahri et al., 2020; Grivas et al., 2020). Some of these techniques are fully supervised (Northcutt et al., 2021a; Chen et al., 2023) or unsupervised (Pleiss et al., 2020; Swayamdipta et al., 2020; Grivas et al., 2020; Bahri et al., 2020), use pre-trained generative models (Gertz et al., 2024) or are fully training-free approaches (Zhu et al., 2022; Liang et al., 2023). Previous approaches for label error detection closest to this work includes deep k-nearest neighbor (deep k-NN) methods using k-NN entropy on vector space embeddings (Bahri et al., 2020; Grivas et al., 2020) and SimiFeat (Zhu et al., 2022) which employs a local neighborhood-based voting or ranking for noise identification. In contrast to these methods, our work enhances label noise detection by harnessing information across *multiple data modalities*, such as image and text. Finally, though prior works may have utilized the idea of semantic neighborhoods in multimodal data (e.g. for cross-modal retrieval) (Thomas & Kovashka, 2020; 2022), we believe we are the first to extend concepts to the task of label error detection by proposing a novel, theoretically justified score for identifying label errors.

**Contrastive Learning**   Contrastive learning is a representation learning strategy that contrasts positive and negative pairs of data instances (Chen et al., 2020; Misra & Maaten, 2020; Balestriero et al., 2023) to learn an embedding space. The core idea is to embed similar data points (positive pairs) closer together than dissimilar data points (negative pairs) (Schroff et al., 2015; Sohn, 2016; Oord et al., 2018). In this work, we primarily utilize pre-trained models that use the CLIP loss (where the pre-training objective is predicting which text caption goes is paired with which image) for jointly embedding image and text data (Radford et al., 2021).

**Image Captioning**   The goal of image captioning is to describe a given image (Fu et al., 2024) in natural language. Prior approaches for caption generation have included supervised training of end-to-end models from scratch (Wang et al., 2022b; Lin et al., 2022; Hu et al., 2023; Xu et al., 2015; Fu et al., 2024). More recently, vision-language models pretrained on large datasets of noisy image-caption pairs extracted from the web (Li et al., 2022; 2023; Wang et al., 2022a) – such as CC12M (Changpinyo et al., 2021) – have been utilized for captioning. Some of the pretraining tasks include image-text contrastive learning, image-text matching, and/or retrieval (Li et al., 2022), as well as general purpose text generation conditioned on an input image (Wang et al., 2022a). Given that datasets for training such large models are noisy (Kang et al., 2023), several steps have been utilized in prior work to filter out noisy captions during training. The most common strategy involves computing the similarity between representations of the image and caption text using another pretrained model (e.g., CLIP) prior to training (Kang et al., 2023). Another approach in training the BLIP (Li et al., 2022) model is to synthetically generate noisy captions and train a classifier to distinguish between high quality captions and noisy captions with a cross-entropy loss (Li et al., 2022). To the best of our knowledge, no previous work has conducted a comprehensive comparison of various strategies for label error detection in captioning datasets.

**Multimodal Neighborhood Methods**   Previous studies (Li et al., 2021; Thomas & Kovashka, 2020; 2022; Huang et al., 2024; Liang et al., 2022; Cai et al., 2023) have examined the geometry of neighborhood spaces in multimodal models, often with the goal of improving representation learning (Huang et al., 2024; Li et al., 2021) or retrieval (Thomas & Kovashka, 2020; 2022). The closest related work is Thomas & Kovashka (2022), where the authors use the semantic neighborhood of multimodal models to identify samples with high semantic diversity using text-based neighbors of neighbors. However, as the objective of their work is different from ours, their proposed discrepancy and diversity scores would not provide a signal for label error in our setting. We further clarify this in Appendix B, and will empirically compare against their discrepancy score as a baseline. Although prior works have utilized the idea of multimodal neighbors in other settings, we believe we are the first to apply it to the setting of label error detection.

## 3   LEMoN: Label Error Detection using Multimodal Neighbors

We are given a dataset $\mathcal{D} = \{(\mathbf{x}, \mathbf{y})_{i=1}^N\}$ consisting of two modalities $\mathbf{x} \in \mathcal{X}$ and $\mathbf{y} \in \mathcal{Y}$. For example, $\mathcal{X}$ may represent the set of all natural images, and $\mathcal{Y}$ may represent the set of all English text, or a restricted subset such as $\{\texttt{cat}, \texttt{dog}, ...\}$. We assume the existence of, but not access to, an oracle $f^* : \mathcal{X} \times \mathcal{Y} \to \{0, 1\}$, which is able to assign a binary mislabel indicator $z_i = f^*(\mathbf{x}_i, \mathbf{y}_i)$ to each sample in $\mathcal{D}$. Here, $z_i = 1$ indicates that the sample is mislabeled, and $z_i = 0$ indicates that the sample is correctly labeled. Our goal is to output a score $s \in \mathbb{R}$ with some model $s := f(\mathbf{x}, \mathbf{y})$ such that

$$\text{AUROC} = \mathbb{E}_{(\mathbf{x}, \mathbf{y}) \sim \mathbb{P}(\cdot | z=1), (\mathbf{x}', \mathbf{y}') \sim \mathbb{P}(\cdot | z=0)}[\mathbf{1}_{f(\mathbf{x}, \mathbf{y}) \geq f(\mathbf{x}', \mathbf{y}')}]$$

is maximized. Prior works have alternatively aimed to maximize the F1 score, optimizing over a threshold $t$:

$$\text{F1} = \max_{t \in \mathbb{R}} \frac{2 \cdot \mathbb{P}(z = 1 | s \geq t) \cdot \mathbb{P}(s \geq t | z = 1)}{\mathbb{P}(z = 1 | s \geq t) + \mathbb{P}(s \geq t | z = 1)}$$

Here, building on prior work for label error detection in unimodal data (Bahri et al., 2020; Zhu et al., 2022), we propose a method for $f$ based on nearest neighbors, summarized in Figure 2. Suppose we have a query sample $(\mathbf{x}, \mathbf{y})$[1]. Define $B(\mathbf{x}, r) := \{x' \in \mathcal{X} : d_{\mathcal{X}}(\mathbf{x}, \mathbf{x}') \leq r\}$, the ball of radius $r$ around $\mathbf{x}$, and $B(\mathbf{y}, r)$ similarly. Let $r_k(\mathbf{x}) := \inf\{r : |B(\mathbf{x}, r) \cap \mathcal{D}| \geq k\}$, the minimum radius required to encompass at least $k$ neighbors. Then, we define $\{\mathbf{x}_{n_1}, \mathbf{x}_{n_2}, ..., \mathbf{x}_{n_k}\} := B(\mathbf{x}, r_k(\mathbf{x})) \cap \mathcal{D}$ the top $k$ nearest neighbors of $\mathbf{x}$, and $\{\mathbf{y}_{m_1}, \mathbf{y}_{m_2}, ..., \mathbf{y}_{m_k}\} := B(\mathbf{y}, r_k(\mathbf{y})) \cap \mathcal{D}$ the top $k$ nearest neighbors of $\mathbf{y}$[2]. We assume that the neighbors are sorted in order of ascending distance, e.g. $d_{\mathcal{X}}(\mathbf{x}, \mathbf{x}_{n_2}) \geq d_{\mathcal{X}}(\mathbf{x}, \mathbf{x}_{n_1})$.

If $\mathcal{Y}$ is a small discrete set, we could choose $d(\mathbf{y}, \mathbf{y}') = \mathbf{1}_{\mathbf{y} = \mathbf{y}'}$. If $\mathcal{X}$ or $\mathcal{Y}$ are unstructured or high dimensional, we assume access to multimodal encoders $h_\theta = (h_\theta^{\mathcal{X}}, h_\theta^{\mathcal{Y}})$, where $h_\theta^{\mathcal{X}} : \mathcal{X} \to \mathbb{R}^d$ and $h_\theta^{\mathcal{Y}} : \mathcal{Y} \to \mathbb{R}^d$. Here, $h_\theta$ may be a CLIP model (Radford et al., 2021) trained on a large internet corpus, or, as we show later, it may be sufficient to train $h_\theta$ from scratch only on $\mathcal{D}$. Then, we could naturally use simple distance metrics in the embedding space, such as the cosine distance $d_{\mathcal{X}}(\mathbf{x}, \mathbf{x}') = d_{\cos}(h_\theta^{\mathcal{X}}(\mathbf{x}), h_\theta^{\mathcal{X}}(\mathbf{x}')) = 1 - \frac{h_\theta^{\mathcal{X}}(\mathbf{x})^T h_\theta^{\mathcal{X}}(\mathbf{x}')}{||h_\theta^{\mathcal{X}}(\mathbf{x})||_2 \cdot ||h_\theta^{\mathcal{X}}(\mathbf{x}')||_2}$. Our proposed score is the linear combination of three terms:

$$s = f(\mathbf{x}, \mathbf{y}) = d_{mm}(\mathbf{x}, \mathbf{y}) + \beta s_n(\mathbf{x}, \mathbf{y}, \mathcal{D}) + \gamma s_m(\mathbf{x}, \mathbf{y}, \mathcal{D}), \tag{1}$$

where $\beta, \gamma \geq 0$ are hyperparameters. Here, $d_{mm}(\mathbf{x}, \mathbf{y}) := d_{\cos}(h_\theta^{\mathcal{X}}(\mathbf{x}), h_\theta^{\mathcal{Y}}(\mathbf{y}))$ is the multimodal distance, which has been shown empirically to provide a meaningful signal in prior label error detection work (Liang et al., 2023; Kang et al., 2023). We thus use this distance as the basis, and augment it with two additional terms based on nearest neighbors:

$$s_n(\mathbf{x}, \mathbf{y}, \mathcal{D}) = \frac{1}{k} \sum_{j=1}^{k} d_{\mathcal{Y}}(\mathbf{y}, \mathbf{y}_{n_j}) e^{-\tau_{1,n} d_{\mathcal{X}}(\mathbf{x}, \mathbf{x}_{n_j})} e^{-\tau_{2,n} d_{mm}(\mathbf{x}_{n_j}, \mathbf{y}_{n_j})}, \tag{2}$$

where $(\mathbf{x}_{n_j}, \mathbf{y}_{n_j}) \in \mathcal{D}$, and $\tau_{1,n}, \tau_{2,n} \geq 0$ are hyperparameters. This corresponds to finding the nearest neighbors of $\mathbf{x}$ in $\mathcal{X}$ space, then averaging the distance between their *corresponding* modality in $\mathcal{Y}$ and $\mathbf{y}$. We weight this average with two additional terms. The $\tau_{1,n}$ term corresponds to downweighting neighbors which are far from $\mathbf{x}$. Intuitively, this is useful when $k$ is too large for $\mathbf{x}$ and not all neighbors are relevant, and can be thought of as an adaptive $k$. The $\tau_{2,n}$ term corresponds to downweighting neighbors which are themselves likely to be mislabeled. If $(\mathbf{x}_{n_j}, \mathbf{y}_{n_j})$ is itself mislabeled, then $d_{\mathcal{Y}}(\mathbf{y}, \mathbf{y}_{n_j})$ would contribute an erroneous signal to whether $(\mathbf{x}, \mathbf{y})$ is mislabeled, and we thus want to downweight those instances.

The third term is analogous to $s_n$, but uses neighbors of $\mathbf{y}$:

$$s_m(\mathbf{x}, \mathbf{y}, \mathcal{D}) = \frac{1}{k} \sum_{j=1}^{k} d_{\mathcal{X}}(\mathbf{x}, \mathbf{x}_{m_j}) e^{-\tau_{1,m} d_{\mathcal{Y}}(\mathbf{y}, \mathbf{y}_{m_j})} e^{-\tau_{2,m} d_{mm}(\mathbf{x}_{m_j}, \mathbf{y}_{m_j})}, \tag{3}$$

where $(\mathbf{x}_{m_j}, \mathbf{y}_{m_j}) \in \mathcal{D}$, and $\tau_{1,m}, \tau_{2,m} \geq 0$ are hyperparameters. Crucially, note that notationally, $\mathbf{x}_{n_j} \neq \mathbf{x}_{m_j}$, and $\mathbf{y}_{n_j} \neq \mathbf{y}_{m_j}$. Specifically, $\mathbf{y}_{n_j}$ corresponds to the $\mathcal{Y}$ modality of nearest neighbors taken in $\mathcal{X}$ space, and $\mathbf{y}_{m_j}$ corresponds to the nearest neighbors of $\mathbf{y}$ taken in $\mathcal{Y}$ space.

We note that our method is a generalization of several prior methods. When $\beta = \gamma = 0$, the method is equivalent to CLIP similarity (Liang et al., 2023). When $\beta$ is large, $\tau_{1,n} = \tau_{2,n} = \gamma = 0$, and $d(\mathbf{y}, \mathbf{y}_{n_j}) = \mathbf{1}_{\mathbf{y} = \mathbf{y}_{n_j}}$, the method is equivalent to Deep kNN (Bahri et al., 2020). An algorithm outline and high-level description of the method can be found in Appendix C.

Our method contains several hyperparameters: $k, \beta, \gamma, \tau_{1,n}, \tau_{2,n}, \tau_{1,m}$, and $\tau_{2,m}$. When there is a validation set with known mislabel flags, we perform a grid search over $k$, and use numerical optimization methods to search for an optimal value of the remaining hyperparameters which maximize label error detection performance on this set, which we describe further in Section 5.2. We refer to our method in this setting as LEMON$_{\text{OPT}}$. We will empirically show that only a few hundred labeled validation samples may be sufficient to achieve optimal performance in this setting.

---

[1]One could take, for any $i$, $(\mathbf{x}, \mathbf{y}) := (\mathbf{x}, \mathbf{y})_i$, $D' := D \setminus \{(\mathbf{x}, \mathbf{y})_i\}$

[2]We will use a subscript $n_j$ to index nearest neighbors in $\mathcal{X}$, and subscript $m_j$ for neighbors in $\mathcal{Y}$.

When there is no labeled validation set available, we will show that our method is fairly robust to these hyperparameter choices, and that choosing a set of reasonable fixed values for these hyperparameters yields nearly comparable results. We refer to our method in this setting as LEMON$_{\text{FIX}}$.

## 4 THEORETICAL ANALYSIS

First, we demonstrate that the embedding models trained via the contrastive multimodal objective are natural noisy label detectors.

**Theorem 4.1** (Contrastive Multimodal Embedding Models Detect Noisy Labels). *Let $\mathcal{Y} = \mathbb{R}$ and consider a training dataset $\mathcal{D}$. Suppose that $\hat{h}_\theta^{\mathcal{X}} : \mathcal{X} \to \mathbb{R}^d$ is an embedding function, and $\hat{h}_\theta^{\mathcal{Y}} : \mathcal{Y} \to \mathbb{R}^d$ is a Lipschitz continuous embedding function with constant $L_{\mathcal{Y}} > 0$, meaning that for all $y, y' \in \mathcal{Y}$,*

$$\left\| \hat{h}_\theta^{\mathcal{Y}}(y) - \hat{h}_\theta^{\mathcal{Y}}(y') \right\|_2 \leq L_{\mathcal{Y}} |y - y'|.$$

*For an input $x \in \mathcal{X}$ and its corresponding positive label $y \in \mathcal{Y}$, let $\eta$ be a random variable drawn from a normal distribution: $\eta \sim \mathcal{N}(0, \sigma^2)$. Define a noisy label $y' = y + \eta$. Let $d_{mm}(u, v) = ||u - v||_2$, which is proportional to $\sqrt{d_{cos}(u, v)}$ when $||u||_2 = ||v||_2 = 1$. Then, with probability at least $\delta(\epsilon) = 1 - 2\Phi\left(-\frac{\epsilon}{\sigma}\right)$, where $\epsilon > 0$ and $\Phi$ is the cumulative distribution function of the standard normal distribution, the following inequality holds:*

$$d_{mm}\left(\hat{h}_\theta^{\mathcal{X}}(x), \hat{h}_\theta^{\mathcal{Y}}(y')\right) \geq d_{mm}\left(\hat{h}_\theta^{\mathcal{X}}(x), \hat{h}_\theta^{\mathcal{Y}}(y)\right) - L_{\mathcal{Y}}\, \epsilon.$$

When $L_{\mathcal{Y}}$ is small, this means that the score for the mislabeled sample cannot be much lower than the score for the positive pair with high probability. Thus, we can see that multimodal embeddings are inherently capable of detecting mislabeled pairs, ensuring the distance between the embeddings of positive pairs is smaller than that of negative pairs. This motivates the use of $d_{mm}$ in LEMON and in prior work (Kang et al., 2023; Liang et al., 2023).

Next, we show that our multimodal kNN scores (Equations (2) and (3)) provide a signal for label error. Suppose there exists a "paraphrase function" $\mathcal{H} : \mathcal{Y} \to \mathcal{P}(\mathcal{Y})$, where $\mathcal{P}$ denotes the powerset, such that for a particular sample $(x, y)$ with $\mathcal{H}(y) = (\bar{y}_1, \bar{y}_2..., )$, $(x, \bar{y}_i)$ is considered correctly labeled for all $\bar{y}_i \in \mathcal{H}(y)$. Informally, $\mathcal{H}$ outputs the set of all possible captions which correctly describe $x$. Similarly define $\mathcal{J}(x)$, which outputs the set of images with identical semantics as $x$.

**Assumption 1** (Structure of $\mathcal{H}, \mathcal{J}$):

- Let $(x', y')$ be an arbitrary sample. If $y' \notin \mathcal{H}(y)$, then $x' \notin \mathcal{J}(x)$.
- Let $(x', y')$ be an arbitrary mislabeled sample. Then, $\forall y'' \in \mathcal{H}(y')$, $x'' \notin \mathcal{J}(x')$.

**Assumption 2** (Distribution of Distances): *Let $(X, Y)$ be a randomly drawn sample.*

- $\forall\, X' \notin \mathcal{J}(X) : d_{\mathcal{X}}(X, X') \overset{\text{iid}}{\sim} \mathcal{N}(\mu_1, \sigma_1^2)$.
- $\forall\, \bar{X} \in \mathcal{J}(X) : d_{\mathcal{X}}(X, \bar{X}) \overset{\text{iid}}{\sim} \mathcal{N}(\mu_2, \sigma_2^2)$.

We empirically validate this assumption in Appendix A.3.

Let $N_k(Y) = \{Y_{m_1}, ..., Y_{m_k}\}$ denote the nearest neighbors of $Y$ in the text space. Let $\frac{1}{k}|\mathcal{H}(Y) \cap N_k(Y)| = \zeta_Y$, a random variable. Suppose that $\frac{1}{k}|\{i : (X_{m_i}, Y_{m_i}) \text{ is mislabeled}\}| = p$ is constant for all samples in the support of $(X, Y)$.

Let $S_m(X, Y) = \frac{1}{k}\sum_{Y_{m_i} \in N_k(Y)} d_{\mathcal{X}}(X, X_{m_i})$, which is identical to the proposed Equation (3) with $\tau_1 = \tau_2 = 0$.

**Theorem 4.2** (AUROC of kNN Score). *Let $(X, Y)$ be a randomly selected correctly labeled sample, and $(X', Y')$ a randomly selected incorrectly labeled sample. Under Assumptions 1 and 2:*

$$\mathbb{P}(S_m(X', Y') > S_m(X, Y)) = 1 - \Phi(\frac{-\mu}{\sigma})$$

*where $\mu = \mathbb{E}[\zeta_Y](1-p)(\mu_1 - \mu_2), \sigma = \sqrt{\frac{\mathbb{E}[\zeta_Y](1-p)\sigma_2^2 + (2 - \mathbb{E}[\zeta_Y](1-p))\sigma_1^2}{k} + \text{Var}(\zeta_Y)(1-p)^2(\mu_2 - \mu_1)^2}$, and $\Phi$ is the Gaussian CDF.*

This provides an expression for the detection AUROC of the score $S_m$. The same expression can be derived for $S_n$ by symmetry.

**Lemma 4.3** (Non-random Signal of kNN Score). *If the following three conditions hold: (1) $p < 1$, (2) $\mathbb{E}[\zeta_Y] > 0$, (3) $\mu_1 > \mu_2$. Then,*

$$\mathbb{P}(S_m(X', Y') > S_m(X, Y)) > 0.5$$

Under these conditions, $S_m$, our proposed multimodal neighborhood score, provides a better than random signal at detecting mislabeled samples. Full proofs can be found in Appendix A.

## 5 EXPERIMENTS

### 5.1 DATASETS

We evaluate our method using eight datasets, as shown in Table 1. Four datasets (`cifar10`, `cifar100`, `stanfordCars`, `miniImageNet`) are label error detection datasets from the classification setting. The four remaining datasets are image captioning datasets. For `mscoco` and `flickr30k`, we use the Karpathy split (Karpathy & Fei-Fei, 2015). In the remaining datasets, we randomly split each dataset into three parts in an 80-10-10 ratio: training or reference set for the label detection method, validation set for hyperparameter selection, and test set for testing label error detection performance.

Table 1: Classification and captioning datasets. $n$ is the number of samples. In the main paper, results shown are for the bolded noise type with 40% noise level for synthetic noise. Performance on remaining noise types can be found in the appendices.

| Dataset | $n$ | | | Domain | | Noise Types |
|---|---|---|---|---|---|---|
| | Train | Validation | Test | Image | Text | |
| `cifar10` | 40,000 | 5,000 | 5,000 | Natural images | Object labels | {***human*** (Wei et al., 2021), *sym.*, *asym.*} |
| `cifar100` | 40,000 | 5,000 | 5,000 | Natural images | Object labels | {***human*** (Wei et al., 2021), *sym.*, *asym.*} |
| `miniImageNet` (Jiang et al., 2020) | 49,419 | 24,710 | 24,710 | Natural images | Object labels | {***real***} |
| `stanfordCars` (Jiang et al., 2020) | 13,501 | 6,751 | 6,752 | Car images | Car year and model | {***real***} |
| `mscoco` (Lin et al., 2014) | 82,783 | 5,000 | 5,000 | Natural images | Captions | {***cat.***, *noun*, *random*} |
| `flickr30k` (Young et al., 2014) | 29,000 | 1,014 | 1,000 | Natural images | Captions | {***noun***, *random*} |
| `mmimdb` (Arevalo et al., 2017) | 15,552 | 2,608 | 7,799 | Movie Posters | Plot summaries | {***cat.***, *noun*, *random*} |
| `mimiccxr` (Johnson et al., 2019) | 368,909 | 2,991 | 5,159 | Chest X-rays | Radiology reports | {***cat.***, *random*} |

#### 5.1.1 NOISE TYPES

In `cifar10` and `cifar100`, we utilize a dataset collected in prior work (Wei et al., 2021) with human mislabels (*human*). We also follow prior work (Zhu et al., 2022) in experimenting with class symmetric (*sym.*) and class asymmetric (*asym.*) synthetic noise. For `stanfordCars` and `miniimagenet`, we use datasets from Jiang et al. (2020), which contain noise from real-world (*real*) web annotators .

For the four captioning datasets, we devise several ways to inject synthetic noise of prevalence $p$. The simplest way is to randomly select $p$ fraction (*random*) of the samples and assign their text modality to be that of another random caption. In datasets where additional metadata is available (`mscoco`: object category, `mmimdb`: genre of movie, `mimiccxr`: disease label), we can randomly swap the caption with that of another sample from the same category (*cat*). Finally, in all captioning datasets except `mimiccxr`, we tag each token of each caption with its part-of-speech using SpaCy (Honnibal & Montani, 2017), and then randomly assign a selected sample's text modality to be from another sample with at least one noun in common (*noun*). Dataset processing details are also in Appendix D.

Our motivation for these noise types is to simulate an array of realistic label corruptions that one might face in the real world. We recognize that the resulting synthetic dataset may not have exact noise level $p$, as e.g. a randomly selected caption may actually be correct for the image, as well as noise in the base datasets, which we explore in Section 6.5. Unless otherwise stated, results shown in the main paper are for the bolded noise type in Table 1, with 40% synthetic noise. Additional results for other noise types can be found in the appendices.

### 5.2 MODEL SELECTION AND EVALUATION

We run LEMON on each dataset, using the training split of each dataset to compute nearest neighbors. In classification datasets, we use the discrete metric $d_{\mathcal{Y}}(\mathbf{y}, \mathbf{y}') = \mathbf{1}_{\mathbf{y} = \mathbf{y}'}$. In all other cases and for $d_{\mathcal{X}}$, we utilize cosine or euclidean distance computed in the embedding space of a pretrained CLIP model, selecting the best distance metric on the validation set for LEMON$_{\text{OPT}}$, and keeping the distance as the cosine distance for LEMON$_{\text{FIX}}$. In `mimiccxr`, we use BiomedCLIP (ViT-B/16) (Zhang et al., 2023b), and we use OpenAI CLIP ViT-B/32 (Radford et al., 2021) for all other datasets. A full list of hyperparameters for our method and the baselines are in Appendix G.

Table 2: Label error detection performance across classification datasets. We separate AUM, Datamap, and Confident learning, as they require training a classifier from scratch. Bold denotes best score within each training approach. A full version of this table with AUPRC can be found in Appendix I.1.

| Method | cifar10 | | cifar100 | | miniImageNet | | stanfordCars | |
|---|---|---|---|---|---|---|---|---|
| | **AUROC** | **F1** | **AUROC** | **F1** | **AUROC** | **F1** | **AUROC** | **F1** |
| AUM | **98.3** (0.1) | **94.0** (0.1) | **92.2** (0.2) | **83.8** (0.4) | 83.1 (0.2) | 75.3 (0.2) | 70.5 (2.4) | 62.3 (1.2) |
| Datamap | 98.2 (0.1) | 93.4 (0.5) | 91.8 (0.2) | 83.5 (0.6) | **85.0** (0.2) | **77.0** (0.2) | **72.3** (1.8) | **64.9** (2.1) |
| Confident | 93.7 (0.4) | 92.7 (0.5) | 74.1 (1.7) | 69.3 (2.0) | 70.5 (0.2) | 54.7 (0.4) | 61.0 (0.5) | 43.4 (1.6) |
| CLIP Logits | 95.5 (0.2) | 88.0 (0.5) | 84.9 (0.7) | 75.5 (0.5) | 90.0 (0.2) | **82.5** (0.2) | 68.8 (0.7) | 64.9 (0.4) |
| CLIP Sim. | 93.8 (0.1) | 86.9 (0.4) | 78.5 (0.6) | 69.2 (1.3) | 89.3 (0.2) | 81.3 (0.5) | 69.8 (0.6) | 61.7 (0.8) |
| Simifeat-V | 90.6 (0.3) | 88.0 (0.4) | 79.5 (0.0) | 73.1 (0.5) | 68.2 (0.3) | 55.0 (0.5) | 63.7 (1.2) | 43.7 (1.5) |
| Simifeat-R | 90.7 (0.3) | 88.1 (0.5) | 79.7 (0.2) | 73.6 (0.6) | 68.0 (0.3) | 54.7 (0.4) | 63.5 (1.3) | 43.4 (1.6) |
| Discrepancy | 77.1 (1.9) | 68.2 (1.9) | 66.0 (1.5) | 51.9 (1.8) | 79.4 (0.3) | 69.8 (0.4) | 65.7 (0.7) | 59.9 (0.4) |
| Deep k-NN | 97.8 (0.1) | 92.5 (0.5) | 87.4 (0.3) | 78.0 (0.3) | 83.2 (0.2) | 75.2 (0.4) | 71.4 (0.6) | 65.3 (0.9) |
| LEMoN$_{\text{FIX}}$ (Ours) | 97.7 (0.2) | - | 88.9 (0.7) | - | 89.5 (0.2) | - | 72.6 (0.7) | - |
| LEMoN$_{\text{OPT}}$ (Ours) | **98.1** (0.0) | **93.1** (0.2) | **90.8** (0.0) | **81.3** (0.2) | **90.2** (0.2) | **82.3** (0.1) | **73.1** (0.5) | **67.3** (1.0) |

For LEMoN$_{\text{OPT}}$, we select the hyperparameter combination that maximizes F1 on a labeled validation set. We report the AUROC, AUPRC, and F1 for this model. For LEMoN$_{\text{FIX}}$, we fix the hyperparameters at the following reasonable values: $k = 30$, $\beta = \gamma = 5$, $\tau_{1,n} = \tau_{1,m} = 0.1$, and $\tau_{2,n} = \tau_{2,m} = 5$. We report AUROC and AUPRC, as the F1 requires additional information to compute a threshold for the score. We recognize that access to such a validation set as in LEMoN$_{\text{OPT}}$ may be unrealistic, but we will empirically show that (1) our method is fairly robust to selection of these hyperparameters, (2) only a few hundred labeled samples may be sufficient to select these hyperparameters, (3) using LEMoN$_{\text{FIX}}$ with the fixed hyperparameter setting described above achieves nearly comparable results, and (4) hyperparameters optimized on a dataset with synthetic noise may transfer well to real datasets.

We repeat each experiment three times, using a different random seed for the noise sampling (for *human* and *real* noise, we use a different random data split). Performance metrics shown are test-set results averaged over these three runs, with error bounds corresponding to one standard deviation.

**Baselines** We compare our method versus previous state-of-the-art in both the classification and captioning settings. We additionally adapt several baselines from the classification setting to the captioning setting. We briefly list the baselines here, and a detailed description is in the Appendix E.

**Classification** In the classification setting, we experiment with the following baselines which require training a classifier on the particular dataset: **AUM** (Pleiss et al., 2020), **Datamap** (Swayamdipta et al., 2020), and **Confident Learning** (Northcutt et al., 2021a), and the following baselines which do not require classifier training: **Deep k-NN** (Bahri et al., 2020), **SimiFeat** (Zhu et al., 2022)-Voting and Ranking, discrepancy in the image space (**Discrepancy**) ($\Upsilon_X^{DIS}$ from Thomas & Kovashka (2022)) **CLIP Similarity** (Kang et al., 2023), and **CLIP Logits** (Liang et al., 2023; Feng et al.).

**Captioning** In the captioning setting, we compare our method with **LLaVA** (Liu et al., 2024) prompting (v1.6-vicuna-13b), and **CapFilt** (Li et al., 2022). We note that the latter can be viewed as an oracle for natural image captioning, as it has been trained in a supervised manner on clean mscoco data. **CLIP Similarity** (Kang et al., 2023), **Discrepancy** (Thomas & Kovashka, 2022), and **Datamap** (Swayamdipta et al., 2020) can also be used directly in this setting. Finally, to adapt classification baselines to captioning, we embed the captions using the corresponding CLIP text encoder, and then use K-means clustering to assign the text caption into one of 100 clusters. We then apply **Deep k-NN** (Bahri et al., 2020) and **Confident Learning** (Northcutt et al., 2021a), using the cluster ID as the discretized class.

## 6 RESULTS

### 6.1 LEMoN OUTPERFORMS BASELINES ON LABEL ERROR DETECTION

**Classification** In Table 2, we show the performance of LEMoN against the baselines for label error detection on four classification datasets. We find that our method outperforms existing baselines which do not require classifier training on all classification datasets. Two downstream-task specific approaches (AUM and Datamap) outperform most training-free models (particularly on cifar10), but LEMoN performs comparably and even outperforms them in two datasets. Similar results are also observed on the two synthetic error types (see Appendix Table I.2). We find that LEMoN$_{\text{FIX}}$ performs almost comparably with LEMoN$_{\text{OPT}}$, and still beats almost all baselines.

Table 3: Label error detection performance on captioning datasets. Bold denotes best (highest) score. A full version of this table with AUPRC can be found in Appendix I.2.

| Method | flickr30k | | mscoco | | mmimdb | | mimiccxr | |
|---|---|---|---|---|---|---|---|---|
| | AUROC | F1 | AUROC | F1 | AUROC | F1 | AUROC | F1 |
| LLaVA | 79.3 (0.8) | 65.0 (1.1) | 80.3 (0.1) | 74.9 (0.3) | 58.4 (0.2) | 58.5 (0.1) | 53.9 (0.5) | 28.7 (0.1) |
| Datamap | 54.0 (1.8) | 28.2 (2.1) | 49.9 (0.7) | 28.6 (0.0) | 50.1 (0.5) | 28.9 (0.3) | 50.2 (0.9) | 28.9 (0.4) |
| Discrepancy | 73.0 (0.6) | 64.7 (1.7) | 72.7 (0.3) | 67.3 (0.9) | 57.4 (0.4) | 40.2 (1.7) | 60.0 (0.8) | 32.8 (2.8) |
| Deep k-NN | 71.1 (0.4) | 64.8 (2.7) | 76.6 (0.4) | 73.2 (0.3) | 58.7 (0.7) | 44.5 (1.0) | 62.9 (0.4) | 46.0 (4.4) |
| Confident | 61.6 (0.5) | 54.3 (0.8) | 66.4 (1.2) | 58.9 (1.5) | 52.8 (0.8) | 53.6 (0.7) | 60.2 (0.3) | **59.4** (0.1) |
| CLIP Sim. | **94.8** (0.5) | **88.1** (0.7) | 93.8 (0.2) | 87.5 (0.3) | 85.1 (0.3) | 74.5 (0.3) | 64.1 (0.4) | 48.6 (3.4) |
| LEMoN$_{\text{FIX}}$ (Ours) | 93.6 (0.2) | - | 92.0 (0.1) | - | 84.3 (0.3) | - | 66.5 (0.2) | - |
| LEMoN$_{\text{OPT}}$ (Ours) | 94.5 (0.2) | 87.7 (0.9) | **95.6** (0.2) | **89.3** (0.2) | **86.0** (0.1) | **76.3** (0.1) | **70.4** (2.3) | 57.0 (1.6) |
| CapFilt (Supervised Training) | 98.6 (0.1) | 94.8 (0.5) | 99.3 (0.0) | 96.2 (0.3) | 82.7 (0.7) | 71.6 (0.8) | 49.2 (0.3) | 28.5 (0.0) |

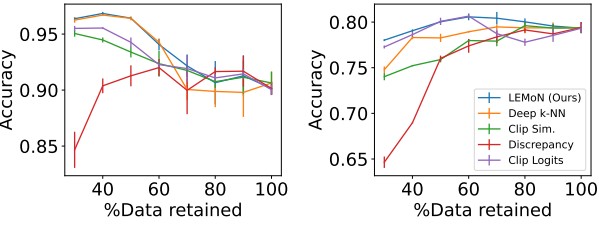

Figure 3: Downstream classification accuracy on cifar10 (left) and cifar100 (right) with LEMoN$_{\text{OPT}}$ with *human* noise versus the baselines. Note that the noise prevalence is 40% in both datasets.

Table 4: Downstream captioning performance when removing 40% samples with highest mislabel scores. We observe that filtering noisy data with LEMoN$_{\text{OPT}}$ improves captioning.

| Dataset | Method | B@4 | CIDER | ROUGE |
|---|---|---|---|---|
| flickr30k | No Filtering | 28.1±1.1 | 64.6 ±2.6 | 49.6±0.7 |
| | CLIP Sim. | 29.7±1.0 | 71.8 ±1.8 | 50.7±0.5 |
| | LEMoN$_{\text{OPT}}$ | 29.6±0.9 | 71.2 ±2.0 | 50.7±0.6 |
| | Clean | 30.8±0.5 | 74.1 ±1.2 | 51.7±0.4 |
| mscoco | No Filtering | 35.1 ±0.4 | 116.7 ±1.5 | 56.4±0.4 |
| | CLIP Sim. | 37.9 ±0.4 | 126.7 ±0.7 | 58.4±0.3 |
| | LEMoN$_{\text{OPT}}$ | 38.4 ±0.2 | 127.3 ±0.2 | 58.5±0.1 |
| | Clean | 38.0 ±0.2 | 126.9±0.5 | 58.4±0.2 |

**Captioning** In Table 3, we find that our method outperforms existing neighborhood and similarity-based baselines on three datasets. In two datasets, our model underperforms an open-sourced fully supervised model (CapFilt), where the training objective included distinguishing between accurate and incorrect captions. Results for synthetic error types show similar trends (see Appendix I.2).

**Label Error Detection Performance Consistent Across Noise Ranges** In Figure I.1, we show the performance of LEMoN versus the CLIP similarity baseline on mscoco and mmimdb, varying the level of the synthetic noise. We find that LEMoN performs better uniformly across noise levels.

**Size of Labeled Validation Set** In Appendix Figure I.3, we examine how varying the size of the labeled validation set impacts the performance of LEMoN$_{\text{OPT}}$. We find that in all four captioning datasets, having about 100-500 labeled examples is sufficient to tune hyperparameters in LEMoN$_{\text{OPT}}$ to outperform LEMoN$_{\text{FIX}}$. In the three datasets where LEMoN$_{\text{FIX}}$ underperforms the CLIP similarity baseline, we find again that having 100-500 labeled validation samples is sufficient for tuning LEMoN$_{\text{OPT}}$ to perform on par with this baseline.

**Robustness to Hyperparameters** Here, we test the robustness of our method when there is no labeled validation set available. First, in Appendix I.4, we visualize the F1 of the selected score when varying $\beta$ and $\gamma$, keeping all other hyperparameters at their selected optimal values. We find that for most datasets and noise types, there is a reasonably large space of such hyperparameters, bounded away from the origin, which achieves close to optimal performance.

Next, we compare the performance of LEMoN$_{\text{OPT}}$ and LEMoN$_{\text{FIX}}$ with hyperparameters described in Section 5.2 across all datasets in Table I.8. We find that when there is no labeled validation set available, using these hyperparameters results in an AUROC drop of only 1.6% on average (std = 1.3%), with a worst-case AUROC drop of 3.9% across all 18 dataset and noise type combinations. Thus, even when a labeled validation set is not available, LEMoN$_{\text{FIX}}$ with reasonable hyperparameter settings is able to outperform most baselines which do use such information.

## 6.2 FILTERING MISLABELED DATA IMPROVES DOWNSTREAM PERFORMANCE

**Classification** To assess the impact of label error detection on the performance of the downstream classification tasks, we filter out samples from the training set with mislabel scores in the top $q$ percentile. We vary $q$, train ViT (Dosovitskiy et al., 2020) models on the filtered dataset, and evaluate the downstream test accuracy using clean data. We compare the performance of LEMoN$_{\text{OPT}}$ with all training-free baselines that produce a continuous score (i.e. all except Simifeat and Confident).

In Figure 3, we find that training with LEMoN$_{\text{OPT}}$ filtered samples leads to the highest accuracy on `cifar10` (96.84%), after removing more than 20% of the data. Training with LEMoN$_{\text{OPT}}$ filtered samples is also on par with baselines on the other datasets (either outperforming or within 0.5% points of best baseline) as shown in Appendix I.14. Further, unlike other baselines, LEMoN is consistently in the top-2 best performing methods across all four datasets. We also show that filtering data in this manner does not reduce classifier robustness (Appendix I.16).

**Captioning**  We finetune a pre-trained Huggingface checkpoint[3] of a transformer decoder conditioned on CLIP image and text tokens – the GenerativeImage2Text (GIT) (Wang et al., 2022a) model – to generate captions. Note that this model is pre-trained on `mscoco`, and evaluated on the Karpathy test split following Wang *et al.* (Wang et al., 2022a). Given the large size of the model, we use the parameter-efficient Low-Rank Adaptation (LoRA) (Hu et al., 2021) for all captioning models. We train models with clean data, noisy captions (*No Filtering*), and by filtering data detected as being mislabeled by a label detection method. In Table 4, we compare results of using either our model or a strong baseline (CLIP Sim.) for filtering data, as measured by the BLEU-4 (Papineni et al., 2002), CIDER (Vedantam et al., 2015), and ROUGE (Lin, 2004) scores. In all cases, we filtered out the top-40% percentile of data predicted to be mislabeled (i.e., equal to the expected prevalence of noisy data). We find that (1) filtering out data predicted to be mislabeled helps recover performance as compared to training on fully clean data along multiple metrics, and (2) our method performs comparably to the baseline in improving downstream results, with some marginal improvements over CLIP Similarity on `mscoco`.

## 6.3 ABLATIONS

In Table I.9, we show the performance of our method after ablating each component. We find that mislabel detection performance almost decreases monotonically as we remove additional components until we reach the CLIP Similarity baseline. We find that ablating the $\tau_1$ and $\tau_2$ terms results in a performance loss of about $1\%$. In Table I.10, we examine the performance of each of the three components of our score and their combinations. We find that $d_{mm}$ is the most critical term. Of the two nearest neighbors terms, we find that $s_n$ (nearest image neighbors) is more important in general, though this is highly dataset dependent, e.g. error detection in `mmimdb` relies much more on neighbors in the text space than the image space, while the opposite is true for `mscoco`.

## 6.4 EXTERNAL PRETRAINING MAY NOT BE REQUIRED

Table 5: Performance of LEMoN for label error detection versus the CLIP similarity baseline on `mimiccxr`, when external pretrained models may not be available. BiomedCLIP (Zhang et al., 2023a) is trained on a large corpus of biomedical image-text pairs. We find that pretraining only on noisy data from MIMIC-CXR outperforms BiomedCLIP, though pretraining on clean `mimiccxr` data (as in CheXzero (Tiu et al., 2022)) does perform better.

| | | Random Noise | | | Cat. Noise | | |
|---|---|---|---|---|---|---|---|
| | | **AUROC** | **AUPRC** | **F1** | **AUROC** | **AUPRC** | **F1** |
| **BiomedCLIP** | Clip Sim. | 66.8 (0.8) | 54.4 (0.9) | 54.3 (1.0) | 64.1 (0.4) | 51.7 (0.5) | 48.6 (3.4) |
| | LEMoN$_{\text{FIX}}$ (Ours) | 69.5 (0.7) | 57.8 (1.0) | - | 66.5 (0.2) | 54.8 (0.4) | - |
| | LEMoN$_{\text{OPT}}$ (Ours) | **73.1** (0.9) | **63.0** (2.0) | **63.1** (3.6) | **70.4** (2.3) | **60.3** (2.3) | **57.0** (1.6) |
| **CLIP Pretrain On** | Clip Sim. | 78.8 (0.1) | 73.4 (0.5) | 70.7 (0.5) | 76.5 (0.5) | 71.2 (0.4) | 67.9 (0.7) |
| **Noisy Data** | LEMoN$_{\text{FIX}}$ (Ours) | **80.5** (0.1) | 76.1 (0.5) | - | 77.0 (0.5) | **72.4** (0.3) | - |
| | LEMoN$_{\text{OPT}}$ (Ours) | **80.5** (0.1) | **76.7** (0.3) | **72.8** (0.7) | **77.2** (0.8) | **72.4** (0.6) | **68.7** (0.2) |
| **CheXzero** | Clip Sim. | 90.8 (0.0) | 89.5 (0.0) | 82.9 (0.2) | 88.4 (0.6) | 86.4 (0.7) | 79.8 (0.7) |
| | LEMoN$_{\text{FIX}}$ (Ours) | 91.4 (0.1) | 90.4 (0.0) | - | 88.4 (0.7) | **87.0** (0.6) | - |
| | LEMoN$_{\text{OPT}}$ (Ours) | **91.6** (0.3) | **90.5** (0.4) | **84.4** (0.5) | **89.0** (0.3) | **87.0** (0.6) | **80.9** (0.6) |

**Medical Images**  Thus far, all of the results for LEMoN (and CLIP Similarity) have utilized CLIP models which have been pretrained on external datasets (e.g. PMC-15M in the case of BiomedCLIP). Here, we examine whether this is necessary, or whether we can achieve comparable performance by pretraining CLIP from scratch *only on the noisy data*. We select `mimiccxr` as it has the most samples out of all captioning datasets. Similar to CheXzero (Tiu et al., 2022), we pretrain a CLIP ViT B/16 from scratch on the `mimiccxr` training set with 40% noise. We train this model for 10 epochs with a batch size of 64, and do not do any model selection or early stopping. We then apply LEMoN and the CLIP similarity baseline using this model, for the same noise level and noise type.

---

[3]`https://huggingface.co/microsoft/git-base`

We present our results in Table 5. Surprisingly, we find that pretraining CLIP only on noisy data from MIMIC-CXR actually outperforms BiomedCLIP. This could be attributed to the pretraining domain (chest X-rays and radiology notes) matching the inference domain exactly (Nguyen et al., 2022). As an upper bound, we evaluate the same methods using CheXzero (Tiu et al., 2022), which has been pretrained on *clean* MIMIC-CXR data. We find that, as expected, it far outperforms this baseline. We conclude that, for large noisy datasets, pretraining a CLIP model from scratch could be a viable solution, though pretraining on clean data from the same domain is certainly superior.

**Web-Scale Corpus**  Motivated by this result, we conduct a large scale experiment on the CC3M dataset (Changpinyo et al., 2021), which contains 2.9 million valid URLs to image-caption pairs. We pretrain CLIP from scratch on this dataset, then use this CLIP model to filter samples in the original dataset using LEMoN$_{FIX}$ and the CLIP similarity baseline. We select the 1 million samples with the lowest mislabel scores from each method, and pretrain another CLIP from scratch on this clean subset. We evaluate the resulting model on zero-shot classification using the VTAB benchmark (Zhai et al., 2019). We find filtering with LEMoN marginally outperforms the baseline on average zero-shot accuracy, though both underperform pretraining on the full corpus. Full details are in Appendix I.9. We additionally conduct an experiment on Datacomp (Gadre et al., 2024) in Appendix I.10

### 6.5 REAL-WORLD ANALYSIS

We conduct a preliminary study of LEMoN on real datasets without known label errors. We run LEMoN$_{FIX}$ and the CLIP similarity baseline on `cifar10`, `cifar100`, `flickr30k`, and `mscoco`. As no labeled validation set is available, we use optimal hyperparameters from models previously run on each dataset with synthetic noise from Section 6.1 (Appendix I.11). For each dataset, we select the top 200 images from the validation and test splits with the highest mislabel scores. We then manually annotated each sample to determine whether it was mislabeled. Crucially, during labeling, images were randomly selected, so the labeler is unaware of whether the candidate image originated from the baseline or our method. We present the accuracy of each method in Table 6. We find that our method outperforms the baseline for every dataset, though we recognize that this is a small-scale study and that many images are ambiguous. Examples of real-world mislabels are also in Figures 1 and I.5. We present a further comparison of our identified error sets in `cifar10` and `cifar100` with a prior work (Northcutt et al., 2021b) which obtained crowd-sourced labels for these datasets in Appendix I.13.

## 7 CONCLUSION

In this work, we proposed LEMoN, a novel method that leverages the neighborhood structure of contrastively pretrained multimodal embeddings to automatically identify label errors in image datasets with natural language text labels.

**Limitations**: Our work has some limitations. For example, we primarily rely on existing open-sourced datasets. While some parts of these datasets may have been used as training data in large pretrained models, we specifically chose pretrained models that take care not to include the test sets of such datasets. Further, we run experiments on a real-world healthcare dataset (`mimiccxr`) to verify our results. Second, in our evaluations, we assume that there exists an oracle binary indicator for whether a sample is mislabeled. As we saw in practice, real-world mislabels contain much more uncertainty and ambiguity, e.g. due to blurry images (Gao et al., 2017; Beyer et al., 2020; Basile et al., 2021; Gordon et al., 2021; 2022). Evaluating the effectiveness of our score as a measure of this uncertainty, in the case of a non-binary target, is an area of future work.

Table 6: We manually label 200 images from real-world datasets that each method identifies as the most likely to be mislabeled and show the percentage (%) of times where it is actually mislabeled. Numbers in parentheses are 95% confidence intervals from a binomial proportion.

|  | CLIP Sim. | Ours |
|---|---|---|
| `cifar10` | 5.5 (3.2) | **10.0** (4.2) |
| `cifar100` | 11.0 (4.3) | **20.5** (5.6) |
| `flickr30k` | 32.5 (6.5) | **41.0** (6.8) |
| `mscoco` | 19.5 (5.5) | **25.5** (6.0) |

Regardless, we believe that our approach is a promising step to automatically detecting and filtering data mislabels at scale. Through experiments on multiple datasets with synthetic and real-world noise, we demonstrated LEMoN's effectiveness in detecting label errors and its ability to improve downstream model performance. Extending such methods to allow for correcting label errors is another promising area of future work.

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

# A  THEORETICAL RESULTS

## A.1  PROOF: THEOREM 4.1

Since $\hat{h}_\theta^{\mathcal{Y}}$ is Lipschitz continuous with constant $L_{\mathcal{Y}}$, for any $y, y' \in \mathcal{Y}$, we have:

$$\left\| \hat{h}_\theta^{\mathcal{Y}}(y') - \hat{h}_\theta^{\mathcal{Y}}(y) \right\|_2 \leq L_{\mathcal{Y}}|y' - y| = L_{\mathcal{Y}}|\eta| \tag{4}$$

Let $d_{mm}(u, v) = ||u - v||_2$ be the Euclidean distance. Note that when $||u||_2 = ||v||_2 = 1$ (as in our experiments), we have that $||u - v||_2 = \sqrt{2(1 - u^T v)} = \sqrt{2d_{cos}(u, v)}$, and so the two distances provide the same ordering of scores. Applying the triangle inequality, we get:

$$d_{mm}\left( \hat{h}_\theta^{\mathcal{X}}(x), \hat{h}_\theta^{\mathcal{Y}}(y') \right) \geq d_{mm}\left( \hat{h}_\theta^{\mathcal{X}}(x), \hat{h}_\theta^{\mathcal{Y}}(y) \right) - \left\| \hat{h}_\theta^{\mathcal{Y}}(y) - \hat{h}_\theta^{\mathcal{Y}}(y') \right\|_2 .$$

When $|\eta| \leq \epsilon$, and substituting from Equation (4), it follows that:

$$d_{mm}\left( \hat{h}_\theta^{\mathcal{X}}(x), \hat{h}_\theta^{\mathcal{Y}}(y') \right) \geq d_{mm}\left( \hat{h}_\theta^{\mathcal{X}}(x), \hat{h}_\theta^{\mathcal{Y}}(y) \right) - L_{\mathcal{Y}}\epsilon$$

Since $\eta \sim \mathcal{N}(0, \sigma^2)$, the probability that $|\eta| \leq \epsilon$ is:

$$P\left(|\eta| \leq \epsilon\right) = 1 - 2\Phi\left(-\frac{\epsilon}{\sigma}\right) = \delta(\epsilon),$$

where $\Phi$ is the cumulative distribution function of the standard normal distribution.

Thus, with probability at least $\delta(\epsilon)$, we have:

$$d_{mm}\left( \hat{h}_\theta^{\mathcal{X}}(x), \hat{h}_\theta^{\mathcal{Y}}(y') \right) \geq d_{mm}\left( \hat{h}_\theta^{\mathcal{X}}(x), \hat{h}_\theta^{\mathcal{Y}}(y) \right) - L_{\mathcal{Y}}\epsilon$$

When $L_{\mathcal{Y}}$ is small, this means that the score for the mislabeled sample cannot be much lower than the score for the positive pair with high probability.

## A.2  PROOF: THEOREM 4.2

Suppose that $\zeta_Y$ is distributed such that $\text{supp}(k\zeta_Y(1-p)) \subseteq \{0, 1, ..., k\}$. For a correctly labeled sample $(X, Y)$, we have that $k\zeta_Y(1-p)$ of the neighbors are relevant and have correct labels, and so each contribute $d_{\mathcal{X}}(X, \bar{X})$ to $S_m(X, Y)$, and all remaining samples are either incorrectly labeled, or are not relevant to $Y$, and so each contribute $d_{\mathcal{X}}(X, X')$. Since $S_m(X, Y)$ is the sum of iid Gaussians, it is also a Gaussian, with:

$$\mathbb{E}[S_m(X, Y)] = \frac{1}{k}\left( \mathbb{E}[\mathbb{E}[d(X, \bar{X}_1) + ... + d(X, \bar{X}_{k\zeta_Y(1-p)})|\zeta]] + \mathbb{E}[\mathbb{E}[d(X, X_1') + ... + d(X, X_{k-k\zeta_Y(1-p)}')|\zeta]] \right)$$

$$= \mathbb{E}[\zeta_Y](1-p)\mu_2 + (1 - \mathbb{E}[\zeta_Y](1-p))\mu_1$$

$$= \mathbb{E}[\zeta_Y](1-p)(\mu_2 - \mu_1) + \mu_1$$

$$\text{Var}[S_m(X, Y)] = \mathbb{E}[\text{Var}(S_m(X, Y)|\zeta_Y)] + \text{Var}(\mathbb{E}[S_m(X, Y)|\zeta_Y])$$

$$= \mathbb{E}[\frac{1}{k^2}\text{Var}\left( d(X, \bar{X}_1) + ... + d(X, \bar{X}_{k\zeta_Y(1-p)}) + d(X, X_1') + ... + d(X, X_{k-k\zeta_Y(1-p)}')|\zeta_Y \right)]$$

$$\quad + \text{Var}(\mathbb{E}[S_m(X, Y)|\zeta_Y])$$

$$= \mathbb{E}[\frac{1}{k}\left( \zeta_Y(1-p)\sigma_2^2 + (1 - \zeta_Y(1-p))\sigma_1^2 \right)] + \text{Var}(\zeta_Y(1-p)(\mu_2 - \mu_1) + \mu_1)$$

$$= \frac{1}{k}\left( \mathbb{E}[\zeta_Y](1-p)\sigma_2^2 + (1 - \mathbb{E}[\zeta_Y](1-p))\sigma_1^2 \right) + \text{Var}(\zeta_Y)(1-p)^2(\mu_2 - \mu_1)^2$$

Similarly,

$$S(X', Y') \sim \mathcal{N}(\mu_1, \frac{\sigma_1^2}{k})$$

Putting it all together:

$$\mathbb{P}(S_m(X', Y') - S_m(X, Y) > 0) = 1 - \Phi(\frac{-\mu}{\sigma})$$

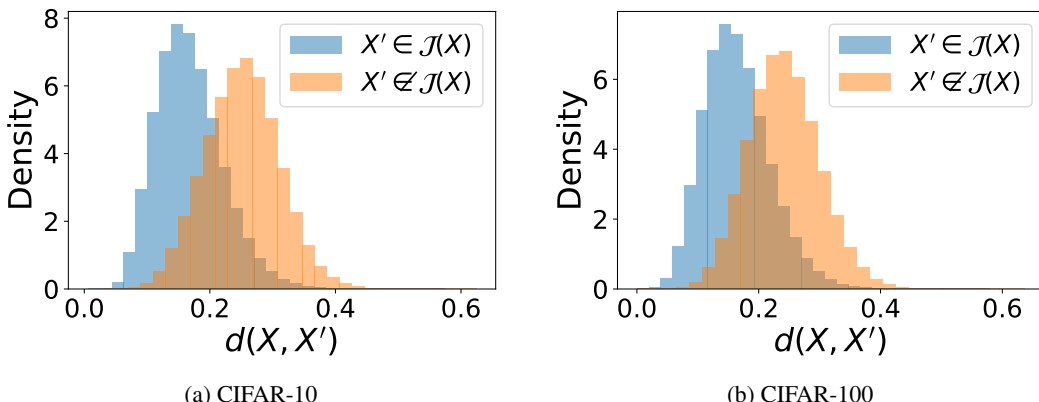

(a) CIFAR-10             (b) CIFAR-100

Figure A.1: Histogram of cosine distances in the CLIP image embedding space

Where $\mu = \mathbb{E}[\zeta_Y](1-p)(\mu_1 - \mu_2), \sigma = \sqrt{\frac{1}{k}\left(\mathbb{E}[\zeta_Y](1-p)\sigma_2^2 + (2 - \mathbb{E}[\zeta_Y](1-p))\sigma_1^2\right) + \mathrm{Var}(\zeta_Y)(1-p)^2(\mu_2 - \mu_1)^2}$,
and $\Phi$ is the Gaussian CDF. Note that $\mathrm{Var}(\zeta_Y)$ is finite as $\zeta_Y$ is bounded by $[0, 1]$.
Setting $\mu > 0$ gives Lemma 4.3.

### A.3 EMPIRICALLY VALIDATING ASSUMPTION 2

To empirically validate Assumption 2, we utilize the training sets from the original CIFAR-10 and CIFAR-100 datasets. As these are classification datasets, we naturally define $\mathcal{J}$ as: $x_2 \in \mathcal{J}(x_1) \iff y_1 = y_2$, i.e. all images with the same label are paraphrases. We encode these images using the image encoder from OpenAI CLIP ViT-B/32 (Radford et al., 2021), and utilize the cosine distance as $d_{\mathcal{X}}$. We compute pairwise distance between all 40,000 samples, and categorize these distances into either $x' \in J(x)$ or $x' \notin J(x)$. We plot a histogram of these distances in Figure A.1. Visually, both of these distributions appear to be normal, and we also observe that $\mu_1 > \mu_2$ from Lemma 4.3. We then run a Shapiro–Wilk test on all four distributions to test for normality, randomly subsampling to 100 samples, as the Shapiro-Wilk test is not suitable for large sample sizes (Ghasemi & Zahediasl, 2012). We find that in all four cases, the null hypothesis cannot be rejected ($p > 0.05$), and the test statistics are all greater than 0.97, indicating a high degree of normality.

## B COMPARISON WITH THOMAS & KOVASHKA (2022)

The goal of Thomas & Kovashka (2022) to identify samples with semantic diversity, which is different from our goal of identifying mislabeled examples. As such, their proposed scores (i.e. $\Upsilon^{DIS}$ and $\Upsilon^{DIV}$) may not be effective in identifying mislabeled samples. As an example, consider the score $\Upsilon_Y^{DIS}$, which computes the similarity between the original caption, and the captions of its second-degree neighbors in text-space. Given a particular caption, e.g. "This is a plane from the front view" in Figure 2, it could have second-degree neighbors in text-space that are semantically very similar to this caption (e.g. "A plane facing the viewer"). However, only computing the distance of these captions in text space does not provide any signal for whether the *image* is correctly paired to the caption. Similarly, the $\Upsilon^{DIV}$ scores also would not necessarily work, as the closeness of neighbors to each other in either modality do not provide a signal for whether the original sample is mislabeled.

However, the score from Thomas & Kovashka (2022) that would intuitively provide a signal for mislabeling is $\Upsilon_X^{DIS}$, which computes second-degree neighbors in text space, then examines similarity between images. This is essentially the sum over $d_{\mathcal{X}}(\mathbf{x}, \mathbf{x}_{m_j})$ terms in our Equation (3), but using second-degree neighbors instead of nearest neighbors. In addition, our Equation (3) contains two additional weighting terms (which we show improve label error performance in our ablation experiments). Finally, our proposed score contains the sum of two additional terms, which are not explored in Thomas & Kovashka (2022).

We compare the performance of our method against the $\Upsilon_X^{DIS}$ score in the main paper, and show performance of all four scores from Thomas & Kovashka (2022) in Appendix I.8.

## C   LEMON Algorithm

---

**Algorithm 1:** LEMON: **L**abel **E**rror Detection Using **M**ulti**m**odal **N**eighbors

---

**Input:** Dataset $\mathcal{D} = \{(\mathbf{x}_i, \mathbf{y}_i)\}_{i=1}^{N}$, Multimodal encoders $h_\theta^{\mathcal{X}}, h_\theta^{\mathcal{Y}}$, Distance functions $d_{\mathcal{X}}, d_{\mathcal{Y}}$
Hyperparameters: $k, \beta, \gamma, \tau_{1,n}, \tau_{2,n}, \tau_{1,m}, \tau_{2,m}$

**Output:** Scores $\{s_i\}_{i=1}^{N}$

1 Cache embeddings $h_\theta^{\mathcal{X}}(\mathbf{x}_i)$ and $h_\theta^{\mathcal{Y}}(\mathbf{y}_i)$ for $(\mathbf{x}_i, \mathbf{y}_i) \in \mathcal{D}$ ;

2 Cache $d_{mm}(\mathbf{x}_i, \mathbf{y}_i) = 1 - \frac{h_\theta^{\mathcal{X}}(\mathbf{x}_i) \cdot h_\theta^{\mathcal{Y}}(\mathbf{y}_i)}{\|h_\theta^{\mathcal{X}}(\mathbf{x}_i)\|_2 \|h_\theta^{\mathcal{Y}}(\mathbf{y}_i)\|_2}$ for $(\mathbf{x}_i, \mathbf{y}_i) \in \mathcal{D}$ ;

3 **for** $i = 1$ **to** $N$ **do**

4      Find indices $\{n_j\}_{j=1}^{k}$ of $k$ nearest neighbors of $\mathbf{x}_i$ from $\mathcal{D} \setminus \{(\mathbf{x}_i, \mathbf{y}_i)\}$ using $d_{\mathcal{X}}$ ;     `//  d`$_{\mathcal{X}}$
      `can use cached` $h_\theta^{\mathcal{X}}$

5      Find indices $\{m_j\}_{j=1}^{k}$ of $k$ nearest neighbors of $\mathbf{y}_i$ from $\mathcal{D} \setminus \{(\mathbf{x}_i, \mathbf{y}_i)\}$ using $d_{\mathcal{Y}}$ ;     `//  d`$_{\mathcal{Y}}$
      `can use cached` $h_\theta^{\mathcal{Y}}$

6      Compute $s_{n,i} := \frac{1}{k} \sum_{j=1}^{k} d_{\mathcal{Y}}(\mathbf{y}_i, \mathbf{y}_{n_j}) e^{-\tau_{1,n} d_{\mathcal{X}}(\mathbf{x}_i, \mathbf{x}_{n_j})} e^{-\tau_{2,n} d_{mm}(\mathbf{x}_{n_j}, \mathbf{y}_{n_j})}$;

7      Compute $s_{m,i} := \frac{1}{k} \sum_{j=1}^{k} d_{\mathcal{X}}(\mathbf{x}_i, \mathbf{x}_{m_j}) e^{-\tau_{1,m} d_{\mathcal{Y}}(\mathbf{y}_i, \mathbf{y}_{m_j})} e^{-\tau_{2,m} d_{mm}(\mathbf{x}_{m_j}, \mathbf{y}_{m_j})}$;

8      $s_i := d_{mm}(\mathbf{x}_i, \mathbf{y}_i) + \beta s_{n,i} + \gamma s_{m,i}$

9 **return s**;

---

For each image-caption pair in the dataset, we first compute how similar the image and caption are to each other using a pre-trained CLIP model ($d_{mm}$), which gives a basic measure of how well they match. To compute $s_m$, we compute the nearest neighbors of the caption among other captions in the dataset. For each neighbor, we look at how similar their corresponding image is to the original image. The intuition is that if a sample is correctly labeled, the image should be similar to images of other samples with similar captions. We weight each neighbor based on how close it is to our original sample and how well-matched the neighboring pairs themselves are. Finally, we repeat this for nearest neighbors in the image space to get $s_n$. LEMON is then the weighted sum of these three scores.

Table C.1: Notation and definitions used in Section 3.

| Symbol/Notation | Meaning |
|---|---|
| $\mathcal{D}$ | Dataset consisting of samples $(\mathbf{x}, \mathbf{y})_{i=1}^{N}$ |
| $\mathbf{x}, \mathcal{X}$ | First modality and its corresponding space (e.g., images) |
| $\mathbf{y}, \mathcal{Y}$ | Second modality and its corresponding space (e.g., text) |
| $f^*$ | Oracle function that assigns a binary mislabel indicator $z_i$ |
| $z_i$ | Mislabel indicator for sample $i$ ($z_i = 1$ if mislabeled, $z_i = 0$ otherwise) |
| $f(\mathbf{x}, \mathbf{y}) = s$ | Model output score |
| $d_{\mathcal{X}}, d_{\mathcal{Y}}$ | Distance functions in $\mathcal{X}$ and $\mathcal{Y}$ spaces |
| $B(\mathbf{x}, r)$ | Ball of radius $r$ centered at $\mathbf{x}$ in $\mathcal{X}$ space |
| $B(\mathbf{y}, r)$ | Ball of radius $r$ centered at $\mathbf{y}$ in $\mathcal{Y}$ space |
| $r_k(\mathbf{x})$ | Radius such that the ball $B(\mathbf{x}, r)$ contains at least $k$ neighbors |
| $\mathbf{x}_{n_j}$ | Nearest neighbor $j$ in $\mathcal{X}$ space |
| $\mathbf{y}_{m_j}$ | Nearest neighbor $j$ in $\mathcal{Y}$ space |
| $h_\theta = (h_\theta^{\mathcal{X}}, h_\theta^{\mathcal{Y}})$ | Multimodal encoder mapping $\mathcal{X}$ and $\mathcal{Y}$ to $\mathbb{R}^d$ |
| $d_{mm}(\mathbf{x}, \mathbf{y})$ | Multimodal distance between $\mathbf{x}$ and $\mathbf{y}$ |
| $s_n(\mathbf{x}, \mathbf{y}, \mathcal{D})$ | Score component based on $\mathbf{x}$'s neighbors, see Equation (2). |
| $s_m(\mathbf{x}, \mathbf{y}, \mathcal{D})$ | Score component based on $\mathbf{y}$'s neighbors, see Equation (3). |
| $\beta, \gamma$ | Hyperparameters weighting $s_n$ and $s_m$ |
| $\tau_{1,n}, \tau_{2,n}, \tau_{1,m}, \tau_{2,m}$ | Hyperparameters for weighting terms in $s_n$ and $s_m$ |
| $k$ | Number of nearest neighbors |

# D  DATA PROCESSING

## D.1  CLASSIFICATION

We utilize CIFAR10N (`cifar10`) and CIFAR100N (`cifar100`) object detection (Zhu et al., 2022) datasets for all classification-based experiments. Each image is associated with a label indicating the primary object present in the image. These datasets contain 50,000 image-label pairs, with a clean and noisy label available per image. The noisy labels are examples of real human errors within the dataset. Further, we also generate synthetically noised labels as described in the main text. All images are resized to 224x224, center cropped, and normalized using mean and standard deviations corresponding to CLIP during the pre-processing stage. These two datasets are released under the Creative Commons Attribution-NonCommercial 4.0 license.

For `miniImageNet` and `stanfordCars`, we use the "red" datasets from Jiang et al. (2020), which contain noise from real-world web annotators. We split the full dataset (containing all annotations) into 75%/12.5%/12.5% train/val/test sets, stratifying by the mislabel flag. The annotations are licensed by Google under CC BY 4.0 license, and the images are under CC BY 2.0 license.

To generate the "text" modality for these classification datasets, we utilize the label name correspond to each class. For example, class 0 in `cifar10` is "airplane", and this is the caption associated we associate with all images of that class. In contrast to the caption-based datasets, there will be multiple k-nearest neighbors in the text modality with zero distance (i.e., with the same class label).

## D.2  CAPTIONING

We preprocess MSCOCO (Lin et al., 2014) and Flickr30k (Young et al., 2014) by using the Karpathy split (Karpathy & Fei-Fei, 2015), and then selecting one random annotation from the ones available. For the MMIMDB dataset (Arevalo et al., 2017), we utilize the plot outline as the text, and use the dataset splits provided. For MIMIC-CXR (Johnson et al., 2019), we use all images in the database and the provided data splits, and extract the findings and impression sections from the radiology note for the text modality. Images were normalized and transformed using the same procedure described above.

For downstream captioning, we use the pre-trained tokenizer and image processor corresponding to the pre-trained model (GIT (Wang et al., 2022a)) to pre-process image and captions.

Note that `flickr30k` is available under Flickr terms of use for non-commercial research and/or educational purposes[4]. `mscoco` is available under Creative Commons Attribution 4.0 License. `mmimdb` is available for personal and non-commercial use[5]. Finally, `mimiccxr` is available under the PhysioNet Credentialed Health Data License 1.5.0[6].

# E  BASELINE METHODS

## E.1  CLASSIFICATION

### TRAINING-DEPENDENT

**AUM (Pleiss et al., 2020)**: This model assumes access to a classifier that can predict the class that an image likely belongs to. Then, the margin of difference between the prediction probability from the trained classifier for the assigned class and the class with the (next) highest probability is computed and averaged over training epochs. This score is thresholded to identify potential label errors.

**Datamap (Swayamdipta et al., 2020)**: Similar to AUM, this method requires access to a pretrained classifier. In this baseline, it is assumed that instances with label errors are 'hard to learn', and thus low confidence in prediction throughout training epochs. To produce a single score, we combine the mean and standard deviation of the probability associated with the assigned class into a single score[7].

**Confident Learning** (Northcutt et al., 2021a) is designed to identify labeling errors in classification datasets by modeling the relationship between true class labels and noisy ones. It sets thresholds for

---

[4]`https://shannon.cs.illinois.edu/DenotationGraph/`
[5]`https://developer.imdb.com/non-commercial-datasets/`
[6]`https://physionet.org/content/mimic-cxr/view-license/2.0.0/`
[7]We experimented with different strategies, and the square root of the product of the mean and (1-standard deviation) and (1-mean) and standard deviation led to comparable, high validation F1 scores.

each true-noisy label pair. Using these thresholds, the model employs predicted class probabilities to rank predictions for each class, filtering out the noisy data.

### Training-free

**CLIP Logits (Liang et al., 2023)**: CLIP is used as a zero-shot classifier to obtain the softmax-based probability for the assigned class. This value is then thresholded to identify label errors. Recently, (Feng et al.) used a similar zero-shot prediction jointly with a semi-supervised training approach for learning in the presence of label noise.

**CLIP Similarity (Kang et al., 2023)**: The distance (either euclidean or cosine) between image and text embeddings from CLIP are computed and thresholded.

**Deep $k$-NN**(Bahri et al., 2020) The proportion of $k$ nearest neighbors[8] with the same label is computed for each image of interest. Prior works have utilized different representations for obtaining neighbors, including logits and representations from pre-trained (Zhu et al., 2022) vision models. We find that pre-trained representations from CLIP outperformed logits from a zero-shot CLIP classifier (Zhu et al., 2022).

**SimiFeat** (Zhu et al., 2022) uses nearby features to detect noisy labels under the assumption that local groups of features share clean or noisy labels. **SimiFeat-V** (Zhu et al., 2022) uses local voting and **SimiFeat-R** leverages ranking to detect noisy labels based on HOC estimator. The binary outputs produced are used for all score computations. Note that the difference between Simifeat-V and deep k-NN is in the data processing and augmentation.

**Discrepancy** (Thomas & Kovashka, 2022) finds second-degree nearest neighbors in the text space, then computes the average distance of these neighbors to the original sample in image space. We utilize the same CLIP model to compute semantic distance here as in LEMoN.

### E.2   Captioning

#### Pre-trained or Supervised

**LLaVA** (Liu et al., 2024): We prompt LLaVA (v1.6-vicuna-13b) with the following prompt: `The proposed caption for this image is "{}". Is this caption correct? Only answer with "Yes" or "No".` We examine the probability distribution over the first non-special token, and find the likelihood of the token with the highest probability. If the corresponding token in lower case starts with "yes", we return $1-$ this probability as the mislabel score. Otherwise, we return the probability.

**CapFilt (oracle-like)**: We generate predictions using pre-trained model trained on distinguishing between high-quality MSCOCO and noisy synthetic captions (Li et al., 2022). This forms an oracle-like, fully supervised baseline.

#### Unsupervised

**Datamap**: We compute the cross-entropy across training epochs and compute the ratio of the mean and variance in loss across epochs. That is, we expect captioning loss for instances with label errors to be consistently high. We train captioning models for 3 epochs, with LoRA rank set to 4, and a maximum length of $100$[9] for the finetuning task.

**Confident Learning**: We adapt this approach for dual-modality datasets, such as image-text pairs, by clustering text embeddings to serve as class labels for noise detection.

#### Downstream-task Unaware

**Deep KNN**: We cluster captions similar to confident learning, adapting classification baseline.

**CLIP Similarity**: This is the same setup as classification.

**Discrepancy**: This is the same setup as classification.

---

[8]Note that this score is not continuous.

[9]This is longer than captions in the train sets of all datasets except the medical dataset, and we verified that higher maximum length does not change results.

## F    COMPUTE SETUP

We run our experiments on a shared Slurm cluster. Each experiment used one RTX A6000 with 48 GB VRAM, 10 CPU cores of Intel Xeon Ice Lake Platinum 8368, and 50 GB RAM.

## G    HYPERPARAMETERS IN LABEL ERROR DETECTION

The hyperparameters in each case were selected based on the validation set F1-score. Note that LEMON$_{\text{FIX}}$ does not require hyperparameter tuning. Baseline code is included in the supplementary material. For SimiFeat-V and -R, we use the official open-sourced implementation directly.

### G.1    CLASSIFICATION

The search space for each method:

1. AUM, Datamap: learning rate $\in \{5e-5, 5e-6\}$, training for epochs $\in \{5, 10\}$[10]
2. Confident learning: learning rate $\in \{5e-7, 5e-6, 5e-5\}$, upto 30 epochs with early stopping with a patience of 10.
3. CLIP Sim.: cosine distance metric, no other hyperparameters
4. CLIP Zero shot: distance metric
5. Discrepancy: $k \in \{1, 2, 5, 10, 15, 20, 30, 50\}$
6. deep k-NN: $k$, cosine distance metric
7. Simifeat: we set $k = 10$ following the original paper (Zhu et al., 2022).

### G.2    CAPTIONING

For most baselines requiring a class index–obtained by clustering captions–we set the number of clusters to be 100.

1. LLaVA: Small amount of prompt tuning. The optimal prompt selected was `The proposed caption for this image is "{}". Is this caption correct? Only answer with "Yes" or "No".'`
2. Confident learning: learning rate $= 5e-6$, upto 30 epochs with early stopping with a patience of 10, number of clusters for captions
3. Discrepancy: $k \in \{1, 2, 5, 10, 15, 20, 30, 50\}$
4. deep k-NN: representation type, $k \in \{1, 2, 5, 10, 15, 20, 30, 50\}$, distance metric (either cosine or euclidean)

### G.3    OUR METHOD

We search the following hyperparameters for our LEMON$_{\text{OPT}}$:

1. $k \in \{1, 2, 5, 10, 15, 20, 30, 50\}$
2. Distance metric (either cosine or euclidean)
3. $\beta, \gamma, \tau_{1,n}, \tau_{2,n}, \tau_{1,m}, \tau_{2,m}$: We take the hyperparameter set which achieves the best validation set F1 from these two strategies: (1) Using Scipy's `minimize` function, with initial guess $(1, 1, ..., 1)$, and with no explicit bounds. (2) Using a grid search with the following grid:
   - $\beta \in \{0, 5, 10, 15, ..., 100\}$
   - $\gamma \in \{0, 5, 10, 15, ..., 100\}$
   - $\tau_{1,n}, \tau_{2,n}, \tau_{1,m}, \tau_{2,m} \in \{0, 1, 5, 10\}$

### G.4    OPTIMAL HYPERPARAMETERS

Optimal hyperparameters for classification datasets can be found in Table G.1, and optimal hyperparameters for captioning datasets can be found in Table G.2.

---

[10]Note that we experiment with training for fewer epochs to avoid memorization, following (Pleiss et al., 2020).

Table G.1: Optimal hyperparameters for methods shown in Table 2. Note that Simifeat, CLIP Sim., and LEMON$_{\text{FIX}}$ have no tunable hyperparameters.

| | cifar10 | cifar100 | miniImageNet | stanfordCars |
|---|---|---|---|---|
| **AUM** | LR = 5E-6
Epochs = 5 | LR = 5E-5
Epochs = 5 | LR = 5E-6
Epochs = 10 | LR = 5E-5
Epochs = 10 |
| **Datamap** | LR = 5E-6
Epochs = 5 | LR = 5E-5
Epochs = 5 | LR = 5E-5
Epochs = 5 | LR = 5E-5
Epochs = 10 |
| **Confident** | LR=5e-06
Epochs=30
Batch size=128 | LR=5e-06
Epochs=30
Batch size=128 | LR=5e-06
Epochs=30
Batch size=128 | LR=5e-06
Epochs=30
Batch size=128 |
| **CLIP Logits** | Cosine distance | Cosine distance | Cosine distance | Cosine distance |
| **Discrepancy** | k=20 | k=50 | k=30 | k=20 |
| **Deep k-NN** | k=50
cosine distance | k=20
cosine distance | k=50
cosine distance | k=50
cosine distance |
| **LEMON$_{\text{OPT}}$** | k=50
cosine distance
$\beta = 20$
$\gamma = 35$
$\tau_{1,n} = 0$
$\tau_{2,n} = 5$
$\tau_{1,m} = 0$
$\tau_{2,m} = 5$ | k=20
cosine distance
$\beta = 2.14$
$\gamma = -0.024$
$\tau_{1,n} = -1.71$
$\tau_{2,n} = 4.85$
$\tau_{1,m} = -0.068$
$\tau_{2,m} = -0.019$ | k=50
Euclidean distance
$\beta = 0.664$
$\gamma = 0.395$
$\tau_{1,n} = 1.91$
$\tau_{2,n} = 1.04$
$\tau_{1,m} = 1.00$
$\tau_{2,m} = 1.35$ | k=15
Euclidean distance
$\beta = 0.631$
$\gamma = 0.431$
$\tau_{1,n} = 0.898$
$\tau_{2,n} = -0.192$
$\tau_{1,m} = 0.0$
$\tau_{2,m} = -0.001$ |

Table G.2: Optimal hyperparameters for methods shown in Table 3. Note that LLaVA, CLIP Sim. and LEMON$_{\text{FIX}}$ have no tunable hyperparameters.

| | flickr30k | mscoco | mmimdb | mimiccxr |
|---|---|---|---|---|
| **Datamap** | Batch size = 16
Epochs = 3
LoRA rank = 4 | Batch size = 16
Epochs = 3
LoRA rank = 4 | Batch size = 16
Epochs = 3
LoRA rank = 4 | Batch size = 16
Epochs = 3
LoRA rank = 4 |
| **Discrepancy** | k=5 | k=10 | k=10 | k=10 |
| **Deep k-NN** | k=50
cosine distance | k=50
cosine distance | k=20
cosine distance | k=50
cosine distance |
| **Confident** | LR=5e-06
Epochs=30
Batch size=128
n_cluster=10 | LR=5e-06
Epochs=30
Batch size=128,
n_cluster=10 | LR=5e-06
Epochs=30
Batch size=128
n_cluster=10 | LR=5e-06
Epochs=30
Batch size=16
n_cluster=10 |
| **LEMON$_{\text{OPT}}$** | k=30
cosine distance
$\beta = 0.092$
$\gamma = 0.177$
$\tau_{1,n} = 0.274$
$\tau_{2,n} = 0.074$
$\tau_{1,m} = 0.072$
$\tau_{2,m} = 0.0$ | k=30
cosine distance
$\beta = 5.324$
$\gamma = 11.057$
$\tau_{1,n} = 5.143$
$\tau_{2,n} = 10.498$
$\tau_{1,m} = 7.233$
$\tau_{2,m} = 15.637$ | k=10
Euclidean distance
$\beta = 1.001$
$\gamma = 1.202$
$\tau_{1,n} = 0.983$
$\tau_{2,n} = 1.000$
$\tau_{1,m} = 4.450$
$\tau_{2,m} = 1.080$ | k=30
cosine distance
$\beta = 5$
$\gamma = 10$
$\tau_{1,n} = 5$
$\tau_{2,n} = 10$
$\tau_{1,m} = 5$
$\tau_{2,m} = 10$ |

## H   Hyperparameters in Downstream Models

### H.1   Classification

We train a Vision Transformer (ViT)-based image classification  (Dosovitskiy et al., 2020)[11] model pre-trained on ImageNet-21k (Ridnik et al., 2021) and fine-tuned on ImageNet 2012 (Russakovsky et al., 2015) with an additional linear layer.  We add a linear layer above the classification logits, with an initial learning rate of 0.01, and learning rate scheduling for 10 epochs, and early stopping with a patience of 3. For `miniImageNet`, we use linear probing with just a layer added on top of the standard ViT classification logits (since the pre-trained task matches the downstream task to an extent).

### H.2   Captioning

The hyperparameter tuning grid for the captioning model[12] are: learning rate in $\{1e-5, 1e-4\}$, batch size: 16, maximum number of epochs: 10. The model checkpoint from the epoch with lowest validation loss is used for caption generation at test time. For LoRA, we use a rank in $\{4,16\}$. For text generation, we use beam search with 4 beams, following  (Wang et al., 2022a). We use the AdamW optimizer (Loshchilov & Hutter, 2018), with cosine scheduling for learning rate with 1000 warmup steps.

## I   Additional Experimental Results

### I.1   Label Error Detection in Classification Settings

Full results on classification datasets using the noise types bolded in Table 1 (including AUPRC) can be found in Table I.1.

The performance of all baselines and our method on the two types of synthetic errors are shown in Table I.2, all at a noise level of 40% (comparable to the amount of error in the noisy CIFAR datasets).

### I.2   Label Error Detection in Captioning Settings

Full results on classification datasets using the noise types bolded in Table 1 (including AUPRC) can be found in Table I.3.

Results on the remaining synthetic noise types (at 40%) can be found in: `flickr30k` I.4, `mscoco` I.5, `mmimdb` I.6, and `mimic-cxr` I.7.  Across all datasets and noising types, we find that our model outperforms other non-oracle/supervised baselines.

### I.3   Varying Noise Level

We show the AUROC for varying noise levels in Figure I.1.

### I.4   Robustness to Hyperparameters

We show the test-set F1 of LEMoN for varying $\beta$ and $\gamma$, keeping all other hyperparameters at their fixed optimal values, in Figure I.2. In Table I.8, we show the performance of LEMoN when hyperparameters are fixed (at $k = 30$, cosine distance, $\beta = \gamma = 5$, $\tau_{1,n} = \tau_{1,m} = 0.1$, and $\tau_{2,n} = \tau_{2,m} = 5$) versus when they are optimized using a labeled validation set. Note that F1 is not computed as it requires external information to select a threshold.

### I.5   Ablations of our Method

Ablations of our method can be found in Table I.9 and Table I.10.

### I.6   Runtime Comparison

We compare the runtime of LEMoN with baselines in Table I.11.

---

[11]`https://huggingface.co/google/vit-base-patch16-224`
[12]`https://huggingface.co/microsoft/git-base`

Table I.1: Label error detection performance on classification datasets.

| Dataset | Method | Training-free | AUROC (%) | AUPRC (%) | F1 (%) |
|---|---|---|---|---|---|
| cifar10 | AUM | ✗ | **98.3** (0.1) | **97.9** (0.1) | **94.0** (0.1) |
| | Datamap | | 98.2 (0.1) | 97.6 (0.1) | 93.4 (0.5) |
| | Confident | | 93.7 (0.4) | 89.4 (0.6) | 92.7 (0.5) |
| | CLIP Logits | ✓ | 95.5 (0.2) | 93.9 (0.3) | 88.0 (0.5) |
| | CLIP Sim. | | 93.8 (0.1) | 92.4 (0.2) | 86.9 (0.4) |
| | Simifeat-V | | 90.6 (0.3) | 87.9 (0.7) | 88.0 (0.4) |
| | Simifeat-R | | 90.7 (0.3) | 88.0 (0.4) | 88.1 (0.5) |
| | Discrepancy | | 77.1 (1.9) | 70.4 (2.7) | 68.2 (1.9) |
| | Deep k-NN | | 97.8 (0.1) | 96.5 (0.2) | 92.5 (0.5) |
| | LEMoN$_{\text{FIX}}$ (Ours) | | 97.7 (0.1) | 96.8 (0.3) | - |
| | LEMoN$_{\text{OPT}}$ (Ours) | | 98.1 (0.0) | 97.4 (0.1) | 93.1 (0.2) |
| cifar100 | AUM | ✗ | **92.2** (0.2) | **90.0** (0.4) | **83.8** (0.4) |
| | Datamap | | 91.8 (0.2) | 89.4 (0.3) | 83.5 (0.6) |
| | Confident | | 74.1 (1.7) | 59.3 (2.2) | 69.3 (2.0) |
| | CLIP Logits | ✓ | 84.9 (0.7) | 80.3 (1.2) | 75.5 (0.5) |
| | CLIP Sim. | | 78.5 (0.6) | 72.1 (0.7) | 69.2 (1.3) |
| | Simifeat-V | | 79.5 (0.0) | 71.1 (0.8) | 73.1 (0.5) |
| | Simifeat-R | | 79.7 (0.2) | 71.1 (0.8) | 73.6 (0.6) |
| | Discrepancy | | 66.0 (1.5) | 57.4 (2.3) | 51.9 (1.8) |
| | Deep k-NN | | 87.4 (0.3) | 77.9 (1.0) | 78.0 (0.3) |
| | LEMoN$_{\text{FIX}}$ (Ours) | | 88.9 (0.7) | 84.6 (1.1) | - |
| | LEMoN$_{\text{OPT}}$ (Ours) | | 90.8 (0.0) | 87.4 (0.3) | 81.3 (0.2) |
| miniImageNet | AUM | ✗ | 83.1 (0.2) | **73.2** (0.5) | 75.3 (0.2) |
| | Datamap | | **85.0** (0.2) | 71.9 (0.7) | **77.0** (0.2) |
| | Confident | | 70.5 (0.2) | 52.8 (0.3) | 54.7 (0.4) |
| | CLIP Logits | ✓ | 90.0 (0.2) | 80.9 (0.5) | **82.5** (0.2) |
| | CLIP Sim. | | 89.3 (0.2) | 80.8 (0.3) | 81.3 (0.5) |
| | Simifeat-V | | 68.2 (0.3) | 53.0 (0.4) | 55.0 (0.5) |
| | Simifeat-R | | 68.0 (0.3) | 52.8 (0.3) | 54.7 (0.4) |
| | Discrepancy | | 79.4 (0.3) | 65.6 (0.7) | 69.8 (0.4) |
| | Deep k-NN | | 83.2 (0.2) | 70.9 (0.6) | 75.2 (0.4) |
| | LEMoN$_{\text{FIX}}$ (Ours) | | 89.5 (0.2) | **81.5** (0.3) | - |
| | LEMoN$_{\text{OPT}}$ (Ours) | | **90.2** (0.2) | 81.4 (1.3) | 82.3 (0.1) |
| stanfordCars | AUM | ✗ | 70.5 (2.4) | **42.8** (1.6) | 62.3 (1.2) |
| | Datamap | | **72.3** (1.8) | 39.8 (0.5) | **64.9** (2.1) |
| | Confident | | 61.0 (0.5) | 33.2 (1.7) | 43.4 (1.6) |
| | CLIP Logits | ✓ | 68.8 (0.7) | 39.7 (0.9) | 64.9 (0.4) |
| | CLIP Sim. | | 69.8 (0.5) | 40.7 (1.0) | 61.7 (0.8) |
| | Simifeat-V | | 63.7 (1.2) | 33.7 (1.2) | 43.7 (1.5) |
| | Simifeat-R | | 63.5 (1.3) | 33.2 (1.7) | 43.4 (1.6) |
| | Discrepancy | | 65.7 (0.7) | 33.1 (0.6) | 59.9 (0.4) |
| | Deep k-NN | | 71.4 (0.6) | 42.7 (0.5) | 65.3 (0.9) |
| | LEMoN$_{\text{FIX}}$ (Ours) | | 72.6 (0.7) | **44.9** (1.4) | - |
| | LEMoN$_{\text{OPT}}$ (Ours) | | **73.1** (0.5) | 40.5 (0.5) | **67.3** (1.0) |

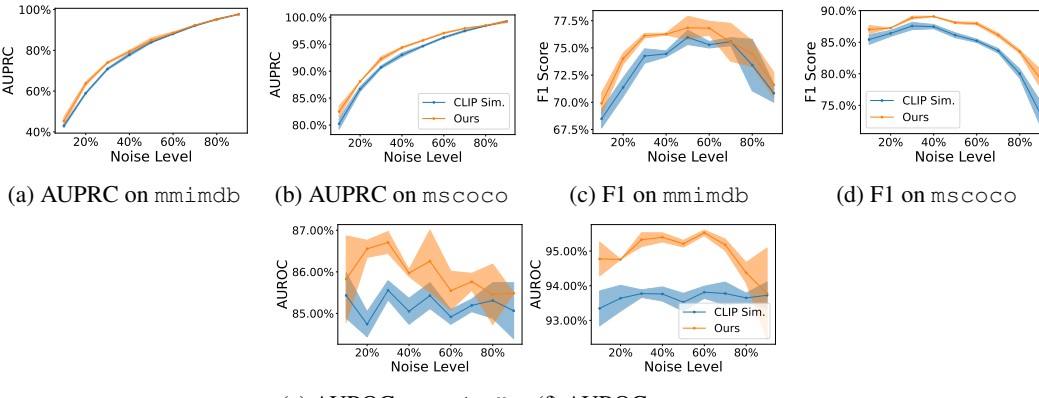

(a) AUPRC on `mmimdb`  (b) AUPRC on `mscoco`  (c) F1 on `mmimdb`  (d) F1 on `mscoco`

(e) AUROC on `mmimdb`  (f) AUROC on `mscoco`

Figure I.1: Test-set performance of LEMoN$_{\text{OPT}}$ compared to the CLIP similarity baseline for varying levels of the synthetic noise.

Table I.2: Label error detection performance on synthetic errors

| Dataset | Flip Type | Method | AUROC (%) | | AUPRC (%) | | F1 (%) | |
| | | | mean | std | mean | std | mean | std |
|---|---|---|---|---|---|---|---|---|
| cifar10 | asymmetric | AUM | 93.6% | 0.6% | 86.6% | 0.6% | 88.9% | 0.8% |
| | | Confident | 96.2% | 0.8% | 91.3% | 1.6% | 95.0% | 1.0% |
| | | CLIP Logits | 98.8% | 0.2% | 97.9% | 0.3% | 94.3% | 0.4% |
| | | CLIP Sim. | 98.2% | 0.2% | 97.1% | 0.3% | 93.4% | 0.1% |
| | | Datamap | 93.6% | 0.5% | 86.2% | 0.8% | 88.2% | 0.8% |
| | | Simifeat-V | 69.8% | 0.5% | 58.4% | 0.9% | 60.4% | 0.7% |
| | | Simifeat-R | 70.1% | 0.5% | 58.5% | 1.0% | 61.1% | 0.7% |
| | | Deep k-NN | 85.2% | 0.7% | 66.2% | 0.9% | 81.1% | 1.2% |
| | | LEMoN$_{FIX}$ | 97.5% | 0.2% | 94.8% | 0.6% | - | - |
| | | LEMoN$_{OPT}$ | 98.8% | 0.2% | 97.8% | 0.5% | 94.9% | 0.3% |
| | symmetric | AUM | 99.8% | 0.0% | 99.7% | 0.0% | 98.4% | 0.2% |
| | | Confident | 97.6% | 0.4% | 94.1% | 1.3% | 96.8% | 0.7% |
| | | CLIP Logits | 98.5% | 0.0% | 97.9% | 0.1% | 93.4% | 0.1% |
| | | CLIP Sim. | 97.9% | 0.0% | 97.1% | 0.2% | 92.5% | 0.3% |
| | | Datamap | 99.8% | 0.0% | 99.7% | 0.0% | 98.3% | 0.1% |
| | | Simifeat-V | 96.6% | 0.0% | 94.1% | 0.1% | 94.3% | 0.1% |
| | | Simifeat-R | 96.4% | 0.2% | 93.8% | 0.5% | 94.1% | 0.3% |
| | | Deep k-NN | 99.2% | 0.1% | 98.1% | 0.2% | 96.7% | 0.3% |
| | | LEMoN$_{FIX}$ | 99.5% | 0.1% | 99.2% | 0.1% | - | - |
| | | LEMoN$_{OPT}$ | 99.6% | 0.1% | 99.4% | 0.1% | 97.3% | 0.2% |
| cifar100 | asymmetric | AUM | 82.4% | 2.0% | 67.5% | 2.6% | 75.2% | 1.5% |
| | | Confident | 63.0% | 1.9% | 48.4% | 1.1% | 59.0% | 1.5% |
| | | CLIP Logits | 96.6% | 0.3% | 94.8% | 0.5% | 90.1% | 0.7% |
| | | CLIP Sim. | 94.7% | 0.5% | 92.7% | 0.7% | 87.3% | 0.4% |
| | | Datamap | 74.0% | 1.8% | 58.7% | 2.3% | 65.4% | 1.5% |
| | | Simifeat-V | 65.5% | 1.5% | 52.5% | 1.8% | 57.3% | 1.9% |
| | | Simifeat-R | 65.3% | 1.3% | 53.0% | 1.6% | 56.7% | 1.8% |
| | | Deep k-NN | 63.3% | 0.8% | 48.3% | 1.1% | 55.9% | 0.6% |
| | | LEMoN$_{FIX}$ | 94.9% | 0.3% | 92.1% | 0.4% | - | - |
| | | LEMoN$_{OPT}$ | 96.6% | 0.3% | 95.1% | 0.2% | 90.0% | 0.5% |
| | symmetric | AUM | 99.2% | 0.3% | 99.0% | 0.5% | 96.0% | 1.0% |
| | | Confident | 88.3% | 0.9% | 75.3% | 1.7% | 85.3% | 1.2% |
| | | CLIP Logits | 96.8% | 0.1% | 95.2% | 0.3% | 90.7% | 0.4% |
| | | CLIP Sim. | 95.1% | 0.3% | 93.2% | 0.5% | 87.6% | 0.0% |
| | | Datamap | 99.2% | 0.4% | 98.8% | 0.7% | 95.9% | 1.0% |
| | | Simifeat-V | 91.2% | 0.5% | 85.0% | 1.2% | 84.8% | 0.7% |
| | | Simifeat-R | 90.9% | 0.6% | 84.6% | 1.2% | 84.5% | 0.9% |
| | | Deep k-NN | 96.7% | 0.1% | 91.7% | 0.3% | 92.3% | 0.4% |
| | | LEMoN$_{FIX}$ | 98.4% | 0.1% | 97.7% | 0.2% | - | - |
| | | LEMoN$_{OPT}$ | 99.0% | 0.0% | 98.7% | 0.1% | 95.1% | 0.1% |

Table I.3: Label error detection performance on captioning datasets.

| Dataset | Method | AUROC (%) | AUPRC (%) | F1 (%) |
|---|---|---|---|---|
| flickr30k | LLaVA | 79.3 (0.8) | 58.5 (0.2) | 65.0 (1.1) |
| | Datamap | 54.0 (1.8) | 38.8 (0.6) | 28.2 (2.1) |
| | Discrepancy | 73.0 (0.6) | 59.2 (1.8) | 64.7 (1.7) |
| | Deep k-NN | 71.1 (0.4) | 52.0 (1.0) | 64.8 (2.7) |
| | Confident | 61.6 (0.5) | 40.6 (0.6) | 54.3 (0.8) |
| | CLIP Sim. | **94.8** (0.5) | **92.8** (0.5) | **88.1** (0.7) |
| | LEMoN$_{\text{FIX}}$ (Ours) | 93.6 (0.2) | 92.0 (0.2) | - |
| | LEMoN$_{\text{OPT}}$ (Ours) | 94.5 (0.2) | **92.8** (0.3) | 87.7 (0.9) |
| | CapFilt (Oracle) | 98.6 (0.1) | 98.1 (0.1) | 94.8 (0.5) |
| mscoco | LLaVA | 80.3 (0.1) | 63.4 (0.3) | 74.9 (0.3) |
| | Datamap | 49.9 (0.7) | 40.3 (0.5) | 28.6 (0.0) |
| | Discrepancy | 72.7 (0.3) | 67.2 (0.4) | 67.3 (0.9) |
| | Deep k-NN | 76.6 (0.4) | 70.3 (0.6) | 73.2 (0.3) |
| | Confident | 66.4 (1.2) | 52.1 (1.2) | 58.9 (1.5) |
| | CLIP Sim. | 93.8 (0.2) | 93.0 (0.4) | 87.5 (0.3) |
| | LEMoN$_{\text{FIX}}$ (Ours) | 92.0 (0.1) | 91.8 (0.3) | - |
| | LEMoN$_{\text{OPT}}$ (Ours) | **95.6** (0.2) | **94.6** (0.3) | **89.3** (0.2) |
| | CapFilt (Oracle) | 99.3 (0.0) | 99.1 (0.0) | 96.2 (0.3) |
| mmimdb | LLaVA | 58.4 (0.2) | 46.4 (0.2) | 58.5 (0.1) |
| | Discrepancy | 57.4 (0.4) | 45.5 (0.9) | 40.2 (1.7) |
| | Datamap | 50.1 (0.5) | 40.0 (0.3) | 28.9 (0.3) |
| | deep k-NN | 58.7 (0.7) | 45.0 (0.5) | 44.5 (1.0) |
| | Confident | 52.8 (0.8) | 41.4 (0.4) | 53.6 (0.7) |
| | CLIP Sim. | 85.1 (0.3) | 77.8 (0.7) | 74.5 (0.3) |
| | LEMoN$_{\text{FIX}}$ (Ours) | 84.3 (0.3) | 77.7 (0.8) | - |
| | LEMoN$_{\text{OPT}}$ (Ours) | **86.0** (0.1) | **79.4** (0.6) | **76.3** (0.1) |
| | CapFilt | 82.7 (0.7) | 73.3 (1.2) | 71.6 (0.8) |
| mimiccxr | LLaVA | 53.9 (0.5) | 42.7 (0.7) | 28.7 (0.1) |
| | Datamap | 50.2 (0.9) | 39.5 (0.9) | 28.9 (0.4) |
| | Discrepancy | 60.0 (0.8) | 50.3 (0.7) | 32.8 (2.8) |
| | deep k-NN | 62.9 (0.4) | 48.0 (0.3) | 46.0 (4.4) |
| | Confident | 60.2 (0.3) | 45.6 (0.3) | 59.4 (0.1) |
| | CLIP Sim. | 64.1 (0.4) | 51.7 (0.5) | 48.6 (3.4) |
| | LEMoN$_{\text{FIX}}$ (Ours) | 66.5 (0.2) | 54.8 (0.4) | - |
| | LEMoN$_{\text{OPT}}$ (Ours) | **70.4** (2.3) | **60.3** (2.3) | **57.0** (1.6) |
| | CapFilt | 49.2 (0.3) | 39.3 (0.6) | 28.5 (0.0) |

Table I.4: `flickr30k`: Label Error Detection

| Dataset | Noise Type | Method | AUROC | | AUPRC | | F1 | |
|---|---|---|---|---|---|---|---|---|
| | | | mean | std | mean | std | mean | std |
| flickr30k | noun | **LLAVA** | 79.3% | 0.8% | 58.5% | 0.2% | 65.0% | 1.1% |
| | | **captfilt** | 98.6% | 0.1% | 98.1% | 0.1% | 94.8% | 0.5% |
| | | **Datamap** | 54.0% | 1.8% | 38.8% | 0.6% | 28.2% | 2.1% |
| | | **Deep kNN** | 71.1% | 0.4% | 52.0% | 1.0% | 64.8% | 2.7% |
| | | **Confident** | 61.6% | 0.5% | 40.6% | 0.6% | 54.3% | 0.8% |
| | | **CLIP Sim.** | 94.8% | 0.5% | 92.8% | 0.5% | 88.1% | 0.7% |
| | | **LEMoN$_{\text{FIX}}$** | 93.6% | 0.2% | 92.0% | 0.2% | - | - |
| | | **LEMoN$_{\text{OPT}}$** | 94.5% | 0.2% | 92.8% | 0.3% | 87.7% | 0.9% |
| | random | **LLAVA** | 81.3% | 1.0% | 65.6% | 1.4% | 72.2% | 1.1% |
| | | **captfilt** | 99.9% | 0.0% | 99.8% | 0.0% | 98.3% | 0.2% |
| | | **Datamap** | 50.1% | 1.5% | 40.6% | 1.3% | 29.6% | 0.9% |
| | | **Deep kNN** | 81.1% | 1.6% | 65.3% | 1.8% | 73.0% | 1.0% |
| | | **Confident** | 68.5% | 1.8% | 52.0% | 1.5% | 66.3% | 1.6% |
| | | **CLIP Sim.** | 99.5% | 0.1% | 99.3% | 0.1% | 96.4% | 0.4% |
| | | **LEMoN$_{\text{FIX}}$** | 99.4% | 0.2% | 99.3% | 0.2% | - | - |
| | | **LEMoN$_{\text{OPT}}$** | 99.5% | 0.2% | 99.4% | 0.3% | 96.9% | 0.8% |

Table I.5: `msccoco`: Label Error Detection

| Dataset | Noise Type | Method | AUROC | | AUPRC | | F1 | |
|---|---|---|---|---|---|---|---|---|
| | | | mean | std | mean | std | mean | std |
| mscoco | cat | LLAVA | 80.3% | 0.1% | 63.4% | 0.3% | 74.9% | 0.3% |
| | | captfilt | 99.3% | 0.0% | 99.1% | 0.0% | 96.2% | 0.3% |
| | | Datamap | 49.9% | 0.7% | 40.3% | 0.5% | 28.6% | 0.0% |
| | | Deep kNN | 76.6% | 0.4% | 70.3% | 0.6% | 73.2% | 0.3% |
| | | Confident | 66.4% | 1.2% | 52.1% | 1.2% | 58.9% | 1.5% |
| | | CLIP Sim. | 93.8% | 0.2% | 93.0% | 0.4% | 87.5% | 0.3% |
| | | LEMoN$_{FIX}$ | 92.0% | 0.1% | 91.8% | 0.3% | - | - |
| | | LEMoN$_{OPT}$ | 95.6% | 0.2% | 94.6% | 0.3% | 89.3% | 0.2% |
| | noun | LLAVA | 79.4% | 0.2% | 61.3% | 0.3% | 72.6% | 0.2% |
| | | captfilt | 98.7% | 0.2% | 98.4% | 0.2% | 94.9% | 0.4% |
| | | Datamap | 51.2% | 1.4% | 39.4% | 1.4% | 27.8% | 0.4% |
| | | Deep kNN | 76.1% | 1.3% | 68.9% | 1.2% | 72.3% | 1.0% |
| | | Confident | 64.6% | 1.1% | 48.4% | 1.1% | 55.6% | 1.9% |
| | | CLIP Sim. | 92.1% | 0.2% | 90.5% | 0.2% | 84.8% | 0.7% |
| | | LEMoN$_{FIX}$ | 90.4% | 0.5% | 89.5% | 0.4% | - | - |
| | | LEMoN$_{OPT}$ | 92.9% | 0.5% | 91.5% | 0.5% | 86.1% | 0.3% |
| | random | LLAVA | 82.6% | 0.3% | 65.1% | 0.6% | 76.7% | 0.2% |
| | | captfilt | 99.9% | 0.0% | 99.9% | 0.0% | 99.1% | 0.1% |
| | | Datamap | 49.9% | 0.2% | 40.2% | 0.3% | 28.6% | 0.0% |
| | | Deep kNN | 93.8% | 0.2% | 85.8% | 0.3% | 89.2% | 0.5% |
| | | Confident | 83.5% | 1.5% | 69.4% | 2.3% | 80.2% | 1.6% |
| | | CLIP Sim. | 99.5% | 0.1% | 99.4% | 0.1% | 97.6% | 0.1% |
| | | LEMoN$_{FIX}$ | 99.5% | 0.2% | 99.4% | 0.1% | - | - |
| | | LEMoN$_{OPT}$ | 99.6% | 0.1% | 99.5% | 0.1% | 97.9% | 0.1% |

Table I.6: `mmimdb`: Label Error Detection

| Dataset | Noise Type | Method | AUROC | | AUPRC | | F1 | |
|---|---|---|---|---|---|---|---|---|
| | | | mean | std | mean | std | mean | std |
| mmimdb | cat | LLAVA | 58.4% | 0.2% | 46.4% | 0.2% | 58.5% | 0.1% |
| | | captfilt | 82.7% | 0.7% | 73.3% | 1.2% | 71.6% | 0.8% |
| | | Datamap | 50.1% | 0.5% | 40.0% | 0.3% | 28.9% | 0.3% |
| | | Deep kNN | 58.7% | 0.7% | 45.0% | 0.5% | 44.5% | 1.0% |
| | | Confident | 52.8% | 0.8% | 41.4% | 0.4% | 53.6% | 0.7% |
| | | CLIP Sim. | 85.1% | 0.3% | 77.8% | 0.7% | 74.5% | 0.3% |
| | | LEMoN$_{FIX}$ | 84.3% | 0.3% | 77.7% | 0.8% | - | - |
| | | LEMoN$_{OPT}$ | 86.0% | 0.1% | 79.4% | 0.6% | 76.3% | 0.1% |
| | noun | LLAVA | 59.1% | 0.3% | 44.2% | 0.6% | 55.2% | 0.2% |
| | | captfilt | 79.9% | 0.1% | 66.2% | 0.4% | 70.0% | 0.3% |
| | | Datamap | 50.3% | 0.4% | 37.2% | 0.7% | 28.0% | 1.5% |
| | | Deep kNN | 61.4% | 0.1% | 44.2% | 0.3% | 45.3% | 4.1% |
| | | Confident | 52.1% | 2.2% | 38.0% | 1.3% | 50.3% | 1.6% |
| | | CLIP Sim. | 82.8% | 0.4% | 72.8% | 0.5% | 72.7% | 0.4% |
| | | LEMoN$_{FIX}$ | 82.1% | 0.4% | 72.7% | 0.6% | - | - |
| | | LEMoN$_{OPT}$ | 84.4% | 0.2% | 75.9% | 1.2% | 75.2% | 0.1% |
| | random | LLAVA | 58.5% | 0.8% | 46.7% | 0.5% | 58.5% | 0.1% |
| | | captfilt | 84.9% | 0.4% | 76.4% | 0.7% | 73.6% | 0.2% |
| | | Datamap | 50.6% | 0.2% | 40.4% | 0.4% | 29.3% | 0.6% |
| | | Deep kNN | 62.1% | 0.5% | 47.3% | 0.3% | 50.0% | 0.6% |
| | | Confident | 52.9% | 1.8% | 41.5% | 0.9% | 54.1% | 2.1% |
| | | CLIP Sim. | 88.1% | 0.1% | 82.0% | 0.2% | 78.2% | 0.9% |
| | | LEMoN$_{FIX}$ | 87.6% | 0.1% | 81.9% | 0.3% | - | - |
| | | LEMoN$_{OPT}$ | 89.4% | 0.3% | 84.1% | 0.8% | 80.1% | 0.4% |

Table I.7: `mimiccxr`: Label Error Detection

| Dataset | Noise Type | Method | AUROC | | AUPRC | | F1 | |
|---|---|---|---|---|---|---|---|---|
| | | | mean | std | mean | std | mean | std |
| mimiccxr | cat | LLAVA | 53.9% | 0.5% | 42.7% | 0.7% | 28.7% | 0.1% |
| | | captfilt | 49.2% | 0.3% | 39.3% | 0.6% | 28.5% | 0.0% |
| | | Datamap | 50.2% | 0.9% | 39.5% | 0.9% | 28.9% | 0.4% |
| | | Deep kNN | 62.9% | 0.4% | 48.0% | 0.3% | 46.0% | 4.4% |
| | | Confident | 60.2% | 0.3% | 45.6% | 0.3% | 59.4% | 0.1% |
| | | CLIP Sim. | 64.1% | 0.4% | 51.7% | 0.5% | 48.6% | 3.4% |
| | | LEMoN$_{\text{FIX}}$ | 66.6% | 0.2% | 54.8% | 0.4% | - | - |
| | | LEMoN$_{\text{OPT}}$ | 70.4% | 2.3% | 60.3% | 2.3% | 57.0% | 1.6% |
| | random | LLAVA | 50.8% | 0.4% | 40.6% | 0.2% | 57.1% | 0.0% |
| | | captfilt | 50.8% | 0.4% | 40.5% | 0.7% | 28.6% | 0.0% |
| | | Datamap | 51.1% | 0.9% | 40.7% | 0.5% | 28.8% | 0.2% |
| | | Confident | 61.1% | 0.7% | 46.3% | 0.5% | 60.7% | 0.5% |
| | | CLIP Sim. | 66.8% | 0.8% | 54.4% | 0.9% | 54.3% | 1.0% |
| | | LEMoN$_{\text{FIX}}$ | 69.5% | 0.7% | 57.8% | 1.0% | - | - |
| | | LEMoN$_{\text{OPT}}$ | 73.1% | 0.9% | 63.0% | 2.0% | 63.1% | 3.6% |

Table I.8: We show the AUROC and AUPRC of LEMoN when we search for the optimal hyper-parameters using a labeled validation set (LEMoN$_{\text{OPT}}$) and when we use fixed hyperparameters (LEMoN$_{\text{FIX}}$: $k = 30$, cosine distance, $\beta = \gamma = 5$, $\tau_{1,n} = \tau_{1,m} = 0.1$, and $\tau_{2,n} = \tau_{2,m} = 5$). The mean gap in AUROC is -1.6 (1.3), and the mean gap in AUPRC is -1.6 (2.2). Note that F1 is not computed as it requires external information to select a threshold.

| | | AUROC | | | AUPRC | | |
|---|---|---|---|---|---|---|---|
| Dataset | Noise Type | LEMoN$_{\text{OPT}}$ | LEMoN$_{\text{FIX}}$ | Gap | LEMoN$_{\text{OPT}}$ | LEMoN$_{\text{FIX}}$ | Gap |
| cifar10 | asymmetric | 98.8 (0.2) | 97.5 (0.2) | -1.4 (0.1) | 97.8 (0.5) | 94.8 (0.6) | -3.0 (0.1) |
| | real | 98.1 (0.0) | 97.7 (0.2) | -0.5 (0.2) | 97.4 (0.1) | 96.8 (0.3) | -0.5 (0.2) |
| | symmetric | 99.6 (0.1) | 99.5 (0.1) | -0.2 (0.1) | 99.4 (0.1) | 99.2 (0.1) | -0.2 (0.1) |
| cifar100 | asymmetric | 96.6 (0.3) | 94.9 (0.3) | -1.8 (0.0) | 95.1 (0.2) | 92.1 (0.4) | -3.0 (0.2) |
| | real | 90.8 (0.0) | 88.9 (0.7) | -1.8 (0.7) | 87.4 (0.3) | 84.6 (1.1) | -2.8 (0.9) |
| | symmetric | 99.0 (0.0) | 98.4 (0.1) | -0.7 (0.1) | 98.7 (0.1) | 97.7 (0.2) | -1.0 (0.1) |
| miniImageNet | human | 90.2 (0.2) | 89.5 (0.2) | -0.7 (0.2) | 81.4 (1.3) | 81.5 (0.3) | 0.0 (0.1) |
| StanfordCars | human | 73.1 (0.5) | 72.6 (0.7) | -0.7 (0.1) | 40.5 (0.5) | 44.9 (1.4) | 4.3 (0.7) |
| flickr30k | noun | 94.5 (0.2) | 93.6 (0.2) | -0.9 (0.3) | 92.8 (0.3) | 92.0 (0.2) | -0.8 (0.1) |
| | random | 99.5 (0.2) | 99.4 (0.2) | -0.0 (0.1) | 99.4 (0.3) | 99.3 (0.2) | -0.1 (0.2) |
| mimiccxr | cat | 70.4 (2.3) | 66.5 (0.2) | -3.9 (2.1) | 60.3 (2.3) | 54.8 (0.4) | -5.5 (1.9) |
| | random | 73.1 (0.9) | 69.5 (0.7) | -3.6 (0.2) | 63.0 (2.0) | 57.8 (1.0) | -5.1 (1.0) |
| mmimdb | cat | 86.0 (0.1) | 84.3 (0.3) | -1.6 (0.3) | 79.4 (0.6) | 77.7 (0.8) | -1.7 (0.2) |
| | noun | 84.4 (0.2) | 82.1 (0.4) | -2.3 (0.4) | 75.9 (1.2) | 72.7 (0.6) | -3.2 (0.8) |
| | random | 89.4 (0.3) | 87.6 (0.1) | -1.8 (0.4) | 84.1 (0.8) | 81.9 (0.3) | -2.2 (0.8) |
| mscoco | cat | 95.6 (0.2) | 92.0 (0.1) | -3.6 (0.1) | 94.6 (0.3) | 91.8 (0.3) | -2.8 (0.1) |
| | noun | 92.9 (0.5) | 90.4 (0.5) | -2.5 (0.2) | 91.5 (0.5) | 89.5 (0.4) | -2.0 (0.3) |
| | random | 99.6 (0.1) | 99.5 (0.2) | -0.1 (0.0) | 99.5 (0.1) | 99.4 (0.1) | -0.1 (0.0) |

Table I.9: Performance of our method after ablating various components. We find that mislabel detection performance almost decreases monotonically as we remove additional components, with the exception of two metrics on `mmimdb` where one ablation is statistically comparable to the original method.

| | mmimdb | | | mscoco | | |
|---|---|---|---|---|---|---|
| | AUROC | AUPRC | F1 | AUROC | AUPRC | F1 |
| LEMoN$_{\text{OPT}}$ (Ours) | 86.0 (0.1) | 79.4 (0.6) | **76.3** (0.1) | **95.5** (0.1) | **94.5** (0.3) | **89.3** (0.3) |
| $-\tau_1$ | 85.3 (0.3) | 78.2 (1.1) | 75.4 (0.5) | 94.6 (0.3) | 93.8 (0.4) | 88.0 (0.5) |
| $-\tau_2$ | 85.4 (0.6) | 77.1 (2.4) | 75.4 (0.2) | 94.7 (0.3) | 93.6 (0.5) | 87.7 (0.8) |
| $-\tau_1, \tau_2$ | 85.4 (0.2) | 78.1 (0.7) | 75.2 (0.3) | 94.7 (0.3) | 93.8 (0.5) | 88.0 (0.8) |
| $-s_n$ | **86.1** (0.3) | **79.6** (0.5) | 76.1 (1.1) | 94.6 (0.3) | 93.6 (0.5) | 87.5 (0.6) |
| $-s_m$ | 85.3 (0.3) | 77.9 (0.7) | 75.5 (0.4) | 94.9 (0.2) | 94.0 (0.4) | 89.0 (0.6) |
| $-s_n, s_m$ (CLIP Sim.) | 85.1 (0.3) | 77.8 (0.7) | 74.5 (0.3) | 93.8 (0.2) | 93.0 (0.4) | 87.5 (0.3) |

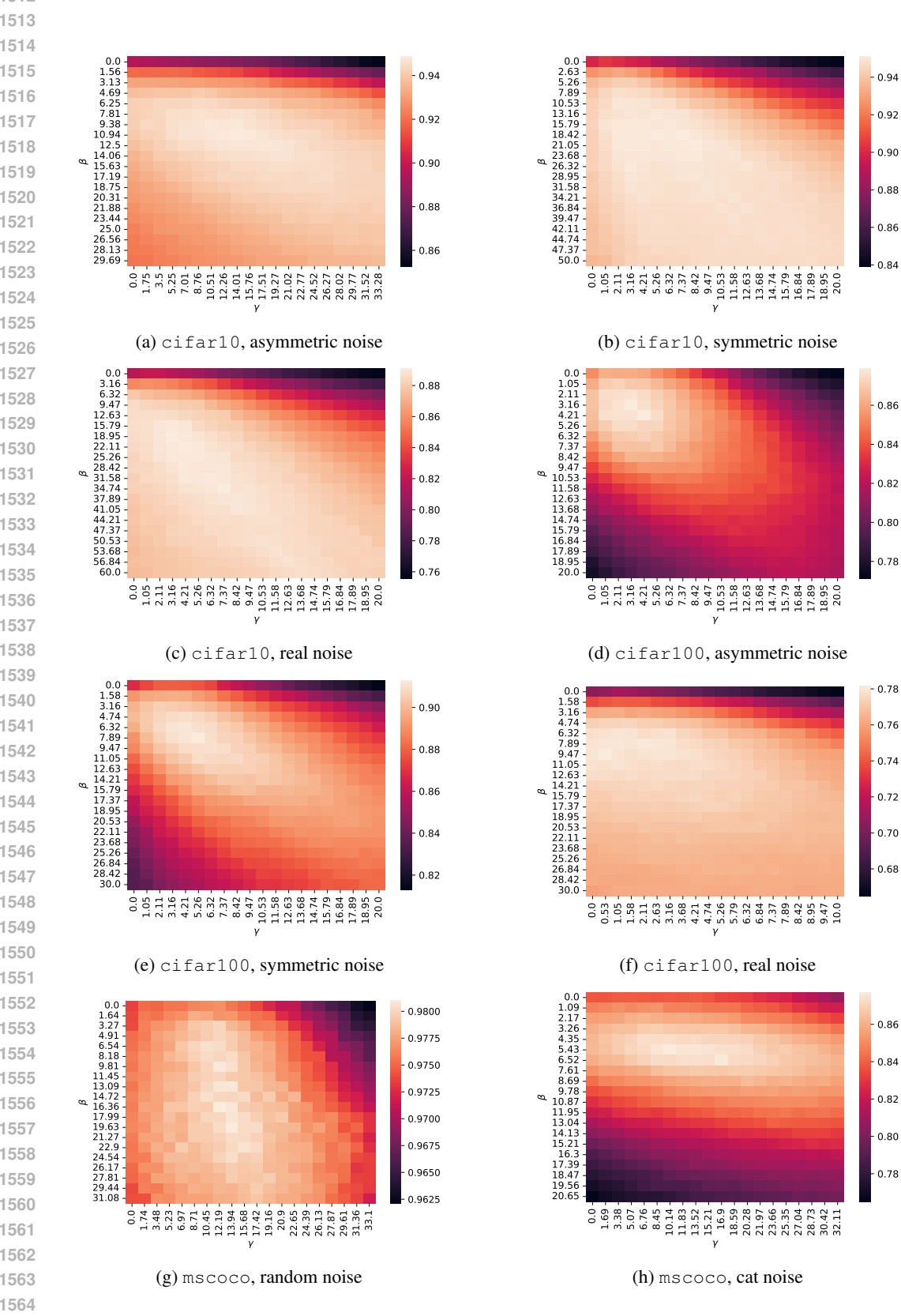

(a) `cifar10`, asymmetric noise

(b) `cifar10`, symmetric noise

(c) `cifar10`, real noise

(d) `cifar100`, asymmetric noise

(e) `cifar100`, symmetric noise

(f) `cifar100`, real noise

(g) `mscoco`, random noise

(h) `mscoco`, cat noise

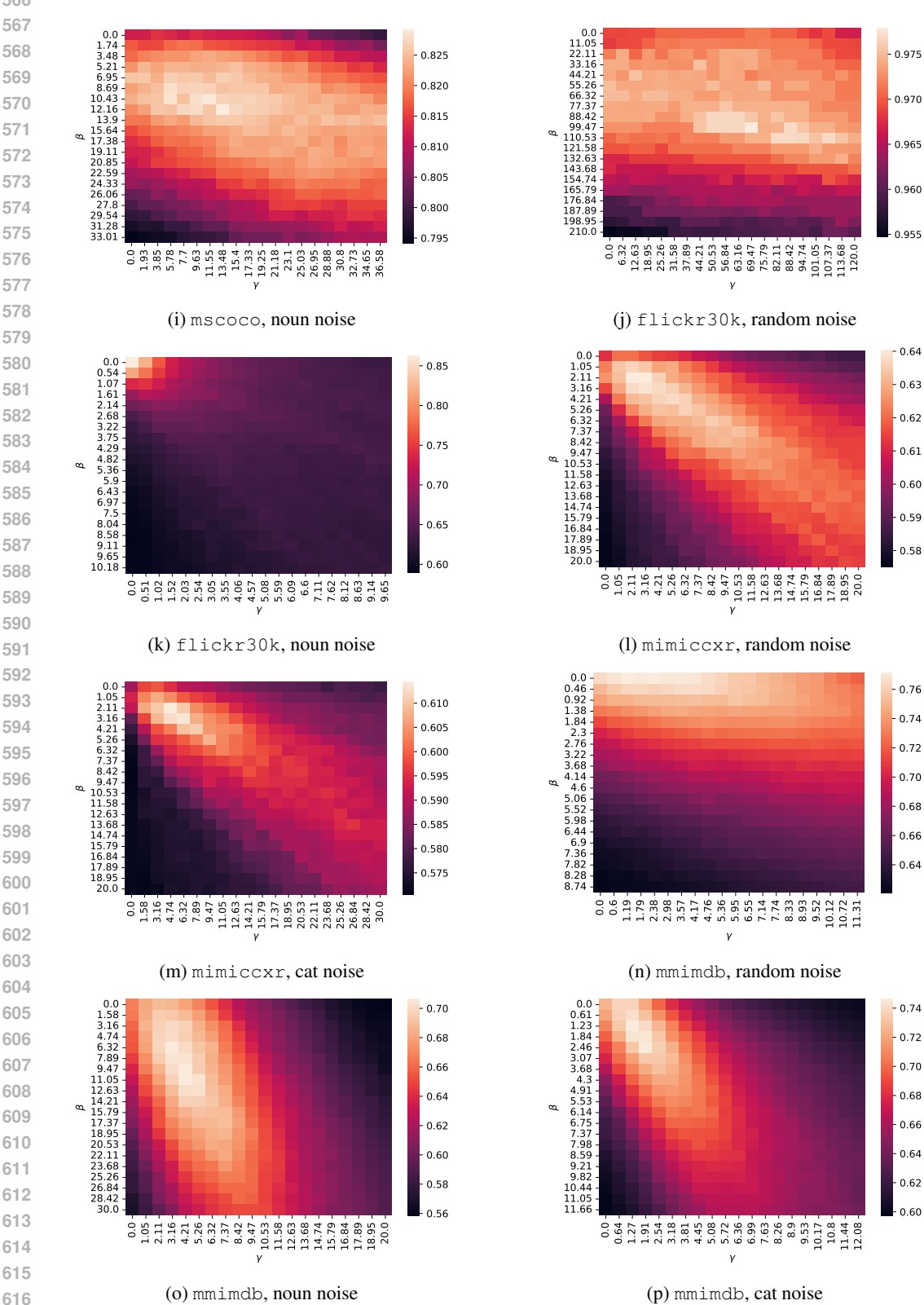

(i) `mscoco`, noun noise

(j) `flickr30k`, random noise

(k) `flickr30k`, noun noise

(l) `mimiccxr`, random noise

(m) `mimiccxr`, cat noise

(n) `mmimdb`, random noise

(o) `mmimdb`, noun noise

(p) `mmimdb`, cat noise

Figure I.2: F1 of our method for varying $\beta$ and $\gamma$, keeping all other hyperparameters their fixed optimal values.

Table I.10: AUROC of label error detection for each component of our score. We find that $d_{mm}$ is the most critical component of the score. Of the two nearest neighbor terms, we find that $s_n$ (nearest image neighbors) is the more important term for most datasets.

| | cifar10 | cifar100 | miniImageNet | stanfordCars | flickr30k | mscoco | mmimdb | mimiccxr |
|---|---|---|---|---|---|---|---|---|
| $d_{mm}$ (CLIP Sim.) | 93.8 (0.1) | 78.5 (0.6) | 89.3 (0.2) | 69.8 (0.6) | 94.8 (0.5) | 93.8 (0.2) | 85.1 (0.3) | 64.1 (0.4) |
| $s_m$ | 79.3 (2.8) | 65.4 (2.0) | 80.8 (0.3) | 66.0 (0.9) | 76.3 (1.8) | 75.8 (0.3) | 60.1 (0.4) | 59.0 (0.6) |
| $s_n$ | 98.1 (0.0) | 88.4 (0.1) | 84.3 (0.2) | 72.8 (0.7) | 71.4 (1.6) | 76.5 (0.5) | 55.1 (0.3) | 57.9 (2.1) |
| $d_{mm} + s_m$ | 92.5 (0.5) | 81.3 (1.1) | 89.6 (0.2) | 69.7 (0.5) | **95.0** (0.5) | 94.6 (0.3) | **86.0** (0.4) | 64.5 (0.6) |
| $s_n + s_m$ | 98.0 (0.2) | 88.8 (0.2) | 84.5 (0.4) | 72.8 (0.7) | 83.5 (0.5) | 86.1 (0.6) | 67.6 (0.9) | 63.6 (0.6) |
| $d_{mm} + s_n$ | **98.2** (0.1) | **90.8** (0.1) | 89.9 (0.3) | **73.9** (0.7) | 94.9 (0.3) | 94.9 (0.2) | 85.3 (0.3) | 66.4 (2.4) |
| $d_{mm} + s_n + s_m$ (LEMoN) | 98.1 (0.0) | **90.8** (0.0) | **90.2** (0.2) | 73.1 (0.5) | 94.5 (0.2) | **95.6** (0.2) | **86.0** (0.1) | **70.4** (2.3) |

Table I.11: Average per-sample runtime (miliseconds) of each method for label error detection. Standard deviation across 3 random data seeds are shown in parentheses.

| | cifar10 | cifar100 | miniImageNet | stanfordCars | mscoco | flickr30k | mimiccxr | mmimdb |
|---|---|---|---|---|---|---|---|---|
| LEMoN | 10.1 (0.5) | 9.6 (0.5) | 7.8 (1.6) | 11.0 (2.0) | 18.8 (1.8) | 35.9 (1.2) | 52.2 (2.7) | 21.1 (1.4) |
| CLIP Sim. | 1.8 (0.0) | 1.8 (0.0) | 2.7 (0.4) | 3.5 (0.5) | 20.3 (0.0) | 15.6 (0.0) | 16.8 (0.0) | 30.5 (0.0) |
| Deep kNN | 7.0 (1.3) | 5.1 (0.1) | 8.7 (1.2) | 6.0 (0.1) | 19.9 (0.9) | 10.6 (1.2) | 47.1 (12.7) | 20.5 (1.9) |
| Datamap | 37.6 (0.2) | 37.5 (0.3) | 37.7 (1.6) | 37.2 (0.3) | 39.7 (0.1) | 38.1 (4.8) | 41.4 (1.3) | 62.6 (9.5) |

### I.7 VARYING VALIDATION SET SIZE

In Figure I.3, we examine the effect of varying validation set size (by random subsampling) on $\text{LEMON}_{\text{OPT}}$.

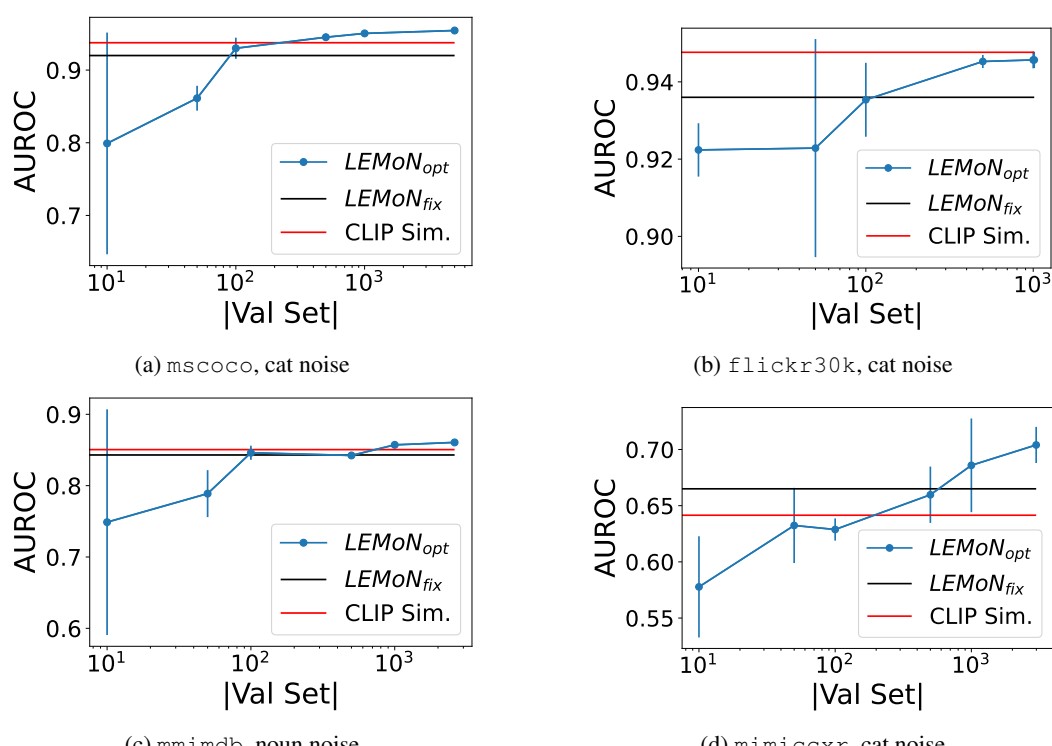

(a) `mscoco`, cat noise

(b) `flickr30k`, cat noise

(c) `mmimdb`, noun noise

(d) `mimiccxr`, cat noise

Figure I.3: Test-set AUROC of mislabel detection with varying size of the labeled validation set for $\text{LEMON}_{\text{OPT}}$. Note that $\text{LEMON}_{\text{FIX}}$ and CLIP Sim. do not have any hyperparameters and as such do not rely on a labeled validation set.

### I.8 EMPIRICAL COMPARISON WITH THOMAS & KOVASHKA (2022)

In Table I.12, we compare the performance of $\text{LEMON}_{\text{OPT}}$ against the four scores proposed in Thomas & Kovashka (2022), using the datasets and noise types shown in Table 1.

Table I.12: Comparison of label error detection performance of LEMoN versus baselines from Thomas & Kovashka (2022).

| | AUROC | | | | | AUPRC | | | | | F1 | | | | |
|---|---|---|---|---|---|---|---|---|---|---|---|---|---|---|---|
| | $\Upsilon_X^{DIS}$ | $\Upsilon_Y^{DIS}$ | $\Upsilon_X^{DIV}$ | $\Upsilon_Y^{DIV}$ | $\text{LEMON}_{\text{OPT}}$ | $\Upsilon_X^{DIS}$ | $\Upsilon_Y^{DIS}$ | $\Upsilon_X^{DIV}$ | $\Upsilon_Y^{DIV}$ | $\text{LEMON}_{\text{OPT}}$ | $\Upsilon_X^{DIS}$ | $\Upsilon_Y^{DIS}$ | $\Upsilon_X^{DIV}$ | $\Upsilon_Y^{DIV}$ | $\text{LEMON}_{\text{OPT}}$ |
| cifar10 | 77.1 (1.9) | 48.2 (1.2) | 50.3 (1.9) | 45.0 (1.9) | **98.1** (0.0) | 70.4 (2.7) | 41.2 (1.1) | 41.6 (1.6) | 38.9 (2.1) | **97.4** (0.1) | 68.2 (1.9) | 29.2 (0.4) | 29.2 (0.4) | 29.2 (0.4) | **93.1** (0.2) |
| cifar100 | 66.0 (1.5) | 49.4 (1.1) | 49.9 (1.4) | 49.7 (1.9) | **90.8** (0.0) | 57.4 (2.3) | 39.2 (0.7) | 39.9 (1.2) | 39.3 (0.8) | **87.4** (0.3) | 51.9 (1.8) | 29.4 (1.4) | 32.5 (5.5) | 29.4 (0.4) | **81.3** (0.2) |
| miniImageNet | 79.4 (0.3) | 47.4 (0.5) | 64.6 (0.2) | 48.0 (0.5) | **90.2** (0.2) | 65.6 (0.7) | 32.5 (0.0) | 46.3 (0.2) | 32.7 (0.8) | **81.4** (1.3) | 69.8 (0.4) | 28.0 (2.3) | 55.8 (2.3) | 27.0 (0.9) | **82.3** (0.1) |
| stanfordCars | 65.7 (0.7) | 50.8 (1.1) | 51.9 (0.9) | 50.1 (0.5) | **73.1** (0.5) | 33.1 (0.6) | 23.3 (0.7) | 24.5 (0.8) | 23.4 (0.2) | **40.5** (0.5) | 59.9 (0.4) | 20.6 (1.3) | 25.3 (5.6) | 20.6 (1.4) | **67.3** (1.0) |
| flickr30k | 73.0 (0.6) | 53.3 (1.4) | 49.9 (2.9) | 52.9 (0.2) | **94.5** (0.2) | 59.2 (1.8) | 37.1 (1.8) | 33.7 (2.4) | 37.0 (0.8) | **92.8** (0.3) | 64.7 (1.7) | 26.2 (0.8) | 27.4 (1.7) | 26.1 (1.0) | **87.7** (0.9) |
| mimiccxr | 60.0 (0.8) | 49.6 (0.4) | 50.0 (1.3) | 49.1 (1.3) | **70.4** (2.3) | 50.3 (0.7) | 39.3 (0.5) | 39.8 (1.2) | 39.6 (0.7) | **60.3** (2.3) | 32.8 (2.8) | 28.5 (0.0) | 28.5 (0.0) | 28.5 (0.0) | **57.0** (1.6) |
| mmimdb | 57.4 (0.4) | 49.8 (0.4) | 48.6 (0.4) | 50.0 (0.5) | **86.0** (0.1) | 45.5 (0.9) | 40.1 (0.4) | 38.9 (0.5) | 40.1 (0.5) | **79.4** (0.6) | 40.2 (1.7) | 28.6 (0.1) | 29.1 (0.5) | 28.9 (0.6) | **76.3** (0.1) |
| mscoco | 72.7 (0.3) | 48.5 (0.8) | 52.9 (0.8) | 48.7 (0.3) | **95.6** (0.2) | 67.2 (0.4) | 39.1 (0.5) | 42.3 (1.0) | 39.3 (0.1) | **94.6** (0.3) | 67.3 (0.9) | 29.7 (0.1) | 29.0 (0.2) | 28.9 (0.4) | **89.3** (0.2) |

### I.9 REAL-WORLD WEB SCALE CORPUS (CC3M)

We conduct an experiment of $\text{LEMON}_{\text{FIX}}$ on CC3M (Changpinyo et al., 2021), a large web-scraped dataset of images and annotations, where we demonstrate the utility of LEMoN filtered data on CLIP pretraining. We download CC3M, which contains 2.9 million valid URLs to image-caption pairs. We then pretrain a CLIP model (ViT-B/16) from scratch on this dataset for 20 epochs, with a batch size of 128, and using a cyclic learning rate scheduler with a learning rate of $10^{-4}$.

We then use this CLIP model as the basis to compute distances for $\text{LEMON}_{\text{FIX}}$, using the reasonable hyperparameters from the main paper: $k = 30$, cosine distance, $\tau_{1,n} = \tau_{1,m} = 0.1$, and $\tau_{2,n} = \tau_{2,m} = 5$. We then select the 1 million samples with the lowest mislabel scores, filtering out the 1.9 million samples most suspected to be mislabels. We pre-train another CLIP model from scratch

Table I.13: Performance of each method on the Datacomp (Gadre et al., 2024) small benchmark from the filtering track. As of 2024/11/14, only 9.96M images ("Data Available") out of 12.8M are accessible. We compare the performance of LEMON$_{\text{OPT}}$ versus the CLIP score baseline after filtering to 3.5M images.

| | Method | ImageNet | ImageNet Dist. Shifts | VTAB | Retrieval | Avg (38 datasets) |
|---|---|---|---|---|---|---|
| Data Available | LEMON$_{\text{FIX}}$ | **0.045** | **0.053** | **0.188** | 0.116 | **0.168** |
| (9.96M Samples) | CLIP score | 0.043 | 0.049 | 0.177 | **0.119** | 0.160 |
| | No filtering | 0.025 | 0.033 | 0.145 | 0.114 | 0.132 |
| | Basic filtering | 0.038 | 0.043 | 0.150 | 0.118 | 0.142 |
| From Gadre et al. (2024) | Text-based | 0.046 | 0.052 | 0.169 | **0.125** | 0.157 |
| (12.8M Samples) | Image-based | 0.043 | 0.047 | 0.178 | 0.121 | 0.159 |
| | LAION-2B filtering | 0.031 | 0.040 | 0.136 | 0.092 | 0.133 |
| | CLIP score | **0.051** | **0.055** | **0.190** | 0.119 | **0.173** |
| | Image-based + CLIP score | 0.039 | 0.045 | 0.162 | 0.094 | 0.144 |

on this subset using the same architecture and setup as above. We evaluate the resulting model on zero-shot classification using the VTAB benchmark (Zhai et al., 2019), and compare it with CLIP models trained using data filtered to 1 million examples using the CLIP similarity baseline, and the original unfiltered model.

In Table I.9, we find that LEMON$_{\text{FIX}}$ marginally outperforms the CLIP similarity baseline on average zero-shot accuracy, though both underperform pretraining on the full corpus. One likely explanation of this is that although a large proportion of images in the CC3M dataset are technically "mislabelled" in that the caption is not a precisely correct description of the image, some substrings of these noisy captions may, on aggregate, contain useful word associations which the model learns, and thus may be useful to downstream tasks.

We examine images of images selected to be mislabels by our method in Figure I.4. We find that our method identifies images that are completely mislabeled – one cause of which is images changing after they have been indexed. In addition, our method also identifies samples which are ambiguous or imprecise.

## I.10    REAL-WORLD WEB SCALE CORPUS (DATACOMP)

We conduct an experiment of LEMON$_{\text{FIX}}$ on Datacomp (Gadre et al., 2024). We use the small dataset from the filtering track, which originally consisted of 12.8M images. As these images are accessed directly from the web, only 9.96M images were able to be downloaded as of 2024/11/14. We apply LEMON$_{\text{FIX}}$ to this dataset using OpenAI CLIP ViT-L/14 embeddings provided by Datacomp. We select the 3.5M images with lowest mislabel scores, and use the default hyperparameters from Datacomp to train a CLIP model, and evaluate it on the same 38 zero-shot classification datasets. We compare with filtering using only the CLIP score (equivalent to CLIP Sim.) to the same number of images. In Table I.13, we find that given the available images, LEMON$_{\text{FIX}}$ outperforms the baseline on average, and on three of four individual evaluations. However, neither method outperforms the scores reported in the original paper due to their dataset being larger.

## I.11    HYPERPARAMETERS USED FOR REAL-WORLD

We show the hyperparameters used for the real-world experiment in Table I.15. We use $k = 30$, cosine distance, and these hyperparameters, which originate from a hyperparameter search on synthetically noised data. We note that `flickr30k` has some negative hyperparameters, which we attribute to overfitting to a relatively small validation set during hyperparameter selection.

## I.12    EXAMPLES OF DETECTED REAL LABEL ERRORS

We show additional examples of label errors in Figure I.5.

Table I.14: Zero-shot accuracy (%) of various CLIP models on the VTAB benchmark (Zhai et al., 2019). CLIP models (ViT-B/16) are pretrained from scratch on a subset of CC3M (Changpinyo et al., 2021) which has been filtered to 1 million samples using LEMoN$_{\text{FIX}}$ and the CLIP similarity baseline, using a version of CLIP pretrained on the entire dataset.

|  | CLIP Sim. | LEMoN$_{\text{FIX}}$ | Unfiltered |
|---|---|---|---|
| caltech101 | 28.25 | **28.99** | 51.43 |
| cifar100 | **11.02** | 6.79 | 18.65 |
| clevr_closest_object_distance | 18.11 | **22.58** | 25.76 |
| clevr_count_all | **12.98** | 12.65 | 12.05 |
| dmlab | 14.78 | **16.22** | 16.62 |
| dsprites_label_orientation | **2.44** | 1.34 | 1.98 |
| dsprites_label_x_position | 3.06 | **3.20** | 3.13 |
| dsprites_label_y_position | **3.11** | 2.89 | 3.20 |
| dtd | **6.60** | 3.94 | 12.34 |
| eurosat | 14.37 | **22.07** | 9.93 |
| flowers | **6.11** | 5.19 | 6.83 |
| food101 | 4.94 | **5.31** | 9.02 |
| pets | **7.63** | 4.69 | 8.23 |
| sun397 | 13.89 | **14.22** | 24.02 |
| svhn | 7.80 | **12.35** | 8.00 |
| **Average** | 10.34 | **10.83** | 14.08 |

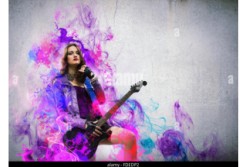

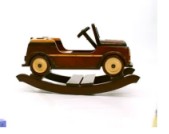

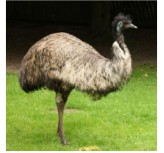

fresh milk in the glass on colour background, illustration

a very young baby girl playing with toys in a white studio

portrait of a stock photo

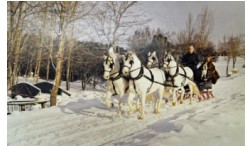

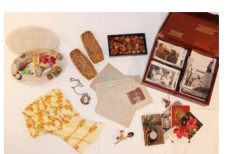

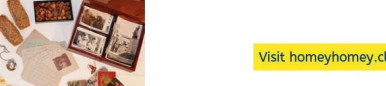

homes for sale and luxury real estate including horse farms and property in the areas

tangled tree roots on a forest trail

a park covered in yellow leaves and lined with tall trees turning bright yellow during an autumn day

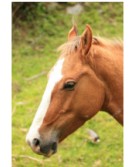

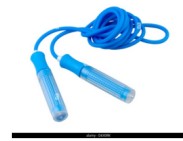

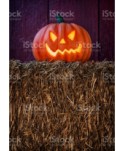

face of people -- stock photo #

begin your exercise with a jump rope easy and funny

evil looking person sitting atop a hay bale royalty - free

Figure I.4: Sample images and captions from CC3M which have been identified as mislabeled by LEMoN$_{\text{FIX}}$.

Table I.15: Hyperparameters used for the real-world experiment. We use $k = 30$, cosine distance, and the hyperparameters below, which originate from a hyperparameter search on synthetically noised data.

|            | $\beta$ | $\gamma$ | $\tau_{1,n}$ | $\tau_{2,n}$ | $\tau_{1,m}$ | $\tau_{2,m}$ |
|------------|---------|----------|--------------|--------------|--------------|--------------|
| cifar10    | 20      | 10       | 0            | 5            | 0            | 5            |
| cifar100   | 15      | 0        | 0            | 5            | 0            | 0            |
| mscoco     | 5.324   | 11.057   | 5.143        | 10.498       | 7.233        | 15.637       |
| mmimdb     | 15      | 5        | 5            | 10           | 5            | 10           |
| flickr30k  | 0.092   | -0.177   | -0.274       | -0.074       | -0.072       | 0.000        |
| mimiccxr   | 5       | 10       | 5            | 10           | 5            | 10           |

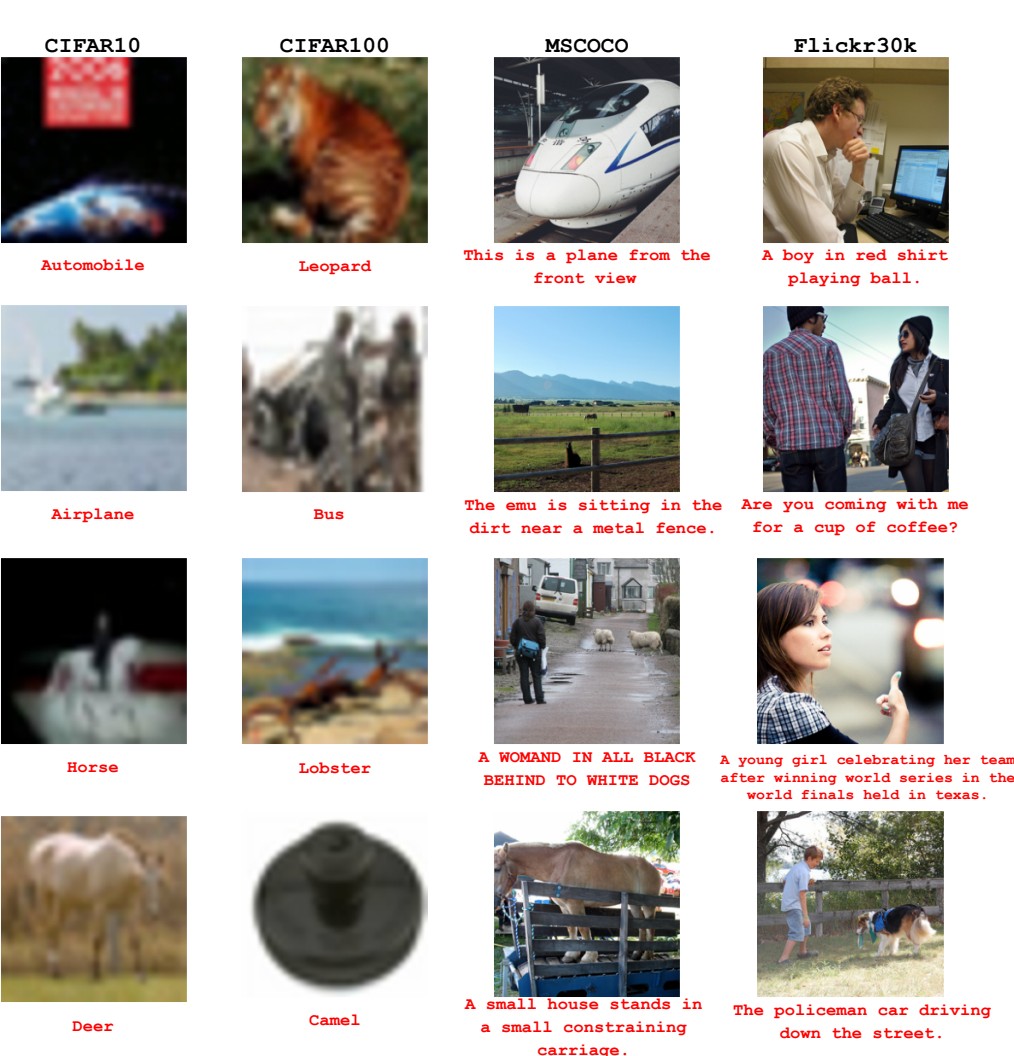

Figure I.5: Example images in each dataset identified by our method to be mislabels, and labeled as errors by a human annotator.

## I.13 COMPARISON WITH NORTHCUTT ET AL., 2021 (NORTHCUTT ET AL., 2021B)

In Northcutt et al., 2021 (Northcutt et al., 2021b), the authors utilized confident learning (Northcutt et al., 2021a) to identify suspected errors in the test sets of `cifar10` and `cifar100`. They then obtained 5 human labels for each suspected error using Amazon Mechanical Turk, and confirmed the image to be a mislabel if at least 3 of 5 workers stated so. This amounts to 54 confirmed mislabels in `cifar10` (out of 221 suspected), and 585 confirmed mislabels in `cifar100` (out of 1650 suspected). In this section, we compare the performance of LEMoN$_{\text{FIX}}$ versus the CLIP similarity baseline on this set. As this set is a subset of the images identified to be mislabels by confident learning, we are not able to compare our model performance with confident learning itself. In addition, this presents a pessimistic view (lower bound) of the performance of our method, as there are many images identified by LEMoN which *are* mislabeled, but were not selected by confident learning in (Northcutt et al., 2021b). We demonstrate examples of these images in Figure I.6.

In Table I.16, we compare the performance of LEMoN$_{\text{FIX}}$ with the CLIP similarity baseline on the error set from Northcutt et al., 2021 (Northcutt et al., 2021b). First, we compute the mean ranking of all error set samples as ranked by each method, out of 10,000 test-set samples. We find that our method ranks error set samples higher on average than the baseline, though the variance is large. Next, we subset to the top |CL Set| ranked samples for each method, and compute the percentage of which are actually in the error set. We note that this precision metric is upper bounded by the precision of the reference method (confident learning). Again, we find that LEMoN$_{\text{FIX}}$ outperforms the baseline, and is able to identify more actual label errors than CLIP similarity at this threshold.

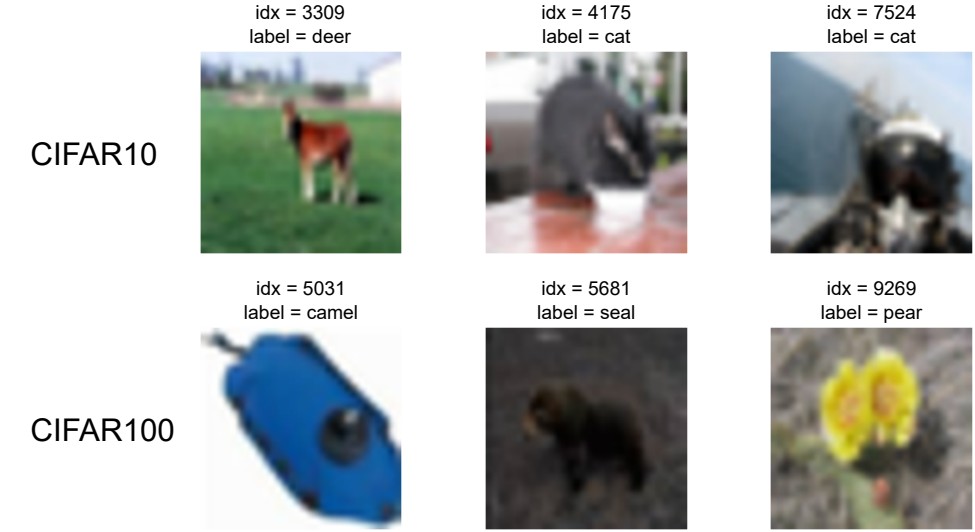

Figure I.6: Demonstrative examples of mislabeled samples in `cifar10` and `cifar100` which have been identified by our method in the top |CL Set|, but was not identified by confident learning in Northcutt et al., 2021 (Northcutt et al., 2021b) and thus was not a part of their error set.

Table I.16: Comparison of LEMoN$_{\text{FIX}}$ (Ours) with the CLIP similarity baseline on the human labeled error set from Northcutt et al., 2021 (Northcutt et al., 2021b). In this prior work, the authors used confident learning to identify |CL Set| candidate label errors in `cifar10` and `cifar100`, |Error Set| of which are confirmed to be mislabels by Mechanical Turkers. Mean Ranking denotes the average ranking of all error set samples as ranked by each method. Precision @ Top |CL Set| involves taking the top |CL Set| samples as ranked by each method, and computing the percentage of which are in the error set. Note that each dataset's test set consists of 10,000 samples. Numbers in parentheses represent one standard deviation.

| Dataset | \|CL Set\| | \|Error Set\| | Mean Ranking | | Precision @ Top \|CL Set\| | | |
|---|---|---|---|---|---|---|---|
| | | | LEMoN$_{\text{FIX}}$ | CLIP Sim. | Oracle | LEMoN$_{\text{FIX}}$ | CLIP Sim. |
| cifar10 | 275 | 54 | 1269.7 (1905.1) | 2681.0 (2507.1) | 19.64% | 6.55% | 1.45% |
| cifar100 | 2235 | 585 | 2357.5 (1981.5) | 3642.1 (2719.5) | 26.17% | 14.41% | 10.16% |

## I.14 DOWNSTREAM CLASSIFICATION WITH LABEL ERROR DETECTION-BASED FILTERING

Here, we show the impact of filtering out different proportions of the training data based on label error predictions, and obtaining test performance.

### I.14.1 AVERAGE ACCURACY

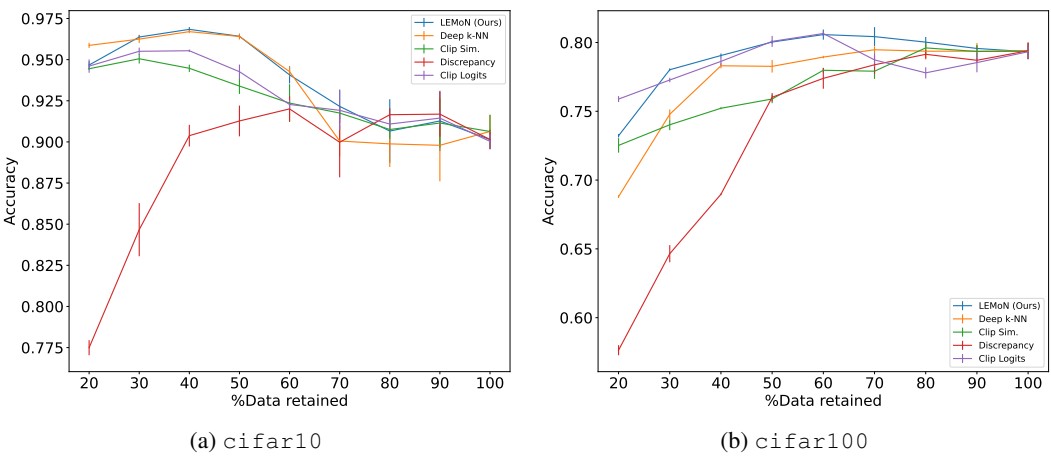

(a) `cifar10`  (b) `cifar100`

Figure I.7: Downstream classification performance.

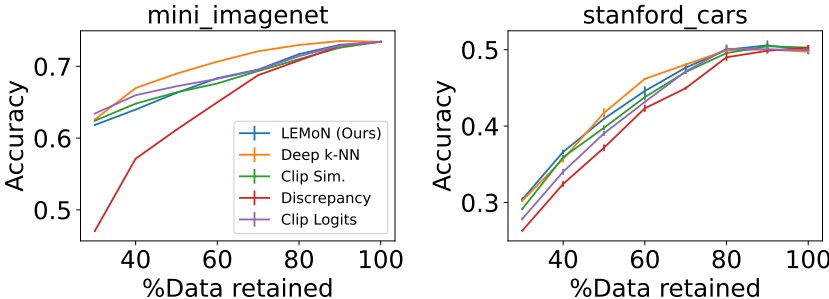

Figure I.8: Downstream accuracy on `stanfordCars`, and `miniImageNet`.

## I.15 AREA UNDER TEST ERROR VS % DATA RETAINED CURVE

We compute the area under the test error (i.e., 1-accuracy) vs % data retained curve in Table I.17. Note that the minimum data retained is 20% (i.e., the minimum amount of data required for training the downstream model).

On both `cifar10` and `cifar100`, we observe that LEMoN performs the best in terms of AUC (i.e., lowest test error). On `stanfordCars` and `miniImageNet`, Deep k-NN performs better. However, the gap in performance is low between LEMON$_{\text{OPT}}$ and the best method (less than 0.9% on `stanfordCars` and 1.2% on `miniImageNet`).

Table I.17: Area under the curve: test error vs % data retained for all four classification datasets. Lower is better, and bold denotes best method.

| Method | cifar10 | cifar100 | stanfordCars | miniImageNet |
|---|---|---|---|---|
| CLIP Sim. | 5.85 | 18.41 | 46.81 | 26.02 |
| CLIP Logits | 5.56 | 17.07 | 47.34 | 25.48 |
| Discrepancy | 8.45 | 20.82 | 48.30 | 30.03 |
| Deep k-NN | 5.34 | 17.74 | **46.19** | **24.69** |
| Ours | **4.98** | **16.60** | 46.29 | 25.95 |

### I.16 OUT-OF-DOMAIN ROBUSTNESS

We report the test performance on an Out-of-Domain (OOD) dataset CIFAR-10C (Hendrycks & Dietterich, 2018), when models are trained on the `cifar10` noisy train set, and validated and tested with clean data with early stopping. The CIFAR-10C dataset contains 19 corruptions applied to the `cifar10` test set, with varying severity of corruption. Then, robustness is measured as the average test top-1 class accuracy performance on the CIFAR-10C dataset (across all corruption types and severities), following prior work (Diffenderfer et al., 2021). We see that: highest robustness is obtained when the proportion of data retained in the train set = 60%, which matches the degree of noise in the dataset. Thus, this implies that filtering out atypical samples using LEMoN increases robustness to image corruptions.

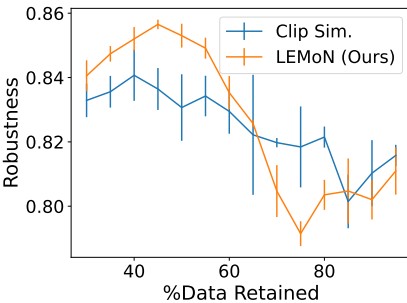

Figure I.9: Downstream accuracy on CIFAR-10C, averaged across all corruption types.

