# OpenReview forum: "LEMoN: Label Error Detection using Multimodal Neighbors"
_ICLR.cc/2025/Conference — Submitted to ICLR 2025_

### Official Review · Reviewer_Ux12 · 2024-10-23

**Soundness:** 2
**Presentation:** 3
**Contribution:** 2
**Rating:** 3
**Confidence:** 5

**Summary:**

The paper presents LEMoN, a novel approach designed to identify inconsistencies in image-caption pairs, with a focus on label noise detection. The proposed method demonstrates improved detection performance in classification and captioning tasks. Additionally, the paper offers theoretical justifications to support the proposed cross-modal scoring method.

**Strengths:**

1. The work introduces an original scoring method for label noise detection.
2. The proposed method is intuitive and clearly explained.
3. The method exhibits notable improvements across most experiments involving synthetic label noise, especially among training-free approaches.

**Weaknesses:**

1. The theoretical justifications provided in the paper contain several significant flaws, limiting their effectiveness as a core claim:
   - Theorem 4.1 appears to contain contradictory conditions, where the variable $\eta$ is defined as normal but also subject to the constraint $|\eta| > \epsilon$.
   - In Proposition A.1, Part 3 (line 885), the inequality does not hold, as the condition on $y' \ne y$ cannot be omitted. This leads to the conclusion $\mathbb{E} \le p\mathbb{E}$ with $p < 1$, implying that $\mathbb{E} = 0$.
   - There is frequent interchange between labels and embeddings, which results in incorrect conclusions. For example, in line 913, the embedding of $y'$ is replaced with label $y$ and an additive term $\eta$. However, according to Assumption 1 (line 855), both $y'$ and $y$ should be labels, not embeddings.
   - The expectation operator is lost when transitioning from the equation in line 915 to the one in line 917.
   - In the proof of Theorem 4.2 (line 954), the variance of $\frac{1}{k} E$ should be expressed as $\frac{1}{k^2} Var E$, rather than $\frac{1}{k} Var E$.
2. While the paper claims novelty in applying multi-modal scoring to label noise detection, this approach has recently been explored in [1].
3. The provided source code does not include implementations of the baseline methods. As a result, it remains unclear how the hyperparameters for these baseline methods were tuned, particularly given that the authors introduced a new set of synthetic datasets.
4. Some previous works [2] evaluate the area under the accuracy/filter-out-rate curve on real datasets, which provides a better understanding of filtering quality in real-world applications. The authors address this metric in Appendix I.12, where the results suggest that Deep k-NN may be a more effective alternative. However, these results are presented for only two datasets, limiting the generalizability of the findings.

[1] Zhu, Zihao, et al. "VDC: Versatile Data Cleanser based on Visual-Linguistic Inconsistency by Multimodal Large Language Models." ICLR 2024.

[2] Bahri, Dara, et al. "Deep k-nn for noisy labels." ICML 2020.

**Questions:**

1. Which implementations of the baseline methods were used, and how were their hyperparameters selected?
2. How does LEMoN compare to VDC [1] in terms of strengths and weaknesses?
3. What is the inference speed of LEMoN relative to other methods? How does it scale with larger datasets, and is it feasible to apply this method to billion-scale datasets?
4. Why does LEMoN not show improvements in terms of the area under the accuracy/filter-out-rate curve?

---

> ### Author Response · Authors · 2024-11-21
>
> Thank you for the detailed review and constructive suggestions!
>
> > W1: The theoretical justifications provided in the paper contain several significant flaws, limiting their effectiveness as a core claim.
>
>
> We appreciate your valuable feedback on our theoretical analysis. We acknowledge that there were flaws in our initial proof. However, the underlying intuition – that contrastive multimodal embedding models can detect noisy labels due to increases in embedding distances caused by label noise – remains valid. We have now revised our theorem and proof (Theorem 4.1 and Appendix A) to correct these errors.
>
>
> **Theorem 4.1** [Contrastive Multimodal Embedding Models Detect Noisy Labels]
> Let $\mathcal{Y} = \mathbb{R}$ and consider a training dataset $\mathcal{D}$. Suppose that $\hat{h}^{\mathcal{X}}\_{\theta}: \mathcal{X} \rightarrow \mathbb{R}^d$ is an embedding function, and $\hat{h}^{\mathcal{Y}}\_{\theta}: \mathcal{Y} \rightarrow \mathbb{R}^d$ is a Lipschitz continuous embedding function with constants $L\_{\mathcal{Y}} > 0$, meaning that for all $y, y' \in \mathcal{Y}$,
> $$ \left\|\left\| \hat{h}^{\mathcal{Y}}\_{\theta}(y) - \hat{h}^{\mathcal{Y}}\_{\theta}(y') \right\|\right\|\_2 \leq L\_{\mathcal{Y}} | y - y' |. $$
> For an input $x \in \mathcal{X}$ and its corresponding positive label $y \in \mathcal{Y}$, let $\eta$ be a random variable drawn from a normal distribution: $ \eta \sim \mathcal{N}(0, \sigma^2). $
> Define a noisy label $y' = y + \eta$. Let $d_{mm}(u, v) = ||u - v||\_2$, which is proportional to $\sqrt{d\_{cos}(u, v)}$ when $||u||\_2 = ||v|\|_2 = 1$. Then, with probability at least $\delta(\epsilon) = 1 - 2 \Phi\left( -\dfrac{\epsilon}{\sigma} \right)$, the following inequality holds:
> $$ d\_{mm}\left( \hat{h}^{\mathcal{X}}\_{\theta}(x), \hat{h}^{\mathcal{Y}}\_{\theta}(y') \right) \geq d\_{mm}\left( \hat{h}^{\mathcal{X}}\_{\theta}(x), \hat{h}^{\mathcal{Y}}\_{\theta}(y) \right) - L\_{\mathcal{Y}} \epsilon , $$
> where $\Phi$ is the cumulative distribution function of the standard normal distribution, and $\epsilon > 0$ is a threshold.
>
> Thus, when $L_{\mathcal{Y}} $ is small, the score for the mislabeled sample cannot be much lower than the score for the positive pair with high probability.
>
> The revised proof can be found in Appendix A.1.
>
>
> > In the proof of Theorem 4.2 (line 954), the variance of 1/k E should be expressed as 1/k² VarE, rather than 1/k VarE.
>
>
> We believe our original proof is actually correct -- there is indeed a $1/k^2$ term, but one of these factors gets canceled out by the summation over k iid random variables. Concretely:
>
> $\\mathbb{E}[Var(S_m(X, Y) | \\zeta_Y)] $
>
> $= \\mathbb{E}[\\frac{1}{k^2}Var\\left(d(X, \\bar{X}_1) + d(X, \\bar{X}\_\{k \\zeta_Y (1-p)\})+  d(X, X'_1) + ... + d(X, X'\_{k - k\\zeta_Y(1 - p)})\\mid \\zeta_Y \\right)]$
>
> $ =  \\mathbb{E}[\\frac{1}{k}(\\zeta_Y (1-p) \\sigma_2^2 + (1 - \\zeta_Y (1-p)) \\sigma_1^2 )] $
>
>
> We have added this intermediate step in the proof in Appendix A.2. Please let us know if this addresses your concerns.

---

> ### Author Response · Authors · 2024-11-21
>
> > W2: While the paper claims novelty in applying multi-modal scoring to label noise detection, this approach has recently been explored in [1].
>
> > Q2: How does LEMoN compare to VDC [1] in terms of strengths and weaknesses?
>
> Thank you for this reference! First, we would like to emphasize that VDC is only evaluated on classification datasets, while our method is able to detect label errors in classification and captioning datasets.
>
> Conceptually, our method is also very different from VDC: VDC relies on an LLM to generate questions about the consistency of a label to a given image, uses a multimodal large language model to automatically generate answers for each image and question, and then evaluates how well the generated caption matches the actual label. Thus, VDC entirely relies on prompting LLMs and VLLMs. In contrast, our method does not utilize any prompt engineering, and instead utilizes the neighborhood information in contrastively trained representations of image and text representations. In addition, our method is more principled in that we have theoretical guarantees for the performance of our multimodal neighborhood score (Theorem 4.2).
>
> Third, our method outperforms SimiFeat-R, a method that VDC performs comparably (Table 4 in VDC paper) by a significant margin (3-6 AUROC points across different noise types; Table 2 and I.2 in our submission).  In principle, the idea of prompting a VLLM is similar to our LLaVa baseline, which LEMoN also far outperforms.
>
> Fourth, our method uses fully open-source models, while VDC relies on ChatGPT based on GPT-3.5-turbo for their question generation and answer evaluation portions, and they do not evaluate any open-source alternatives.
>
> Finally, in order to achieve the prompting abilities required, VDC utilizes models that are much larger than ours. For example, the InstructBLIP used in VDC contains 7B parameters, which is 46x as large as the CLIP ViT-B-32 (151m parameters) used in our paper. This certainly has negative implications for their inference time, which is important in applying such methods to billion-scale datasets as the reviewer pointed out. As VDC also utilizes ChatGPT, this would be extremely costly as well.
>
> We are currently working on adapting VDC to our problem setup of detecting mislabeled image caption pairs, utilizing only open-source models. Results for this baseline will be added in a future revision.
>
>
> > W3: The provided source code does not include implementations of the baseline methods. As a result, it remains unclear how the hyperparameters for these baseline methods were tuned, particularly given that the authors introduced a new set of synthetic datasets.
> > Q1: Which implementations of the baseline methods were used, and how were their hyperparameters selected?
>
> Thanks for raising this point! We highlight that all the hyperparameter settings for the baselines are in Appendix G. Based on the reviewer’s suggestion, we have updated the supplementary code to include baselines, including the hyperparameter grids used for running each baseline in experiments.py. For all baselines, the hyperparameters were selected based on the validation set F1-score, matching the $\text{LEMoN}\_{\text{opt}}$ setting. For Simifeat, we use the open-sourced implementation directly from https://github.com/Docta-ai/docta/tree/master. We have clarified this in Appendix G in the revised paper.

---

> > ### Author Response · Authors · 2024-11-21
> >
> > > W4: Some previous works [2] evaluate the area under the accuracy/filter-out-rate curve on real datasets, which provides a better understanding of filtering quality in real-world applications. The authors address this metric in Appendix I.12, where the results suggest that Deep k-NN may be a more effective alternative. However, these results are presented for only two datasets, limiting the generalizability of the findings.
> >
> > > Q4: Why does LEMoN not show improvements in terms of the area under the accuracy/filter-out-rate curve?
> >
> > Thank you for raising this point! We highlight that we have indeed shown results for all four classification datasets (Figure 3 in main paper and Appendix I.14).
> >
> > We have now computed the AUC using the test error (i.e., 1-accuracy) vs filter-out-rate curves as the reviewer suggested (and similar to deepknn). These AUC scores are shown below. Note that the minimum %data retained is 20% (i.e., the minimum amount of data required for training the downstream model).
> >
> > On both cifar10 and cifar100, we observe that LEMoN performs the best in terms of AUC (i.e., lowest test error). On stanfordCars and miniImagenet, deep kNN performs better. However, the gap in performance is low between LEMoN and the best method (less than 0.9% on stanfordCars and 1.2% on miniImagenet).
> >
> >
> >
> > (Area under test error vs %Data filtered curve, lower is better)
> > | Method      | cifar10 | cifar100 | stanfordCars | miniImagenet |
> > | :---------- | :------ | :------- | :----------- | :----------- |
> > | CLIP Sim.   | 5\.85   | 18\.41   | 46\.81       | 26\.02       |
> > | CLIP Logits | 5\.56   | 17\.07   | 47\.34       | 25\.48       |
> > | Discrepancy | 8\.45   | 20\.82   | 48\.30       | 30\.03       |
> > | Deepknn     | 5\.34   | 17\.74   | **46\.19**       | **24\.69**       |
> > | Ours        | **4\.98**   | **16\.60**   | 46\.29       | 25\.95       |
> >
> > We have added this table and discussion to Appendix I.17.
> >
> >
> >
> >
> > > Q3: What is the inference speed of LEMoN relative to other methods? How does it scale with larger datasets, and is it feasible to apply this method to billion-scale datasets?
> >
> > We have added the per-sample runtime (wall clock time) in ms for LEMoN relative to (1) Clip similarity, (2) Deep k-nn, and (3) Datamap (training-dependent) baselines in Appendix Table I.11. All experiments were conducted using an NVIDIA RTX A6000 GPU and 8 CPU cores. Standard deviation across 3 random data seeds are shown in the parentheses.
> >
> >
> > |           | cifar10     | cifar100    | miniImageNet | stanfordCars | mscoco      | flickr30k   | mimiccxr     | mmimdb      |
> > | :-------- | ----------: | ----------: | -----------: | -----------: | ----------: | ----------: | -----------: | ----------: |
> > | LEMoN     | 10\.1 (0.5) | 9\.6 (0.5)  | 7\.8 (1.6)   | 11\.0 (2.0)  | 18\.8 (1.8) | 35\.9 (1.2) | 52\.2 (2.7)  | 21\.1 (1.4) |
> > | CLIP Sim. | 1\.8 (0.0)  | 1\.8 (0.0)  | 2\.7 (0.4)   | 3\.5 (0.5)   | 20\.3 (0.0) | 15\.6 (0.0) | 16\.8 (0.0)  | 30\.5 (0.0) |
> > | Deep kNN  | 7\.0 (1.3)  | 5\.1 (0.1)  | 8\.7 (1.2)   | 6\.0 (0.1)   | 19\.9 (0.9) | 10\.6 (1.2) | 47\.1 (12.7) | 20\.5 (1.9) |
> > | Datamap   | 37\.6 (0.2) | 37\.5 (0.3) | 37\.7 (1.6)  | 37\.2 (0.3)  | 39\.7 (0.1) | 38\.1 (4.8) | 41\.4 (1.3)  | 62\.6 (9.5) |
> >
> >
> >
> > We observe that LEMoN generally has comparable runtime to deep knn, and significantly lower runtime than Datamap. Note that we use Datamap with LoRA on the captioning datasets, which is why runtime differences between Datamap and LEMoN are lower in these datasets.
> >
> > Finally, we note that LEMoN is embarrassingly parallelizable, and can easily be distributed across multiple processes across multiple servers, whereas trying to distribute training-dependent methods across multiple servers is a bigger challenge.

---

> ### Comment · Reviewer_Ux12 · 2024-11-27
>
> Thank you for addressing the concerns raised earlier. However, there are still several areas that require further clarification and elaboration. Please consider the following questions and comments:
>
> ### W1. **Theory**
> 1. **Line 236 (Theorem 1):** The statement implies that the distance from the incorrect label $y'$ can be equal to the distance from the correct label $y$, even in a carefully designed case. It follows that Lemon may NOT be able to distinguish between correct and incorrect labels.
>
> 2. **Line 839 (Proof of Theorem 2):** In the proof, $p$ is described as a probability, meaning the exact number of relevant neighbors is unknown. Consequently, $S_m$ cannot be directly decomposed into two sums with a fixed number of terms in each.
>
> 3. **Line 844 (Proof of Theorem 2):** The variable over which the expectation is computed appears to be missing. Additionally, why are nested expectations of the form $\mathbb{E}[\mathbb{E}[\dots]]$ necessary in this context?
>
> ### W2. **Novelty**
> - **Line 151:** The claim that "we believe we are the first to apply it to the setting of label error detection" must be revised in light of VDC.
>
> ### W3. **Hyperparameters**
> - The appendix currently includes only the ranges of hyperparameters. For reproducibility and transparency, could you provide the exact hyperparameter values used during the evaluation process?
>
> We appreciate your effort and look forward to seeing the updated evaluation results and responses to these questions.

---

> ### Author Response · Authors · 2024-11-28
>
> Thank you for your detailed response and additional questions!
>
> > W1.1: The statement implies that the distance from the incorrect label $y'$ can be equal to the distance from the correct label $y$.
>
>
> We agree with this characterization, and admit that the lower bound is weak as a result. We stand by our statement that *"when $L_Y$ is small, the score for the mislabeled sample cannot be much lower than the score for the positive pair with high probability."*
>
> Additionally, we emphasize that Theorem 4.1 is a relatively minor contribution of our paper. It largely serves to justify prior works which already utilize the CLIP score for label error detection (Kang et al., 2023; Liang et al., 2023). We believe that Theorem 4.2 sufficiently justifies the main methodological novelties of our work (the scores $s_m$ and $s_n$).
>
>
> > W1.2: In the proof, $p$ is described as a probability, meaning the exact number of relevant neighbors is unknown. Consequently, $S_m$ cannot be directly decomposed into two sums with a fixed number of terms in each.
>
> Thank you for pointing this out! We have now clarified this (in L258) by defining $p$ to be the fraction of mislabeled examples in the nearest neighbors set:
>
> *Suppose that $\frac{1}{k}|\\{i: (X_{m_i}, Y_{m_i}) \text{ is mislabeled}\\}| = p$ is constant for all samples in the support of $(X, Y)$.*
>
> > W1.3: The variable over which the expectation is computed appears to be missing.
>  Additionally, why are nested expectations of the form E[E[…]] necessary in this context?
>
> The nested expectations originate from the law of iterated expectations, i.e. $\mathbb{E}[S_m(X, Y)] = \mathbb{E}[\mathbb{E}[S_m(X, Y) | \zeta_Y]]$. Here, we have used the standard notation that an expectation (without subscript) is computed with respect to the joint distribution of all random variables in the expectation.
>
> Conditioned on $\zeta_Y$, the term $k\zeta_Y(1-p)$ (the exact number of relevant neighbors) is a known constant. There is one implicit assumption that we have made, which is that $\zeta_Y$ is distributed such that the random variable $k\zeta_Y(1-p)$ has support only over the integers $\\{0, 1, ..., k\\}$. We have clarified this assumption in L838.
>
>
>
> > W2: The claim that "we believe we are the first to apply it to the setting of label error detection" must be revised in light of VDC.
>
> To clarify, Line 151 in our paper states:  *"Although prior works have utilized the idea of multimodal neighbors in other settings, we believe we are the first to apply it to the setting of label error detection."* We believe this is still true since VDC does not utilize multimodal neighbors in any form.
>
> We have cited VDC in our updated revision. We are working on adapting VDC to our problem setting now with open-source models, and hope to provide results for it before the end of the rebuttal period.
>
>
>
>
> > W3: The appendix currently includes only the ranges of hyperparameters. For reproducibility and transparency, could you provide the exact hyperparameter values used during the evaluation process?
>
> We have added Tables G.1 and G.2 in the appendix, which lists the optimal hyperparameters for all baselines.
>
>
>
> Please let us know if this sufficiently addresses your concerns. We very much welcome any additional feedback that can further strengthen the paper!

---

> > ### Author Response · Authors · 2024-12-02
> >
> > Dear Reviewer Ux12,
> >
> > We have now implemented VDC for our captioning problem setup and conducted experiments on the four captioning datasets used in our paper. As VDC has only been implemented and evaluated on classification datasets, we make the following adaptations to our problem setup:
> >
> > 1. As we only utilize open-source models in our method, to preserve fairness, we also implement VDC with only open-source models. Specifically, we use Llama-3.1-8B-Instruct for the LLM in the Visual Question Generation (VQG) and Visual Answer Evaluation (VAE) stages (note that the VDC paper uses the OpenAI API), and InstructBLIP-Vicuna-7b as the VLLM in the Visual Question Answering (VQA) stage (as in the VDC paper).
> >
> > 2. In the VQG stage, instead of generating specific questions for each class, we generate six specific questions for each caption. We slightly modify the VQG prompt (Table 8 in the VDC paper) to omit providing the label set, as the set of all possible captions is very large. We keep the two general questions used in the VDC paper.
> >
> >
> > We compare the performance of VDC versus LEMoN below.
> >
> >
> > |            | *flickr30k*     |                 | *mscoco*        |                 | *mmimdb*        |                 | *mimiccxr*      |             |
> > | :--------- | --------------: | --------------: | --------------: | --------------: | --------------: | --------------: | --------------: | ----------: |
> > |            | **AUROC**       | **F1**          | **AUROC**       | **F1**          | **AUROC**       | **F1**          | **AUROC**       | **F1**      |
> > | VDC        | 92\.9 (1.0)     | 85\.5 (0.6)     | 94\.1 (0.2)     | 88\.6 (0.3)     | 80\.5 (0.3)     | 70\.6 (2.2)     | 50\.8 (0.4)     | 29\.3 (1.3) |
> > | $\text{LEMoN}\_{\text{fix}}$ | 93\.6 (0.2)     | -               | 92\.0 (0.1)     | -               | 84\.3 (0.3)     | -               | 66\.5 (0.2)     | -           |
> > | $\text{LEMoN}\_{\text{opt}}$ | **94\.5** (0.2) | **87\.7** (0.9) | **95\.6** (0.2) | **89\.3** (0.2) | **86\.0** (0.1) | **76\.3** (0.1) | **70\.4** (2.3) | **57\.0** (1.6) |
> >
> >
> > We find that $\text{LEMoN}\_{\text{opt}}$ outperforms VDC in all cases. Further, we note that the per-sample runtime (shown below in milliseconds) of VDC is two orders of magnitude larger than all other methods for the same hardware. This is because VDC requires multiple sequential queries to billion-parameter scale LLMs, each of which are roughly 50x the size of the CLIP model used in LEMoN.
> >
> >
> > |           | mscoco      | flickr30k   | mimiccxr     | mmimdb      |
> > | :-------- | ----------: | ----------: | -----------: | ----------: |
> > | LEMoN     | 18\.8 (1.8) | 35\.9 (1.2) | 52\.2 (2.7)  | 21\.1 (1.4) |
> > | CLIP Sim. | 20\.3 (0.0) | 15\.6 (0.0) | 16\.8 (0.0)  | 30\.5 (0.0) |
> > | Deep kNN  | 19\.9 (0.9) | 10\.6 (1.2) | 47\.1 (12.7) | 20\.5 (1.9) |
> > | Datamap   | 39\.7 (0.1) | 38\.1 (4.8) | 41\.4 (1.3)  | 62\.6 (9.5) |
> > | VDC       | 4460 (880)  | 5160 (503)  | 7932 (3\.4)  | 4672 (357)  |
> >
> >
> > As we are unable to update the revision on OpenReview at this time, we will add these results to Table 3 and Table I.11 in a future revision.
> >
> > Please let us know if our responses have sufficiently addressed all of your concerns. We sincerely appreciate all of the detailed feedback you have provided! If there are any remaining questions or comments, we would be happy to discuss.

---

> > > ### Comment · Reviewer_Ux12 · 2024-12-03
> > >
> > > I appreciate the authors' efforts in conducting experiments with VDC and providing both the revised version of the paper and the associated source code. After reviewing both, I have the following observations.
> > >
> > > **Theory.** The theoretical section of the paper has significant issues and should be reconsidered.
> > > - **Theorem 1** claims that "Contrastive Multimodal Embedding Models Detect Noisy Labels," but the conclusion does not support this claim. The distances in the final inequality can remain equal even with multiple strong constraints.
> > > - **Theorem 2** is trivial, as it simply states that random variables with different distributions can be distinguished to some extent.
> > > - **Assumption 2** is not properly validated in Appendix A.3. The appendix presents dataset-level statistics, whereas Assumption 2 pertains to individual image-label pairs.
> > >
> > > **Motivation and Novelty.** The paper lacks a thorough analysis of the distinction between cross-modal retrieval and label noise detection.
> > > - It uses CLIP, a retrieval model, for label noise detection, but similar approaches have been explored in prior work. For instance, label noise detection via retrieval has been discussed in Bahri et al. [1], and multimodal retrieval has been studied in [2,3,4]. This overlap makes the novelty unclear.
> > > - Appendix B argues that second-order captions can be similar to the original ones for misaligned images, but this claim seems counterintuitive. Misaligned images are more likely to produce captions that differ from the original.
> > >
> > > **Baselines.** The paper does not fairly evaluate the proposed method against its closest baseline [3].
> > > - Only the discrepancy (DIS) score for a single modality is implemented, while the original baseline combines both discrepancy (DIS) and divergence (DIV) scores in a single framework (Equations (7) and (8) from the original paper).
> > > - LeMON evaluates scores from both modalities, whereas the baseline is tested only for one. This creates an unfair comparison.
> > > - For a fair evaluation, the baseline must be implemented using combined DIS and DIV scores across both modalities, consistent with how LeMON is evaluated.
> > >
> > > Based on these observations, I will retain my original score.
> > >
> > > [1] Bahri D. et al., "Deep k-nn for noisy labels," ICML 2020
> > >
> > > [2] Rafailidis D. et al., "A unified framework for multimodal retrieval," Pattern Recognition 2013
> > >
> > > [3] Thomas C. et al., "Emphasizing Complementary Samples for Non-literal Cross-modal Retrieval," CVPR 2022
> > >
> > > [4] Yi C. et al., "Leveraging Cross-Modal Neighbor Representation for Improved CLIP Classification," CVPR 2024

---

> ### Author Response · Authors · 2024-12-03
>
> Thank you for your response.
>
> > Theorem 1 claims that "Contrastive Multimodal Embedding Models Detect Noisy Labels," but the conclusion does not support this claim. The distances in the final inequality can remain equal even with multiple strong constraints.
>
> We have already addressed this point in a previous response. Specifically, we admit that this is a weak lower bound, and that:
>
> *We stand by our statement that "when $L_Y$ is small, the score for the mislabeled sample cannot be much lower than the score for the positive pair with high probability."*
>
> *Additionally, we emphasize that Theorem 4.1 is a relatively minor contribution of our paper. It largely serves to justify prior works which already utilize the CLIP score for label error detection (Kang et al., 2023; Liang et al., 2023). We believe that Theorem 4.2 sufficiently justifies the main methodological novelties of our work (the scores $s_m$ and $s_n$).*
>
> As such, the primary contributions of our paper (L87-L97) hold even without this theorem.
>
>
> > Theorem 2 is trivial, as it simply states that random variables with different distributions can be distinguished to some extent.
>
> We completely disagree that a theorem is trivial just because "it simply states that random variables with different distributions can be distinguished to some extent". For example, a large portion of the field of statistical hypothesis testing deals exactly with designing and analyzing methods to distinguish between random variables with different distributions.
>
> Regardless, we note that (a) we derive an analytical solution for exactly the extent to which the distributions can be distinguished, as a function of the parameters of the distributions, and (b) Theorem 2 provides theoretical justification for our proposed method (the scores $s_m$ and $s_n$). For these reasons, we do not believe this theorem is "trivial", and we believe it is unfair to dismiss it as so.
>
>
> > Assumption 2 is not properly validated in Appendix A.3. The appendix presents dataset-level statistics, whereas Assumption 2 pertains to individual image-label pairs.
>
>
> This is simply incorrect. First, Assumption 2 pertains to the *distribution* of distances between images and their "paraphrases", not individual image-label pairs. Second, this distribution is precisely what we show in Appendix A.3 (Figure A.1), and we even conduct a test for normality, showing that this assumption does indeed hold.
>
>
> > The paper lacks a thorough analysis of the distinction between cross-modal retrieval and label noise detection.
>
> > It uses CLIP, a retrieval model, for label noise detection, but similar approaches have been explored in prior work. For instance, label noise detection via retrieval has been discussed in Bahri et al. [1], and multimodal retrieval has been studied in [2,3,4]. This overlap makes the novelty unclear.
>
> We respectfully disagree with the reviewer’s point that “similar approaches have been explored in prior work" for label noise detection. As we stated in the prior response (L151): *"Although prior works have utilized the idea of multimodal neighbors in other settings, we believe we are the first to apply it to the setting of label error detection."* Our novelty in methodology does not lie in using CLIP, but rather proposing *a novel multimodal neighborhood-based score for the task of label noise detection*. As the reviewer themselves notes in their original review, we propose an “original scoring method for label noise detection”.
>
> Furthermore, we have *already compared* against the baselines the reviewer specified [1,3], and have shown that LEMoN outperforms them: deep kNN [1] in Table 2 and Table 3, and Discrepancy/Diversity [3] in Table 2 and Table I.12. The other references [2,4] that the reviewer has identified do not propose a score that can be used for label noise detection, but focus on improving multimodal matching and retrieval. Thus, they are not directly applicable to this setting. We will note this in our revised paper.
>
> Lastly, the only connection between our work and cross-modal retrieval [2,3,4] is that these methods are multimodal [2,3,4], and may use neighborhood-based strategies [1,4]. Importantly, we *already cite and describe how our works differ from prior multimodal neighborhood-based works in L142-L152 and Appendix B* (which the reviewer already references). Additionally, works in cross-modal retrieval [2,4] are orthogonal and potentially complementary to our work: better multimodal retrieval could lead to more accurate construction of multimodal neighborhoods, and thus better scores for label noise detection under our framework. Thus, we strongly disagree with the reviewer’s point that our work “lacks distinction between cross-modal retrieval and label noise detection”.

---

> > ### Author Response · Authors · 2024-12-03
> >
> > > Appendix B argues that second-order captions can be similar to the original ones for misaligned images, but this claim seems counterintuitive. Misaligned images are more likely to produce captions that differ from the original.
> >
> >
> > For context, we assume the reviewer is referring to our discussion of the $\Upsilon\_Y^{DIS}$ term (L903-L908). We note that the point of contention is not whether *"Misaligned images are more likely to produce captions that differ from the original."*, but whether the distance between *second-order neighbors in text space and the original caption* is larger for mislabeled samples. Note that the authors of [3] "compute neighbors in text space because the text domain provides the cleanest semantic representation of the image-text pair" [3 Section 3.2]. As $\Upsilon\_Y^{DIS}$ does not utilize the image at all in its computation, it cannot possibly give any signal as to whether a particular (image, text) pair is mislabeled.
> >
> > We have provided an intuitive argument of this in L903-L908, and have demonstrated it empirically across eight datasets in Table I.12, where $\Upsilon\_Y^{DIS}$ achieves chance performance at label error detection (average AUROC of 49.5). We believe this sufficiently addresses this point.

---

> ### Author Response · Authors · 2024-12-03
>
> > The paper does not fairly evaluate the proposed method against its closest baseline [3].
>
> > For a fair evaluation, the baseline must be implemented using combined DIS and DIV scores across both modalities, consistent with how LeMON is evaluated.
>
> As we have argued both intuitively (in Appendix B) and empirically (in Table I.12), $\Upsilon\_X^{DIS}$ is the only score from [3] that contributes any signal to label error detection. In fact, the average AUROC (across eight datasets) of the remaining three terms $\Upsilon\_Y^{DIS}$, $\Upsilon\_X^{DIV}$, $\Upsilon\_Y^{DIV}$ are, respectively: 49.5 (1.7), 51.8 (5.0), and 49.3 (2.1), where parentheses show standard deviation across the eight datasets. Clearly, these remaining three terms are no better than random chance at detecting label errors.
>
> Regardless, we have implemented the ensembling of the four scores as the reviewer suggested. For the $\text{Comb-Val}$ strategy, as there are four terms, we sweep over weights in $\\{1, 2, 3, 4, 5\\}^4$, following [3], selecting the best combination using a labeled validation set, identically to LEMoN. For the $\text{Comb-Stat}$ strategy, we use the mean and standard deviations, as in Equation (8) in [3]. We find that none of the combined scores significantly outperform $\Upsilon\_X^{DIS}$. This is because in both combination strategies, a non-zero weight is placed on the other terms, which essentially adds noise to the final score without contributing any signal.
>
>
> |              | **AUROC**            |                      |                      |                      |             |             |                 | F1                   |                      |                      |                      |             |             |                 |
> | :----------- | -------------------: | -------------------: | -------------------: | -------------------: | ----------: | ----------: | --------------: | -------------------: | :------------------- | :------------------- | :------------------- | ----------: | ----------: | :-------------- |
> |              | $\Upsilon_{X}^{DIS}$ | $\Upsilon_{Y}^{DIS}$ | $\Upsilon_{X}^{DIV}$ | $\Upsilon_{Y}^{DIS}$ | $\text{Comb-Val}$      |   $\text{Comb-Stat}$        |  $\text{LEMoN}\_{\text{opt}}$   | $\Upsilon_{X}^{DIS}$ | $\Upsilon_{Y}^{DIS}$ | $\Upsilon_{X}^{DIV}$ | $\Upsilon_{Y}^{DIS}$ | $\text{Comb-Val}$      |   $\text{Comb-Stat}$        |  $\text{LEMoN}\_{\text{opt}}$    |
> | cifar10     | 77\.1 (1.9)          | 48\.2 (1.2)          | 50\.3 (1.9)          | 45\.0 (1.9)          | 77\.1 (1.9) | 77\.1 (1.9) | **98\.1** (0.0) | 68\.2 (1.9)          | 29\.2 (0.4)          | 29\.2 (0.4)          | 29\.2 (0.4)          | 68\.2 (1.9) | 68\.2 (1.9) | **93\.1** (0.2) |
> | cifar100     | 66\.0 (1.5)          | 49\.4 (1.1)          | 49\.9 (1.4)          | 49\.7 (1.9)          | 65\.9 (1.5) | 65\.9 (1.5) | **90\.8** (0.0) | 51\.9 (1.8)          | 29\.4 (1.4)          | 32\.5 (5.5)          | 29\.4 (0.4)          | 51\.9 (1.8) | 51\.9 (1.8) | **81\.3** (0.2) |
> | miniImageNet | 79\.4 (0.3)          | 47\.4 (0.5)          | 64\.6 (0.2)          | 48\.0 (0.5)          | 75\.3 (0.3) | 76\.6 (0.3) | **90\.2** (0.2) | 69\.8 (0.4)          | 28\.0 (2.3)          | 55\.8 (2.3)          | 27\.0 (0.9)          | 63\.7 (0.4) | 65\.5 (0.4) | **82\.3** (0.1) |
> | stanfordCars | 65\.7 (0.7)          | 50\.8 (1.1)          | 51\.9 (0.9)          | 50\.1 (0.5)          | 61\.2 (0.8) | 63\.4 (0.9) | **73\.1** (0.5) | 59\.9 (0.4)          | 20\.6 (1.3)          | 25\.3 (5.6)          | 20\.6 (1.4)          | 48\.9 (0.4) | 53\.1 (0.4) | **67\.3** (1.0) |
> | flickr30k    | 73\.0 (0.6)          | 53\.3 (1.4)          | 49\.9 (2.9)          | 52\.9 (0.2)          | 60\.6 (0.6) | 64\.4 (0.6) | **94\.5** (0.2) | 64\.7 (1.7)          | 26\.2 (0.8)          | 27\.4 (1.7)          | 26\.1 (1.0)          | 48\.7 (1.7) | 54\.9 (1.7) | **87\.7** (0.9) |
> | mimiccxr     | 60\.0 (0.8)          | 49\.6 (0.4)          | 50\.0 (1.3)          | 49\.1 (1.3)          | 55\.7 (0.8) | 57\.9 (0.8) | **70\.4** (2.3) | 32\.8 (2.8)          | 28\.5 (0.0)          | 28\.5 (0.0)          | 28\.5 (0.0)          | 30\.5 (2.8) | 31\.4 (2.8) | **57\.0** (1.6) |
> | mmimdb       | 57\.4 (0.4)          | 49\.8 (0.4)          | 48\.6 (0.4)          | 50\.0 (0.5)          | 52\.6 (0.4) | 54\.6 (0.4) | **86\.0** (0.1) | 40\.2 (1.7)          | 28\.6 (0.1)          | 29\.1 (0.5)          | 28\.9 (0.6)          | 29\.6 (1.7) | 30\.4 (1.7) | **76\.3** (0.1) |
> | mscoco       | 72\.7 (0.3)          | 48\.5 (0.8)          | 52\.9 (0.8)          | 48\.7 (0.3)          | 59\.8 (0.3) | 65\.8 (0.3) | **95\.6** (0.2) | 67\.3 (0.9)          | 29\.7 (0.1)          | 29\.0 (0.2)          | 28\.9 (0.4)          | 37\.4 (0.9) | 58\.1 (0.9) | **89\.3** (0.2) |
>
>
> We will add this result to Table I.12 in the revision.

---

> > ### Author Response · Authors · 2024-12-03
> >
> > We believe the only remaining concern is regarding the weakness of the bound in Theorem 1. We largely agree with you on this point, but highlight that this theorem is a relatively minor contribution and a supplementary result of our paper, which largely serves to justify prior works which already utilize the CLIP score for label error detection. As such, we believe this is not sufficient reason to warrant such a negative assessment on its own. We would be open to moving this theorem to the Appendix if suggested so by the reviewer. We emphasize that our key contributions are independent of this theorem.
> >
> > Given this and the factual errors in your response, we kindly ask that you reconsider your assessment of the paper. We thank you again for your feedback, and your active participation during the rebuttal period.

---

### Official Review · Reviewer_hPEV · 2024-10-24

**Soundness:** 3
**Presentation:** 3
**Contribution:** 2
**Rating:** 6
**Confidence:** 4

**Summary:**

This paper presents an approach to detect misalignment between image-text pairs to clean image-text datasets. To achieve it, they propose a new metric that considers the distance between the image-text pairs and neighbors in image and text space.

The high-level idea of their metric to compute label-error score is as follows.
1. Given an image-text pair that is neighboring a target pair, if the text is far from the target text, but the image is close to the target image, the target pairs can be inconsistent.
2. The neighboring pair used above can be also mismatched. To account for such a case, they weigh the score by using the similarity between the neighboring pair.

Empirically, they evaluate the effectiveness of the proposed metric by image-classification dataset, an image-caption dataset such as COCO, and a medical image-report dataset. For evaluation in the image-text dataset, they randomly inject noise into the supervision of the dataset, e.g., replacing object names. Overall, their metric seems to be better than existing metrics in detecting noises, but the improvement was marginal in image-captioning.

**Strengths:**

1. They provide a new metric to detect label noise in image-text data, which is reasonable. Also, their experiments verify that the proposed metric outperforms an existing metric in detecting errors in their settings.
2. Their writing and presentations are clear, and mostly easy to follow.
3. Their approach includes some hyper-parameters, but the robustness to such parameters is also investigated.
4. They conduct a wide rage of experiments which can be insightful for readers.

**Weaknesses:**

I am concerned that the experiments are not so focused on noisy image-caption datasets although their motivation is to handle issues of noisy data collected from the web.

1. Their approach seems to be mainly designed to detect label errors in image-caption datasets. However, the effectiveness of applying filtering to such a dataset seems to be marginal according to Table 4. I think label-error identification is proven to be effective by improving performance on downstream tasks. In this sense, the effectiveness of the proposed approach is not proven enough.

2. They conduct experiments on COCO and Flickr to show the effectiveness of the image-caption dataset. Then, the effectiveness of their metric is verified only on the synthetic noise they created. However, there can be more diverse types of noise in real image-caption data collected from the web. For example, some captions might focus on a specific aspect of the image while others have details. According to the experiments, it is not clear how their metric behaves in such cases.

3. Also, according to their appendix, their metric does not show much gain in the CC3M, which is webly collected. I actually feel that the authors had to conduct more analysis on this kind of dataset since their main motivation seems to detect errors on this kind of dataset, rather than image-classification dataset.

**Questions:**

1. It took some time to understand the intuition behind Eq. 2. I think it is better to provide high-level ideas of what Eq. 2 is computing.

---

> ### Author Response · Authors · 2024-11-21
>
> Thank you for the detailed review and constructive suggestions!
>
> > W1: Their approach seems to be mainly designed to detect label errors in image-caption datasets. However, the effectiveness of applying filtering to such a dataset seems to be marginal according to Table 4.
>
>
> We highlight that in Table 4, even training with a fully clean dataset only outperforms training with a fully noisy (i.e. no filtering) dataset by 2-3 BLEU-4 points. Thus, the range of potential improvement for any filtering method is bounded by this difference. Adding on the variance associated with model training (standard deviations up to 1.0), it is difficult for any method to outperform another with significance. This is an interesting result, since it means that some pre-trained captioning models are stable or have small-sized performance drops in the presence of noisy captioning data. We interpret the result from Table 4 to be that both LEMoN and CLIP Sim. are nearly able to recover the performance of clean data for both datasets. In addition, we note that LEMoN performs slightly better in mscoco, partly because its larger dataset size results in a smaller variance.
>
> In addition, we highlight that in the classification setting, LEMoN is consistently among the top-2 methods across all four datasets in downstream performance. Lastly, in addition to filtering out incorrect data, such label error detection methods are also useful in and of itself. For example, identifying mislabeled samples (and potentially annotators responsible for these mislabels) can lead to improvements in labeling processes. Thus, we strongly believe that label error detection using LEMoN has real-world utility. We will also demonstrate the utility of LEMoN to effectively filter data for downstream training in Datacomp below.
>
> > W2: However, there can be more diverse types of noise in real image-caption data collected from the web.
>
> > W3: Also, according to their appendix, their metric does not show much gain in the CC3M, which is webly collected. I actually feel that the authors had to conduct more analysis on this kind of dataset since their main motivation seems to detect errors on this kind of dataset, rather than image-classification dataset.
>
> Thank you for this suggestion. We have conducted a new experiment of $\text{LEMoN}\_{\text{fix}}$ on Datacomp [1]. We use the small dataset from the filtering track, which originally consisted of 12.8M images. As these images are accessed directly from the web, only 9.96M images were able to be downloaded as of 2024/11/14. We apply $\text{LEMoN}\_{\text{fix}}$  to this dataset using OpenAI CLIP ViT-L/14 embeddings provided by Datacomp. We select the 3.5M images with lowest mislabel scores, and use the default hyperparameters from Datacomp to train a CLIP model, and evaluate it on the same 38 zero-shot classification datasets. We compare with filtering using only the CLIP score to the same number of images.
>
> |                                            | Method                   | ImageNet   | ImageNet Dist. Shifts | VTAB       | Retrieval  |  Avg (38 Datasets) |
> | :----------------------------------------- | :----------------------- | ---------: | --------------------------: | ---------: | ---------:  |  -------------------------:  |
> | Data Currently Available  (9\.96M Samples)      | LEMoN                    | **0\.045** | **0\.053**            | **0\.188** | 0\.116     | **0\.168**        |
> |                                            | CLIP score               | 0\.043     | 0\.049                | 0\.177     | **0\.119** | 0\.160            |
> | From Datacomp Paper  (12\.8M Samples) | No filtering             | 0\.025     | 0\.033                | 0\.145     | 0\.114     | 0\.132            |
> |                                            | Basic filtering          | 0\.038     | 0\.043                | 0\.150     | 0\.118     | 0\.142            |
> |                                            | Text-based               | 0\.046     | 0\.052                | 0\.169     | **0\.125** | 0\.157            |
> |                                            | Image-based              | 0\.043     | 0\.047                | 0\.178     | 0\.121     | 0\.159            |
> |                                            | LAION-2B filtering       | 0\.031     | 0\.040                | 0\.136     | 0\.092     | 0\.133            |
> |                                            | CLIP score               | **0\.051** | **0\.055**            | **0\.190** | 0\.119     | **0\.173**        |
> |                                            | Image-based + CLIP score | 0\.039     | 0\.045                | 0\.162     | 0\.094     | 0\.144            |
>
>
> We find that given the available images, LEMoN outperforms the baseline on average, and on three of four individual evaluations. However, neither method outperforms the scores reported in the original paper due to their dataset being larger.
>
> We have added this table and discussion to Appendix I.10.

---

> > ### Author Response · Authors · 2024-11-21
> >
> > > Q1: It took some time to understand the intuition behind Eq. 2. I think it is better to provide high-level ideas of what Eq. 2 is computing.
> >
> >
> > Thank you for this suggestion. We have clarified the following plain text description of our method in Appendix C, and added a pointer to it from Section 3.
> >
> >
> > _For each image-caption pair in the dataset, we first compute how similar the image and caption are to each other using a pre-trained CLIP model ($d\_{mm}$), which gives a basic measure of how well they match. To compute $s\_m$, we compute the nearest neighbors of the caption among other captions in the dataset. For each neighbor, we look at how similar their corresponding image is to the original image. The intuition is that if a sample is correctly labeled, the image should be similar to images of other samples with similar captions. We weight each neighbor based on how close it is to our original sample and how well-matched the neighboring pairs themselves are. Finally, we repeat this for nearest neighbors in the image space to get $s\_n$. LEMoN is then the weighted sum of these three scores._

---

> > ### Comment · Reviewer_hPEV · 2024-11-27
> > **response**
> >
> > Thanks for your response.
> > From their new results, I understand that their proposed approach is somewhat effective in filtering noisy image-caption data.
> >
> > I would like to raise my rating to 6 since I still do not think this submission is must-accept one.

---

### Official Review · Reviewer_psZs · 2024-10-31

**Soundness:** 3
**Presentation:** 3
**Contribution:** 2
**Rating:** 6
**Confidence:** 3

**Summary:**

This paper proposed a new way to filter noisy multimodal data. Besides of using image-caption embedding similarity, the new approach leverages the multimodal neighborhood of image-captions pairs to identify label error. The method demonstrates improvements in label error detection and enhances performance on downstream captioning tasks.

**Strengths:**

This paper introduces a novel approach that uses multimodal nearest neighbors to assess the relevance between images and captions, providing both theoretical justification and empirical validation for the proposed method. The experiments evaluate its effectiveness in detecting label errors and its impact on downstream classification and captioning models.

**Weaknesses:**

The current experiments lack the breadth needed to fully demonstrate the impact of adding nearest neighbor terms. It would be beneficial to include a comparison using only single-side nearest neighbor term, and to present the actual values of all three terms for clearer insight.

**Questions:**

1. In section 3, a few symbols are not explained, like r, D, k etc. It's better to split the section 3 into multiple sub sections to explain each term separately
2. Is there any explanation that why using pure CLIP similarity performs better than LEMoN on Flickr30k?

---

> ### Author Response · Authors · 2024-11-21
>
> Thank you for the detailed review and constructive suggestions!
>
> > W1: The current experiments lack the breadth needed to fully demonstrate the impact of adding nearest neighbor terms. It would be beneficial to include a comparison using only single-side nearest neighbor term, and to present the actual values of all three terms for clearer insight.
>
> Thank you for the suggestion. We have conducted a full ablation study of the performance of each of the three terms in LEMoN, for all datasets. The following table shows the AUROC of each score and all possible combinations:
>
>
> |                            | cifar10         | cifar100        | miniImageNet    | stanfordCars    | flickr30k       | mscoco          | mmimdb          | mimiccxr        |
> | :------------------------- | --------------: | --------------: | --------------: | --------------: | --------------: | --------------: | --------------: | --------------: |
> | $d\_{mm}$ (CLIP Sim.)      | 93\.8 (0.1)     | 78\.5 (0.6)     | 89\.3 (0.2)     | 69\.8 (0.6)     | 94\.8 (0.5)     | 93\.8 (0.2)     | 85\.1 (0.3)     | 64\.1 (0.4)     |
> | $s\_m$                   | 79\.3 (2.8)     | 65\.4 (2.0)     | 80\.8 (0.3)     | 66\.0 (0.9)     | 76\.3 (1.8)     | 75\.8 (0.3)     | 60\.1 (0.4)     | 59\.0 (0.6)     |
> | $s\_n$                   | 98\.1 (0.0)     | 88\.4 (0.1)     | 84\.3 (0.2)     | 72\.8 (0.7)     | 71\.4 (1.6)     | 76\.5 (0.5)     | 55\.1 (0.3)     | 57\.9 (2.1)     |
> | $d\_mm + s\_m$           | 92\.5 (0.5)     | 81\.3 (1.1)     | 89\.6 (0.2)     | 69\.7 (0.5)     | **95\.0** (0.5) | 94\.6 (0.3)     | **86\.0** (0.4) | 64\.5 (0.6)     |
> | $s\_n + s\_m$            | 98\.0 (0.2)     | 88\.8 (0.2)     | 84\.5 (0.4)     | 72\.8 (0.7)     | 83\.5 (0.5)     | 86\.1 (0.6)     | 67\.6 (0.9)     | 63\.6 (0.6)     |
> | $d\_mm + s\_n$          | **98\.2** (0.1) | **90\.8** (0.1) | 89\.9 (0.3)     | **73\.9** (0.7) | 94\.9 (0.3)     | 94\.9 (0.2)     | 85\.3 (0.3)     | 66\.4 (2.4)     |
> | $d\_{mm} + s\_n + s\_m$ (LEMoN) | 98\.1 (0.0)     | **90\.8** (0.0) | **90\.2** (0.2) | 73\.1 (0.5)     | 94\.5 (0.2)     | **95\.6** (0.2) | **86\.0** (0.1) | **70\.4** (2.3) |
>
>
>
> We find that $d_{mm}$ is the most critical term. Of the two nearest neighbors terms, we find that $s_n$ (nearest image neighbors) is more important in general, though this is highly dataset dependent, e.g. error detection in mmimdb relies much more on neighbors in the text space than the image space, while the opposite is true for mscoco.
>
> We have added this table to the appendices, and discussed it briefly in Section 6.3.
>
>
> > Q1: In section 3, a few symbols are not explained, like r, D, k etc. It's better to split the section 3 into multiple sub sections to explain each term separately
>
> Thank you for pointing this out. The symbol $\mathcal{D}$ was defined at the start of Section 3, and we have clarified the notation in Section 3 by adding several explanations:
>
> - Define $B(\mathbf{x}, r) := \\{x' \in \mathcal{X}: d_{\mathcal{X}}(\mathbf{x}, \mathbf{x}') \leq r \\}$, the ball of radius $r$ around $\mathbf{x}$, and $B(\mathbf{y}, r)$ similarly.
>
> - Let $r_k(\mathbf{x}) := \inf\\{ r: | B(\mathbf{x}, r) \cap \mathcal{D}| \geq k \\} $, the minimum radius required to encompass at least $k$ neighbors.
>
>
> Further, we have added Table C.1 in Appendix C, which provides a summary of all the notation used in this section. Please let us know if there are any remaining issues!
>
>
>
> > Q2: Is there any explanation that why using pure CLIP similarity performs better than LEMoN on Flickr30k?
>
> We note that due to the large error bars, CLIP similarity does not actually outperform LEMoN with statistical significance on flickr30k (p=0.63 for AUROC and p=0.20 for F1 from paired t-tests). Since LEMoN is a generalization of CLIP similarity, we would expect $\text{LEMoN}\_{\text{opt}}$ to outperform CLIP similarity when sufficient validation data is available. As the dataset size for flickr30k is small, this results in (1) larger variance in test-set metrics as seen in Table 3, and (2) hyperparameters of $\text{LEMoN}\_{\text{opt}}$ overfitting to the smaller validation set.  We explore the second phenomenon further in Appendix I.7., where we vary the validation set size.

---

> > ### Author Response · Authors · 2024-11-30
> > **Have we addressed your concerns?**
> >
> > Dear Reviewer psZs,
> >
> > Thank you again for your time and valuable feedback. Since there are a few days left in the rebuttal period, we were wondering if our response has adequately addressed your concerns. If so, we would appreciate it if you could update your review and raise your score accordingly. If there are any remaining questions or comments, we would be happy to discuss.
> >
> > Thank you!

---

> > > ### Comment · Reviewer_psZs · 2024-11-30
> > >
> > > Thank you for the detailed response and clarification. Most of my questions are now resolved, and I’ve updated my score accordingly.

---

### Official Review · Reviewer_uuuA · 2024-11-04

**Soundness:** 3
**Presentation:** 3
**Contribution:** 2
**Rating:** 6
**Confidence:** 3

**Summary:**

This paper proposes a label error detection method for multimodal datasets. Specifically, the authors first use pre-trained vision-language models to extract image-caption embeddings. Then they leverage the distance of multi-modal neighborhoods to detect the label error of image-caption datasets. This paper also provides a theoretical analysis of the feasibility of the proposed method.

**Strengths:**

1. The research problem is important. The image-caption datasets are widely used to train multimodal models. Detecting the label errors in these datasets is important for downstream tasks.
2. The idea of using pre-trained multimodal models is well-motivated and the theoretical analyses are reasonable.
3. This paper is very well organized and written in general.

**Weaknesses:**

1. The details of the application on the unimodal dataset need to be clarified. How to define the nearest neighbors of text in unimodal datasets like CIFAR10/100？
2. Figure 3 can be improved. These lines overlap too much and are difficult to distinguish.
3. In the downstream captioning task, the improvements over CLIP similarity seem trivial. Can the authors provide some analysis of the possible reason？

**Questions:**

see the weaknesses

---

> ### Author Response · Authors · 2024-11-21
>
> Thank you for the detailed review and constructive suggestions!
>
> > W1: The details of the application on the unimodal dataset need to be clarified. How to define the nearest neighbors of text in unimodal datasets like CIFAR10/100？
>
> Similar to the original CLIP paper [2], to generate the text modality for these datasets, we use the name or description associated with each particular label. For example, class 0 in cifar10 is "airplane", and this is the caption that we associate with all images of that class. The representation corresponding to this text modality is then used to define text-based neighbors.
> We have clarified this further in Appendix D.1. in our revised paper.
>
>
> > W2: Figure 3 can be improved. These lines overlap too much and are difficult to distinguish.
>
> Thank you – the small size of the image was due to the lack of space. We have now updated the figures in the Appendix I.13 with an enlarged scale. We have also computed the area-under-the test-error vs %data retained curves for each of these figures as Reviewer Ux12 suggested in Appendix Table I.17., where we find that LEMoN has better AUC than all other baselines on cifar10 and cifar100 (i.e., the plots in Figure 3).

---

> > ### Author Response · Authors · 2024-11-21
> >
> > > W3: In the downstream captioning task, the improvements over CLIP similarity seem trivial. Can the authors provide some analysis of the possible reason？
> >
> > This is a good point. We highlight that in Table 4, even training with a fully clean dataset only outperforms training with a fully noisy (i.e. no filtering) dataset by 2-3 BLEU-4 points. Thus, the range of potential improvement for any filtering method is bounded by this difference. Adding on the variance associated with model training (standard deviations up to 1.0), it is difficult for any method to outperform another by a large margin. This is an interesting result, since it means that some pre-trained captioning models are stable or have small-sized performance drops in the presence of noisy captioning data. We interpret the result from Table 4 to be that both LEMoN and CLIP Sim. are nearly able to recover the performance of clean data for both datasets. In addition, we note that LEMoN performs slightly better in mscoco, partly because its larger dataset size results in a smaller variance.
> >
> > Further, we have conducted an additional experiment of $\text{LEMoN}\_{\text{fix}}$ on Datacomp [1]. We use the small dataset from the filtering track, which originally consisted of 12.8M images. As these images are accessed directly from the web, only 9.96M images were able to be downloaded as of 2024/11/14. We apply $\text{LEMoN}\_{\text{fix}}$  to this dataset using OpenAI CLIP ViT-L/14 embeddings provided by Datacomp. We select the 3.5M images with lowest mislabel scores, and use the default hyperparameters from Datacomp to train a CLIP model, and evaluate it on the same 38 zero-shot classification datasets. We compare with filtering using only the CLIP score to the same number of images.
> >
> > |                                            | Method                   | ImageNet   | ImageNet Dist. Shifts | VTAB       | Retrieval  |  Avg (38 Datasets) |
> > | :----------------------------------------- | :----------------------- | ---------: | --------------------------: | ---------: | ---------:  |  -------------------------:  |
> > | Data Currently Available  (9\.96M Samples)      | LEMoN                    | **0\.045** | **0\.053**            | **0\.188** | 0\.116     | **0\.168**        |
> > |                                            | CLIP score               | 0\.043     | 0\.049                | 0\.177     | **0\.119** | 0\.160            |
> > | From Datacomp Paper  (12\.8M Samples) | No filtering             | 0\.025     | 0\.033                | 0\.145     | 0\.114     | 0\.132            |
> > |                                            | Basic filtering          | 0\.038     | 0\.043                | 0\.150     | 0\.118     | 0\.142            |
> > |                                            | Text-based               | 0\.046     | 0\.052                | 0\.169     | **0\.125** | 0\.157            |
> > |                                            | Image-based              | 0\.043     | 0\.047                | 0\.178     | 0\.121     | 0\.159            |
> > |                                            | LAION-2B filtering       | 0\.031     | 0\.040                | 0\.136     | 0\.092     | 0\.133            |
> > |                                            | CLIP score               | **0\.051** | **0\.055**            | **0\.190** | 0\.119     | **0\.173**        |
> > |                                            | Image-based + CLIP score | 0\.039     | 0\.045                | 0\.162     | 0\.094     | 0\.144            |
> >
> >
> > We find that given the available images, LEMoN outperforms the baseline on average, and on three of four individual evaluations. However, neither method outperforms the scores reported in the original paper due to their dataset being larger.
> >
> > We have added this table and discussion to Appendix I.10.
> >
> > [2] Learning Transferable Visual Models From Natural Language Supervision. ICML 2021.

---

> > > ### Comment · Reviewer_uuuA · 2024-11-25
> > >
> > > Thanks to the authors for their response and further experiments. They address my concerns. Overall, this paper is above the acceptance threshold, I will maintain my rating.

---

### Author Response · Authors · 2024-11-21

We thank all reviewers for their time and valuable feedback! We are glad that reviewers found the research problem to be "important" (uuuA), the method to be "well-motivated" (uuuA), "intuitive and clearly explained" (Ux12), with "both theoretical justification and empirical validation" (psZs), and the writing "clear and mostly easy to follow" (hPEV). Based on their comments and suggestions, we have made several improvements to the paper. We highlight the major additions here, and provide detailed responses to each reviewer below.

1. **Experiments on Datacomp.** To address a concern from Reviewer hPEV, we have run our method along with a baseline on the Datacomp filtering track small subset, which consists of 12.8M images from the web. We find that training CLIP models with LEMoN-filtered data outperforms the CLIP similarity baseline on downstream zero-shot classification tasks.

2. **Extended Ablations.** Following a suggestion from Reviewer psZs, we have run an extended ablation study measuring the performance of each of the three terms within LEMoN and all of their possible combinations, for all datasets.  We find that $d_{mm}$ is the most critical term. Of the two nearest neighbors terms, we find that $s_n$ (nearest image neighbors) is more important for most datasets.

3. **Writing and Theory Clarifications.** We have improved the clarity of the writing in response to questions raised by the reviewers. We have also corrected the proof to Theorem 4.1 following flaws astutely pointed out by Reviewer Ux12.


All of these changes have also been added to the updated revision. Please feel free to follow up with us if you have additional feedback, questions, or concerns. We very much welcome any feedback that can further strengthen the paper. Thank you again to all reviewers.


[1] Datacomp: In search of the next generation of multimodal datasets. NeurIPS 2024.

---

### Author Response · Authors · 2024-11-24
**Have we addressed your concerns?**

Dear Reviewers,

Thank you again for your thorough reviews and constructive feedback. With the limited time remaining in the rebuttal phase, we were wondering if our responses have adequately addressed your concerns. If so, we would appreciate it if you could update your review and your score accordingly. If there are any remaining questions or comments, we would be happy to discuss.

Thank you!

---

### Meta-Review · Area_Chair_LMot · 2024-12-12

**Metareview:**

The paper proposes a mechanism for coping with label/caption noise. The reviewers praise the intuitive method and extensive experiments. However they raise concerns about effectiveness (e.g. improvement over using CLIP, results in Table 4), applicability (to real-world noise) and novelty (e.g. wrt to an ICLR 2024 paper). No reviewer score exceeds 6 and some 6-scoring reviewers voice numerous substantial concerns (even after the rebuttal).

**Additional Comments On Reviewer Discussion:**

All reviewers participated in the discussion and responded to the rebuttal. Some Reviewer Ux12 concerns were discounted due to failing to respond to the authors' last few comments.

---

### Decision · Program_Chairs · 2025-01-22

Reject